# MCM8 interacts with DDX5 to promote R-loop resolution

Canxin Wen [1,2,3,4,5,6,7], Lili Cao[1,2,3,4,5,6,7], Shuhan Wang[1,2,3,4,5,6,7], Weiwei Xu[1,2,3,4,5,6,7], Yongze Yu [1,2,3,4,5,6,7], Simin Zhao[1,2,3,4,5,6,7], Fan Yang[8,9], Zi-Jiang Chen [1,2,3,4,5,6,7,10,11], Shidou Zhao [1,2,3,4,5,6,7✉], Yajuan Yang [1,2,3,4,5,6,7✉] & Yingying Qin [1,2,3,4,5,6,7✉]

## Abstract

**MCM8 has emerged as a core gene in reproductive aging and is crucial for meiotic homologous recombination repair. It also safeguards genome stability by coordinating the replication stress response during mitosis, but its function in mitotic germ cells remains elusive. Here we found that disabling MCM8 in mice resulted in proliferation defects of primordial germ cells (PGCs) and ultimately impaired fertility. We further demonstrated that MCM8 interacted with two known helicases DDX5 and DHX9, and loss of MCM8 led to R-loop accumulation by reducing the retention of these helicases at R-loops, thus inducing genome instability. Cells expressing premature ovarian insufficiency-causative mutants of MCM8 with decreased interaction with DDX5 displayed increased R-loop levels. These results show MCM8 interacts with R-loop-resolving factors to prevent R-loop-induced DNA damage, which may contribute to the maintenance of genome integrity of PGCs and reproductive reserve establishment. Our findings thus reveal an essential role for MCM8 in PGC development and improve our understanding of reproductive aging caused by genome instability in mitotic germ cells.**

**Keywords** Primordial Germ Cells; MCM8; R-loop; Genome Stability; Fertility
**Subject Categories** Development; DNA Replication, Recombination & Repair; RNA Biology

## Introduction

The reproductive lifespan is determined by the establishment and depletion rate of the reproductive reserve, i.e., the germ cell pool, and maintaining a sufficient germ cell pool and generating functional gametes are fundamental to reproduction and species propagation. Gametogenesis encompasses three crucial biological processes, namely primordial germ cell (PGC) development, meiosis, and gamete development, and any perturbations in these processes can affect the reproductive reserve and fertility. PGCs, the founders of the germline (Saitou and Yamaji, 2012), establish a finite pool for gametogenesis through mitotic proliferation, and therefore their proper development determines the reproductive reserve of both sexes. Exploring the molecular mechanisms underlying PGC development is therefore of paramount importance for understanding the processes behind premature reproductive aging such as premature ovarian insufficiency (POI) and non-obstructive azoospermia (NOA).

Recent advances in human genetics and animal models have shown that the DNA damage response (DDR) is one of the key biological pathways regulating reproductive lifespan (Ruth et al, 2021; Stolk et al, 2012; Titus et al, 2013). DDR is a conserved cellular mechanism that maintains genome stability and integrity. It is initiated upon the detection of DNA damage, and then an elaborate signaling network is activated to promote DNA repair (Blackford and Jackson, 2017; Lanz et al, 2019). Germ cells with a low mutation rate have super-stable genomes (Milholland et al, 2017), and DDR acts across the lifespan of germ cell development, as evidenced by the fact that defective DDR is associated with PGC loss, meiotic defects, and accelerated germ cell loss (Hill and Crossan, 2019; Huang et al, 2023; Ke et al, 2023). Among the identified DDR genes, Minichromosome Maintenance 8 Homologous Recombination Repair Factor (MCM8), a member of the MCM gene family, is one of the core genes highlighted in the regulation of human ovarian aging (Ruth et al, 2021), and the single

[1]State Key Laboratory of Reproductive Medicine and Offspring Health, Center for Reproductive Medicine, Institute of Women, Children and Reproductive Health, Shandong University, Jinan, Shandong 250012, China. [2]National Research Center for Assisted Reproductive Technology and Reproductive Genetics, Shandong University, Jinan, Shandong 250012, China. [3]Key Laboratory of Reproductive Endocrinology (Shandong University), Ministry of Education, Jinan, Shandong 250012, China. [4]Shandong Technology Innovation Center for Reproductive Health, Jinan, Shandong 250012, China. [5]Shandong Provincial Clinical Research Center for Reproductive Health, Jinan, Shandong 250012, China. [6]Shandong Key Laboratory of Reproductive Medicine, Shandong Provincial Hospital Affiliated to Shandong First Medical University, Jinan, Shandong 250012, China. [7]Research Unit of Gametogenesis and Health of ART-Offspring, Chinese Academy of Medical Sciences (No.2021RU001), Jinan, Shandong 250012, China. [8]Advanced Medical Research Institute, Meili Lake Translational Research Park, Cheeloo College of Medicine, Shandong University, Jinan, China. [9]Department of Physiology and Pathophysiology, School of Basic Medical Sciences, Shandong University, Jinan, Shandong 250012, China. [10]Shanghai Key Laboratory for Assisted Reproduction and Reproductive Genetics, Shanghai, China. [11]Department of Reproductive Medicine, Ren Ji Hospital, Shanghai Jiao Tong University School of Medicine, Shanghai, China. ✉E-mail: shidouzhao@sdu.edu.cn; YangYJ0204@sdu.edu.cn; qinyingying@sdu.edu.cn

nucleotide polymorphism rs16991615 in the MCM8 gene has been reported to be associated with early onset of menopause (Chen et al, 2012; Murray et al, 2011). Furthermore, there is increasing evidence supporting the causal relationship between functional defects of MCM8 and POI (AlAsiri et al, 2015; Bouali et al, 2017; Heddar et al, 2020; Tenenbaum-Rakover et al, 2015; Wang et al, 2020). Moreover, homozygous MCM8 loss-of-function variants have also been found in males with NOA (Kherraf et al, 2022; Tenenbaum-Rakover et al, 2015). All of these genetic studies suggest an indispensable role for MCM8 in gonadal development and function. It is known that MCM8 interacts with MCM9 to form a hexamer (MCM8/9), and HROB recruits and stimulates helicase activity of MCM8/9 complex that orchestrates homologous recombination (HR) repair by promoting RAD51 recruitment, facilitating strand invasion and sister chromatid exchange, and interacting with the MRN (MRE11-RAD50-NBS1) complex at DNA double-strand break (DSB) sites for DNA resection (Huang et al, 2020; Hustedt et al, 2019; Lee et al, 2015; Lutzmann et al, 2012; Nishimura et al, 2012; Park et al, 2013). For germ cells, MCM8 deficiency leads to meiotic defects, manifested as dramatically reduced crossovers and increased synapsis defects (Blanton et al, 2005; Lutzmann et al, 2012). These results support the hypothesis that premature reproductive aging in cases with MCM8 variants is attributable to meiotic HR defects in germ cells.

Intriguingly, single-cell transcriptome profiling of fetal gonads from both humans and mice shows that MCM8 is also highly expressed in PGCs undergoing rapid mitosis (Li et al, 2017; Niu and Spradling, 2020; Zhang et al, 2018), but its role in PGC development remains unknown. It has been reported that MCM8 binds to chromatin and acts as a backup DNA helicase to facilitate replication fork (RF) progression when the core replicative helicase subunit MCM2 is deficient (Maiorano et al, 2005; Natsume et al, 2017). Moreover, MCM8-deficient cells display a slower RF rate, inefficient RF restart, and excessive RF degradation under replication stress which is defined as the slowing or stalling of RF caused by various exogenous and endogenous factors (Griffin et al, 2022; Lutzmann et al, 2012). These studies suggest that MCM8 plays an essential role in the replication stress response to maintain genome stability during mitosis.

R-loop is a three-stranded structure containing a DNA-RNA hybrid and a displaced single-stranded DNA strand, which forms when the nascent RNA hybridizes with the template DNA strand. The abnormal accumulation of unscheduled R-loops can pose obstacles to both transcription and replication progressions, acting as a notable source of replication stress (Garcia-Muse and Aguilera, 2019). A great number of proteins have been identified as R-loop regulators to maintain genome stability (Cristini et al, 2018; Wang et al, 2018; Yan et al, 2022). Some of them prevent R-loop formation including RNA binding and processing factors (Salas-Armenteros et al, 2017; Wahba et al, 2011), whereas others resolve R-loops such as ribonuclease RNase H1 (Lockhart et al, 2019), and RNA helicase DDX5, DHX9 and DDX41 (Cristini et al, 2018; Mersaoui et al, 2019; Mosler et al, 2021). In addition, DNA repair factors have been identified to remove R-loop directly or indirectly during replication or repair (D'Alessandro et al, 2018; Hatchi et al, 2015; Yasuhara et al, 2018). Our previous study reported that the Fanconi anemia pathway was involved in resolving a high level of R-loops in rapidly dividing PGCs to maintain their genome stability (Yang et al, 2022). However, the regulating mechanism of R-loops in PGCs needs to be further determined.

In this study, we found that the number of actively proliferating PGCs was dramatically reduced in *Mcm8* knockout mice, and the accumulation of R-loops resulted in genome instability in MCM8-deficient cells. MCM8 interacted with DDX5 and DHX9, two R-loop resolving factors, and retained them at R-loop sites to promote R-loop removal, and cells expressing POI-causative MCM8 mutants with decreased interaction with DDX5 showed increased R-loop levels. Thus, we reveal a novel function of MCM8 in developing PGCs, and this extends our knowledge of premature reproductive aging in individuals with MCM8 deficiency.

## Results

### Loss of MCM8 causes PGC development defects and infertility

To investigate the role of MCM8 in the early germ cell and gonadal development, we generated *Mcm8* knockout (*Mcm8$^{-/-}$*, KO) mice using the CRISPR-Cas9 technology. Exon 7 of *Mcm8* was deleted, producing a frameshift mutation that generated a premature stop codon in exon 8 (Fig. EV1A). PCR was used for genotype validation of KO mice (Fig. EV1B), and western blotting showed the absence of MCM8 protein in the testis of adult KO mice (Fig. EV1C).

Consistent with a previous study (Lutzmann et al, 2012), KO mice were viable, grew normally (Fig. EV1D,E), and were born at the expected Mendelian ratio (Fig. EV1F). Both female and male adult KO mice were sterile (Fig. EV1G) due to the depletion of follicles or spermatogenic cells within the atrophied gonads (Fig. EV1H–K). It was noted that germ cells were dramatically reduced in KO ovaries and testes at postnatal day 3 (PD3) (Fig. EV1J,K), indicating that the loss of germ cells occurred in the fetal period. In the mouse embryo, the precursors of germ cells, i.e., PGCs that are specified around embryonic day 6.5 (E6.5), migrate and colonize the developing genital ridge at E10.5 and thereafter undergo several rounds of rapid mitotic division. At E13.5, after sex determination, oogonia undergo meiotic division in female embryos, whereas pro-spermatogonia undergo mitotic arrest in male embryos (Saitou and Yamaji, 2012). To gain insight into the role of MCM8 in gonad development during the fetal period, we investigated the germ cells and gonadal somatic cells at E13.5, E15.5, and PD3 in both female and male mice. While the expression of the pre-granulosa cell marker FOXL2 and the Sertoli cell marker SOX9 was not obviously different between wild type (WT) and KO mice, oogonia and pro-spermatogonia, both of which express the germ cell marker DDX4, were notably decreased at E13.5 in KO embryos (Fig. 1A,B), indicating that germ cell deficiency started from the PGC period. We subsequently observed the PGC population in KO and WT littermates on different embryonic days using alkaline phosphatase staining (Fig. 1C,D). The locations of the PGC population were almost identical across different periods, suggesting the normal migration of KO PGCs. No obvious difference was detected in the number of PGCs at E8.5 (95.2 ± 9.3 vs. 100.7 ± 10.4, $P > 0.05$), suggesting that the specification of PGCs in KO embryos was not affected. However, the number of PGCs in KO embryos was reduced by 56.4% at E9.5 (110.6 ± 26.1 vs. 253.9 ± 62.4, $P < 0.001$) and remained less than 30% at E11.5 (470.9 ± 169.1 vs. 1861.0 ± 591.9, $P < 0.0001$) when compared to WT embryos. The overall PGC population undergoes constant and

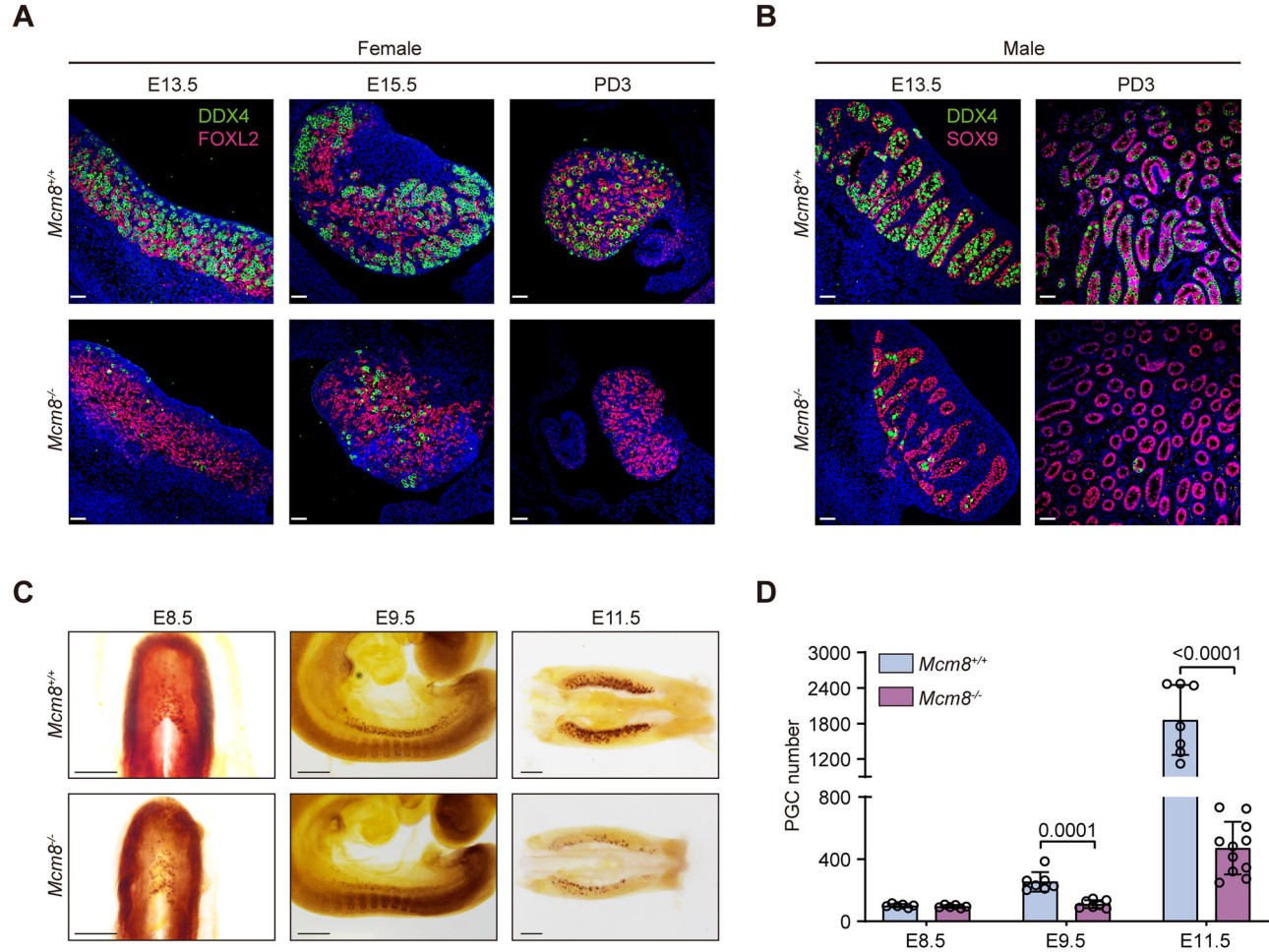

**Figure 1. Loss of MCM8 dramatically decreased the number of PGCs.**

(A) Immunostaining of DDX4 (a marker for late germ cells) and FOXL2 (a marker for granulosa cells) in the gonads of $Mcm8^{+/+}$ and $Mcm8^{-/-}$ mice at E13.5, E15.5, and PD3. Scale bars: 50 μm. (B) Immunostaining of DDX4 and SOX9 (a marker for Sertoli cells) in the gonads of $Mcm8^{+/+}$ and $Mcm8^{-/-}$ mice at E13.5 and PD3. Scale bars: 50 μm. (C) Representative images of alkaline phosphatase (a PGC marker) staining of whole-mount or genital ridges from $Mcm8^{+/+}$ and $Mcm8^{-/-}$ embryos. Scale bars: 200 μm. (D) PGC counts from $Mcm8^{+/+}$ and $Mcm8^{-/-}$ embryos at E8.5, E9.5, and E11.5. n = 6/6/7/7/7/11 embryos. Data information: In (D), data are presented as the mean ± SD, and the dots indicate individual embryos. The statistical significance of the difference was analyzed by unpaired two-tailed Student's t test (D), and the P-values were shown. Source data are available online for this figure.

rapid expansion from E9.5 to E12.5, a period when PGCs divide about twice per day (Kagiwada et al, 2013). These data support the critical role of MCM8 in rapidly proliferating PGCs.

It has been reported that the MCM8/9 complex participates in interstrand crosslink repair by functioning in RAD51-dependent HR repair downstream of the Fanconi anemia (FA) pathway (Nishimura et al, 2012), and *Mcm8* and *Mcm9* KO mice share the similar phenotype with FA-null mice of massive loss of proliferating PGCs (Luo and Schimenti, 2015; Yang et al, 2022). To determine whether MCM8 functions downstream of the FA pathway in vivo, we generated double mutant mice of *Mcm8* KO and ubiquitination-defective *Fancd2* (*Fancd2*$^{K559R/K559R}$) and quantified the PGC number at E11.5 using STELLA immunostaining. Compared to PGCs in WT littermates and single mutants of *Mcm8* KO or ubiquitination-defective *Fancd2*, significantly fewer PGCs were detected in the double mutants (1467.0 ± 307.4 vs. 436.0 ± 130.6 vs. 300.6 ± 117.8 vs. 43.2 ± 6.4, P < 0.001, Fig. EV1L),

suggesting that MCM8 has functions in PGC development that are independent of the FA pathway. The contrasting findings between mitotic cells in vitro and mitotic germ cells in vivo highlight the complexity of DNA replication progress in different contexts as well as the potential roles for MCM8 in these processes. Taken together, our findings demonstrated that profound PGC loss was a key reason for the infertility of *Mcm8* KO mice.

## MCM8 deficiency impairs proliferation and increases DNA damage in PGCs

To investigate the causes of the decreased numbers of PGCs in KO embryos, we examined the key events in PGC development at E11.5. Cleaved PARP1 staining showed that MCM8 deficiency resulted in a higher proportion of apoptosis when compared to WT group (1.05 ± 0.42% vs. 0.24 ± 0.14%, P < 0.05), but the number of apoptotic PGCs was less than 10 per embryo (Fig. EV2A). We next

investigated the impact of MCM8 deficiency on PGC proliferation. Almost all PGCs remained in the cell cycle in KO embryos, as shown by the immunofluorescence staining of Ki67 (Fig. EV2B). To further determine the extent of cell cycle progression, we performed a 5-ethynyl-2'-deoxyuridine (EdU) incorporation assay combined with cyclin B1 immunostaining. The proportion of S-phase PGCs showed a marked reduction in KO embryos compared to WT embryos (31.80 ± 3.39% vs. 46.30 ± 3.05%, $P < 0.0001$). Meanwhile, the proportion of G2-phase PGCs showed a significant elevation in KO embryos (43.07 ± 3.38% vs. 24.86 ± 2.72%, $P < 0.0001$) (Fig. 2A,B). The proportions of G1-phase PGCs and M-phase PGCs were comparable between KO and WT embryos. The decreased proportion of S-phase PGCs and increased G2-phase PGCs suggested that proliferation defects caused by an abnormal cell cycle was the leading cause of PGC loss in KO embryos. Because PGCs are sensitive to DNA damage (Hill and Crossan, 2019; Luo et al, 2014), we reasoned that PGCs had an increased requirement for MCM8 in genome stability maintenance under physiological conditions and that unrepaired DNA damage would result in apoptosis or cell cycle arrest in MCM8-deficient PGCs. We then employed 53BP1 and γH2AX immunostaining to identify DSB formation (Fig. 2C,E), and observed that a higher proportions of PGCs were positive for 53BP1 and γH2AX nuclear foci in KO embryos compared to WT embryos (18.79 ± 1.53% vs. 6.51 ± 2.21%, $P < 0.0001$; 26.96 ± 4.40% vs. 9.35 ± 3.09%, $P < 0.001$; Fig. 2D,F), indicating that unrepaired DSBs accumulated in KO PGCs. Ataxia Telangiectasia Mutated (ATM) is activated by phosphorylation when cells respond to DSBs (Bakkenist and Kastan, 2003). Detection of phosphorylated ATM at Serine 1981 in PGCs indicated increased DNA damage response in KO PGCs (31.87 ± 1.23 vs. 23.49 ± 1.74, $P < 0.01$; Fig. 2G,H).

An ordered sequence of biological events takes place in both migrating and colonized PGCs, including the progressive reduction of 5-methylcytosine (5mC) and histone H3 lysine 9 dimethylation (H3K9me2), upregulation of histone H3 lysine 27 trimethylation (H3K27me3), and increased proliferation rates and transcriptional output (Seki et al, 2007). To comprehensively evaluate the influence of MCM8 loss, we examined the transcription and epigenetic reprogramming processes. The transcriptional level assessed by 5-ethynyl uridine (EU) incorporation assay and the phosphorylation levels of both Ser2 and Ser5 within the C-terminal domain of RNA polymerase II (RNAPII) were comparable between WT and KO PGCs (Fig. EV2C), suggesting no obvious abnormalities in transcriptional upregulation in PGCs after MCM8 deficiency. The majority of both WT and KO PGCs lost DNA methylation and H3K9me2 modifications, and acquired high levels of H3K27me3 (Fig. EV2D), suggesting that the epigenetic reprogramming of KO PGCs was not obviously affected. Taken together, these results suggested that MCM8 deficiency caused PGC proliferation defects and DNA damage accumulation.

## MCM8 deficiency leads to an increase of R-loops in PGCs

Next, we performed a co-immunoprecipitation (co-IP) assay using an anti-MCM8 antibody in P19 cells followed by mass spectrometry (MS) analysis to identify potential interactors of MCM8 due to the scarcity of PGCs (Fig. EV3A). P19 cells are derived from mouse teratocarcinoma and share similar features with PGCs (Yu et al, 2023). We obtained the MCM8-specific interactome, which

included 109 proteins. We removed 25 ribosomal proteins that were always found among the non-specific binding proteins in the immunoprecipitation assay (Mellacheruvu et al, 2013; Mohammed et al, 2016) and analyzed the remaining 84 binding proteins (Dataset EV1). We noted that the MCM8 interactors were mainly involved in RNA splicing processes, as indicated by Gene Ontology (GO) analysis (Fig. 3A). To determine whether MCM8 was potentially involved in regulating RNA splicing, we performed alternative splicing (AS) analysis on RNA-seq data using rMATS. There were few mRNA splicing changes and the read mapping distribution was not obviously changed after MCM8 deficiency (Fig. EV4E,F), suggesting that MCM8 may not play an essential role in RNA splicing. Furthermore, 49 proteins among the MCM8 interactors were found in the known dataset of R-loop regulators (Lin et al, 2022), and the majority of the proteins are shown (Fig. 3B, Dataset EV1). Given that PGCs harbor higher R-loop levels compared to somatic cells (Yang et al, 2022), we hypothesized that the interactions between MCM8 and specific R-loop regulators regulate R-loop homeostasis in order to maintain PGC development. Two well-known RNA helicases DDX5 and DHX9, which can unwind RNA-DNA hybrids in R-loops (Lin et al, 2022) were selected as the targeting interactors (Fig. 3B).

To confirm the interactions, we conducted a co-IP assay in P19 cell extracts using an antibody against MCM8, and western blot results verified the interactions between MCM8 and DDX5, and DHX9 (Fig. 3C). Moreover, MCM8 could be detected in the co-IP assay in P19 cell extracts using antibodies against DDX5 or DHX9 (Fig. 3D). Besides, PCNA that was not found in the proteome did not interact with MCM8 (Fig. EV3C). We also conducted a comparative analysis for MCM8-immunoprecipitated proteins in *MCM8* KO HEK293 cells as a negative control, and confirmed the endogenous interactions between MCM8 and DDX5, as well as DHX9 (Fig. EV3D). Next, actively proliferating PGCs (STELLA-EGFP positive) at E12.5 were isolated by fluorescence-activated cell sorting, and the co-IP assay further confirmed the interactions between MCM8 and DDX5 and DHX9 in PGCs (Fig. 3E). Although the known partner MCM9 wasn't identified in the proteome, it was detected in the MCM8-immunoprecipitated proteins from P19 cell lysates (Fig. EV3B). Moreover, the presence of MCM9 interacting with DDX5 and DHX9 in P19 cells (Fig. EV3E) suggested the probability of a functional MCM8/9 complex in the R-loop resolution process.

It has been established that the recruitment of both DDX5 and DHX9 to R-loops can be enhanced by several interactors, which in turn promote R-loop resolution by regulating their DNA-RNA hybrid unwinding activity (Sessa et al, 2021; Yuan et al, 2021). Thus, MCM8, probably MCM8/9 might be involved in R-loop resolution by recruiting RNA helicases. To test this hypothesis, we first evaluated the level of R-loops in cells lacking MCM8. A significant accumulation of R-loops was observed in MCM8-deficient PGCs by immunostaining with the S9.6 antibody, which recognizes DNA-RNA hybrids (15.98 ± 7.71 vs. 10.31 ± 4.79, $P < 0.0001$) (Fig. 3F,G), as well as in MCM8-deficient mouse embryonic fibroblasts (MEFs, Fig. 4A,B). The S9.6 signals in KO MEFs were reduced by the overexpression of endoribonuclease RNase H1 which could resolve R-loops by specifically degrading the RNA in RNA-DNA hybrids (Fig. 4A,B). Furthermore, R-loop levels were evaluated by dot blot of nuclear DNA with S9.6 antibody, and knockout of MCM8, as well as knockdown of DDX5, led to significant increase of R-loops (Figs. 4C,D and EV5B). Although it

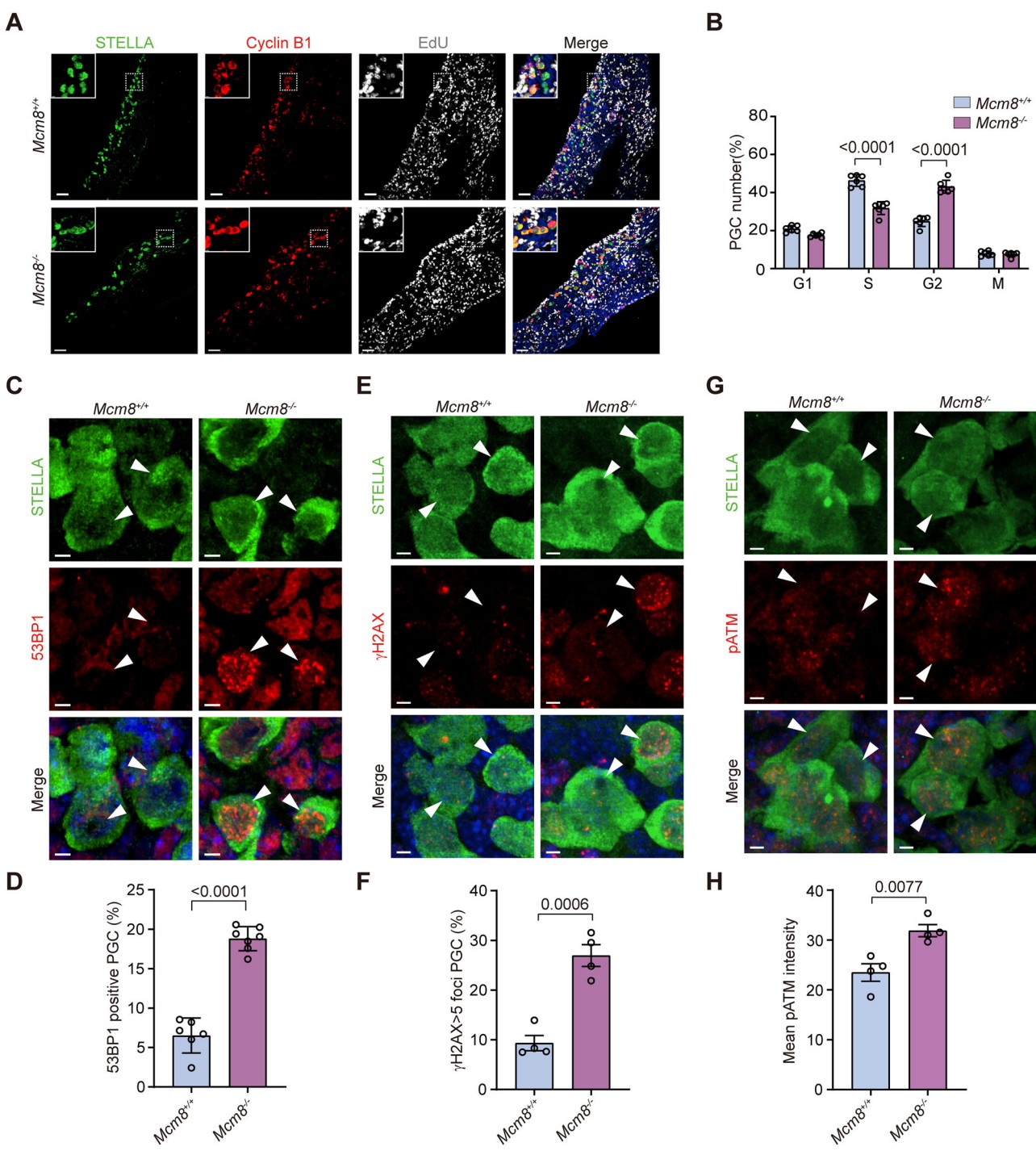

**Figure 2.** DNA damage accumulation led to decreased proliferation of $Mcm8^{-/-}$ PGCs.

(**A, B**) Representative images of EdU incorporation and immunostaining of cyclin B1 to determine the cell cycle phase of PGCs in E11.5 $Mcm8^{+/+}$ and $Mcm8^{-/-}$ genital ridges (**A**) and percentage of G1/S/G2/M-phase PGCs, respectively (**B**). G1-phase cells, cyclin B1 negative; S-phase cells, EdU positive; G2-phase cells, cyclin B1 expressed in the cytoplasm; M-phase cells, cyclin B1 expressed in the nucleus. Scale bars: 50 μm. Scale bars: 5 μm in the enlarged images. $n = 6/6$ embryos. (**C, D**) Representative images of 53BP1 immunostaining to observe the DNA damage in PGCs in E11.5 $Mcm8^{+/+}$ and $Mcm8^{-/-}$ genital ridges (**C**) and the percentage of 53BP1-positive PGCs (**D**). Scale bars: 2 μm. $n = 6/7$ embryos. (**E, F**) Representative images of γH2AX immunostaining in PGCs in E11.5 $Mcm8^{+/+}$ and $Mcm8^{-/-}$ genital ridges (**E**) and percentage of PGCs with more than 5 γH2AX nuclear foci (**F**). Scale bars: 2 μm. n = 4/4 embryos. (**G, H**) Representative images of pATM immunostaining in PGCs in E11.5 $Mcm8^{+/+}$ and $Mcm8^{-/-}$ genital ridges (**G**) and quantification of mean pATM signal intensity in PGCs (**H**). Scale bars: 2 μm. $n = 4/4$ embryos. Data information: STELLA-positive cells indicate PGCs. In (**B, D, F**), data are presented as the mean ± SD, and in (**H**), data are presented as the mean ± SEM, and dots indicate individual embryos. Dotted lines indicate positions of the enlarged images (**A**). Arrowheads indicate representative cells (**C, E, G**). The statistical significance of the difference was analyzed by unpaired two-tailed Student's *t* test (**B, D, F, H**), and the *P*-values were shown. Source data are available online for this figure.

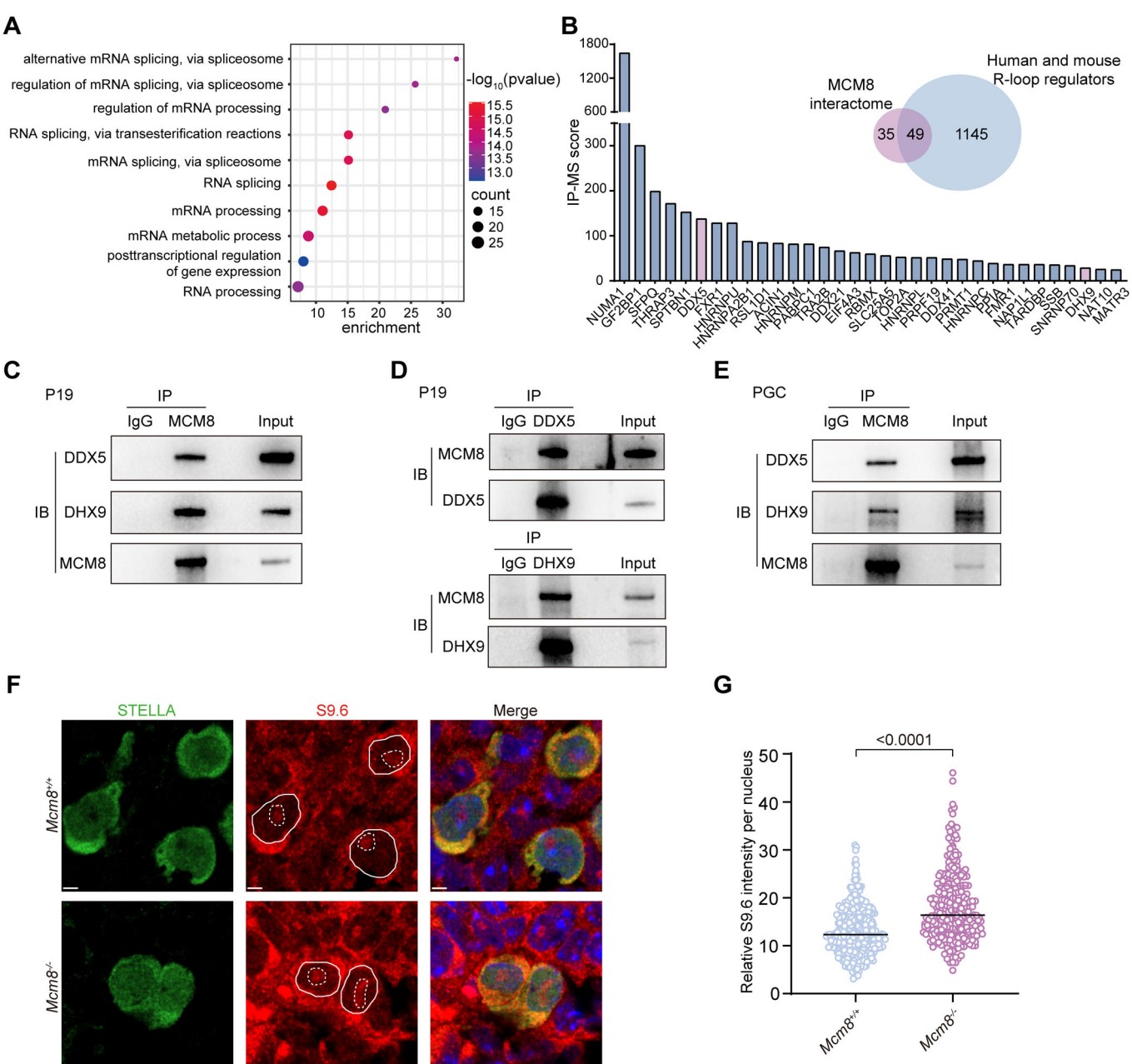

Figure 3. MCM8 interacted with R-loop regulators.

(A) GO enrichment analysis of the MCM8-specific interactome in P19 cells showing the top 10 enriched terms. (B) Overlap between the MCM8-specific interactome with human and mouse R-loop regulators collected in R-loopBase. Some interactors are shown in descending order according to the protein score. (C) Co-immunoprecipitation of endogenous MCM8 from P19 cell lysates, followed by immunoblot of DDX5, DHX9 and MCM8. IgG was used as a negative control. IP, immunoprecipitation. IB, immunoblotting. (D) Top: Co-immunoprecipitation of endogenous DDX5 from P19 cell lysates, followed by immunoblot of MCM8 and DDX5. Bottom: Co-immunoprecipitation of endogenous DHX9 from P19 cell lysates, followed by immunoblot of MCM8 and DHX9. IgG was used as a negative control. (E) Co-immunoprecipitation of endogenous MCM8 from lysates of E12.5 PGCs isolated by fluorescence-activated cell sorting, followed by immunoblot of DDX5, DHX9 and MCM8. IgG was used as a negative control. (F) Representative images of S9.6 immunostaining to evaluate the R-loop levels of PGCs in E11.5 $Mcm8^{+/+}$ and $Mcm8^{-/-}$ genital ridges. Scale bars: 3 μm. (G) Quantification of S9.6 average nuclear intensity after subtracting the nucleolar signal circled with the broken lines in PGCs. $n = 341/263$ PGCs from 5 embryos per group. Data information: STELLA-positive cells indicate PGCs. In (G), data are presented as the median with IQR, and dots indicate individual cells. The statistical significance of the difference was analyzed by two-tailed Mann–Whitney U test (G), and the P-value was shown. Source data are available online for this figure.

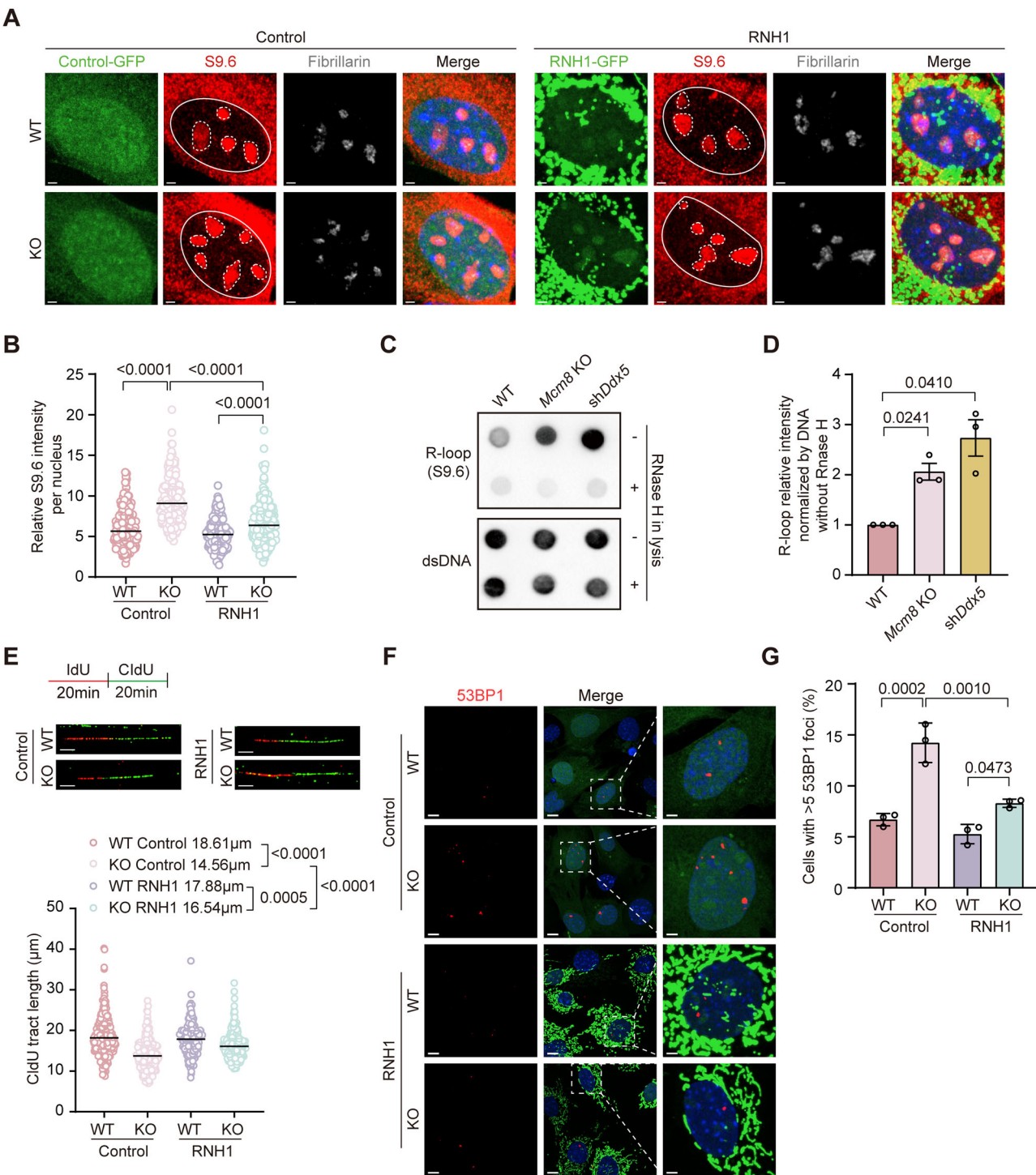

**Figure 4. Elevated R-loop levels induced increased DNA damage in KO MEFs.**

(A) Representative images of S9.6 immunostaining to evaluate the R-loop level of WT and KO MEFs overexpressing control or GFP-RNH1 adenovirus. Nucleoli were shown by fibrillarin staining. RNH1, RNase H1. Scale bars: 2 μm. (B) Quantification of S9.6 nuclear intensity that was exclusive of nucleolar signal circled with the broken lines. At least 150 cells were scored per group. Three independent experiments were conducted. (C) Dot-blot analysis on genomic DNA reflecting R-loop levels in WT, Mcm8 KO and DDX5-knockdown MEFs (as a positive control). Three independent experiments were conducted. (D) Quantification of R-loops by dot blotting of genomic DNA from MEFs. Double-stranded DNA (dsDNA) was used as an internal reference. (E) Top: Immunostaining of IdU and CldU after the DNA fiber assay to evaluate RF speed in WT and KO MEFs overexpressing control or GFP-RNH1 adenovirus. Scale bars: 5 μm. Bottom: Quantification of the CldU tract length. At least 150 DNA fibers were scored per group. Three independent experiments were conducted. (F) Representative images of 53BP1 immunostaining to evaluate the DNA damage level in WT and KO MEFs overexpressing control or GFP-RNH1 adenovirus. Scale bars:10 μm. Scale bars: 3 μm in the enlarged images. (G) Percentage of cells containing more than five 53BP1 foci. At least 150 cells were scored per group. $n = 3$ independent replicates. Data information: In (B, E), data are presented as the median with IQR, and in (D, G), data are presented as the mean ± SD. The statistical significance of the difference was analyzed by Kruskal–Wallis test followed by Dunn's multiple comparison test (B, E), paired $t$ test (D) and one-way ANOVA followed by Tukey's multiple comparison test (G), and the P-values were shown. Source data are available online for this figure.

was reported that R-loop levels were associated with high transcriptional levels in cells (Wahba et al, 2016), the global transcriptional output in MCM8-deficient PGC were not obviously affected when compared to WT PGCs (Fig. EV2C). To further confirm that the increased R-loop accumulation in MCM8-deficient cells was not caused by increased transcription, we detected the transcriptomic changes using bulk RNA sequencing and revealed distinct expression patterns between WT and KO MEFs by principal component analysis (PCA) (Figs. EV4A and EV4B). But the majority of genes (96%) were co-expressed (Fig. EV4C), and the gene expression distribution was comparable between WT and KO MEFs (Fig. EV4D), suggesting that the global transcription activity was not obviously affected after MCM8 deficiency. Furthermore, analysis of differentially expressed genes (DEGs) showed only 356 genes upregulated (1.4%) and 130 genes down-regulated (0.5%) in KO group (Fig. EV4G, Dataset EV2). We overlapped the DEGs found in each pair of KO sample and WT controls and identified 86 common genes which were enriched in stem cell differentiation and cell fate commitment (Figs. EV4H and EV4I, Dataset EV2). Taken together, these results suggested that MCM8 deficiency caused R-loop accumulation in PGCs without obviously altering the transcriptional levels.

## R-loop accumulation induces genome instability in MCM8-deficient cells

The unscheduled R-loops in proliferating cells are a crucial source of endogenous replication stress and, ultimately, genome instability. On the one hand, the displaced single-stranded DNA from the R-loop is susceptible to cleavage by nucleases, which generate DNA breaks (Cristini et al, 2019). On the other hand, the R-loop itself or evoked compact chromatin formation acts as a direct replication roadblock to hinder RF progression (Kemiha et al, 2021). Stalled RFs can then be processed by nucleases or ultimately collapse, thus resulting in DNA breaks (Zeman and Cimprich, 2014). To observe the impact of R-loop accumulation on RF progression in MCM8-deficient cells, a DNA fiber assay was performed to test the RF speed by adding the nucleotide analogs iododeoxyuridine (IdU) and chlorodeoxyuridine (CldU) to the MEF culture medium. The RF speed indicated by the length of the CldU track was slower in KO MEFs and was restored by overexpression of RNase H1 (Fig. 4E). These results suggested that increased R-loops after MCM8 deficiency indeed induced replication stress. The partial removal of R-loops by overexpression of RNase H1 might explain the partial recovery of the fork speed observed in MCM8 KO cells (Fig. 4B,E). However, besides R-loop accumulation, other mechanisms may also be involved in the decreased replication speed in MCM8 KO cells. Finally, consistent with the in vivo data, the absence of MCM8 caused an increase in DNA damage as indicated by 53BP1 immunostaining in MEFs, which was also reduced by RNase H1 overexpression (Fig. 4F,G). In addition, inhibiting R-loop formation with the transcription inhibitor cordycepin (Appendix Figs. S1A–D) partially improved replication fork speed and reduced DNA damage levels in MCM8 KO cells (Appendix Figs. S1E–H). These results indicated that the accumulation of unscheduled R-loops contributed to the slowed replication speed and increased DNA damage in MCM8-deficient cells.

## MCM8 retains DDX5 and DHX9 at R-loops

To further reveal the role of MCM8 in suppressing R-loop accumulation, we performed the cleavage under targets and tagmentation (CUT&Tag) assay using the S9.6 antibody to map genome-wide R-loops in MEFs derived from KO and WT embryos. The genomic distribution analysis displayed a similar pattern of R-loop signals between WT and KO MEFs (Fig. 5A). Notably, the transcription start site-related region had a high R-loop signal intensity (Fig. 5B). Overlap analysis showed that there were 24% shared R-loop peaks between WT and KO MEFs, and the KO group showed more unique peaks (Fig. 5C), indicating unscheduled R-loop formation. Further analysis of R-loop signals in different regions showed that R-loop signals in KO MEFs were stronger than those in WT MEFs in both the shared peak region and the KO-only peak region (Fig. 5D,E), indicating significant R-loop accumulation in cells lacking MCM8. Two representative genes *Chchd7* and *Armc10* with snapshots of R-loop signals for the subsets with shared peaks and the KO-only peaks, respectively, are shown (Fig. 5F). The biological processes of these genes including shared peak region with increased R-loop signals and KO-only peak region were enriched in the positive regulation of developmental growth according to the GO analysis (Fig. 5G). The R-loop accumulation may impair the expression of these genes, which could also affect cell proliferation.

To gain insight into the underlying molecular mechanism for how MCM8 suppresses R-loops, we performed the CUT&Tag assay in HEK293 cells transfected with *FLAG-MCM8* (Appendix Fig. S2A) using an anti-FLAG antibody as previously reported (Lyu et al, 2022), in which a control group using IgG was included. It was revealed that the highest proportion (33%) of the MCM8-binding sites were located in the promoter regions (Fig. 6A, Dataset EV3). GO analysis of MCM8-binding sites showed a notable enrichment in the regulation of mitotic cell cycle phase transition (Fig. 6B). To further determine the association between MCM8 and R-loops, we carried out genomic co-localization analysis between the MCM8-binding sites and R-loop signals in wild-type HEK293 cells. We found that about 10% (1008/10572) of the MCM8-binding sites co-occurred with R-loop signals (Fig. 6C; Dataset EV3), with the majority (82.8%, 835/1008) being located in the promoter regions. These genes were significantly associated with the regulation of mitotic cell cycle phase transition as indicated by the GO analysis (Fig. 6D). Two genes related to the cell cycle phase transition process, including *CDK4* and *GMNN*, are shown by snapshots (Fig. 6E). The cell cycle-regulated genes that follow a periodic pattern are highly expressed in rapidly dividing cells (Dominguez et al, 2016), and R-loop levels are positively associated with transcriptional activity (Wahba et al, 2016). Accordingly, loss of MCM8 in MEFs led to a pronounced accumulation of R-loops in promoters of active genes (Fig. EV5A), as displayed by the snapshots of the representative gene *Cdk4* which is related to the cell cycle phase transition process (Fig. EV5G). These results suggest that MCM8 coordinates R-loop regulation of cell cycle-related genes to promote cell division under unperturbed conditions. In addition, the specificity of sequencing strategy for R-loop mapping through the DNA moieties of DNA-RNA hybrids in CUT&Tag assay was confirmed by including a group with RNH1

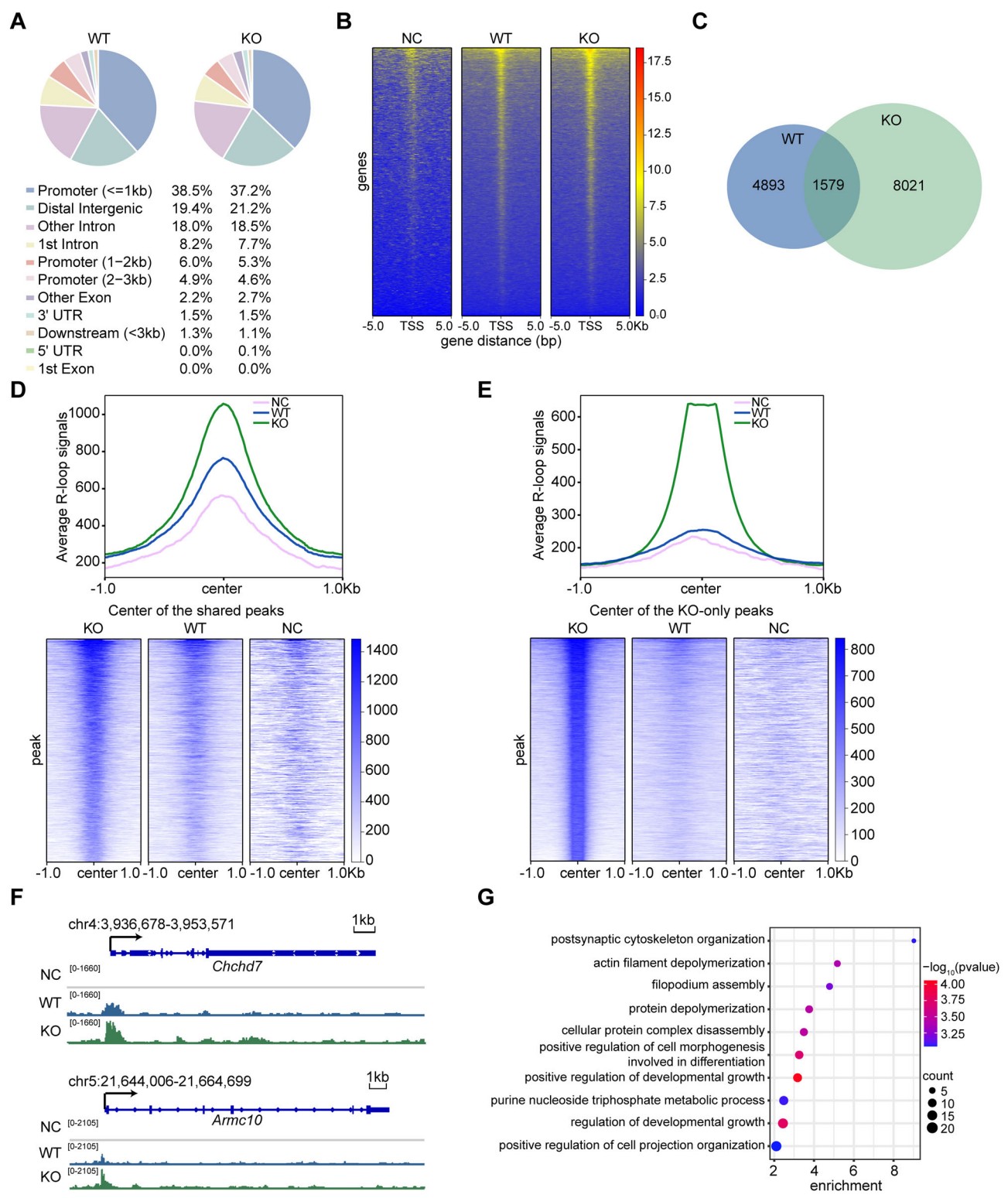

**Figure 5. Genome-wide analysis of R-loop accumulation upon MCM8 deficiency.**

(**A**) The genomic distribution of R-loop CUT&Tag peaks in *Mcm8*$^{+/+}$ (WT) and *Mcm8*$^{-/-}$ (KO) MEFs. UTR, untranslated region. (**B**) Heatmap plots of genes with R-loop signals across the 5 kb window around TSS in the NC, WT and KO group. NC, negative control that represented a group without primary antibody in the CUT&Tag assay. TSS, transcription start site. (**C**) Venn diagram of the overlap of R-loop peaks between the WT and KO group. (**D**, **E**) Average R-loop read density and heatmap of the WT group and KO group in the 1 kb window around the center of the shared peaks (**D**) and KO-only peaks (**E**). (**F**) Snapshots of R-loop signals in the representative genes *Chdhd7* and *Armc10* by genome browser tracks. NC (gray), WT (blue), and KO (green) CUT&Tag data were shown. (**G**) The GO enrichment terms of R-loop accumulated genes in KO group. Source data are available online for this figure.

overexpression (Appendix Fig. S2B, Dataset EV3). Although the R-loop signals were decreased slightly at some specific gene loci, the relative R-loop signals in RNH1 group were reduced compared to S9.6 group (Appendix Figs. S2C and S2D). Moreover, MCM8 was enriched in the co-IP assay in WT MEF extracts using the S9.6 antibody but not in the KO MEF extracts (Fig. 6F). These findings demonstrated that MCM8's binding preference for promoters played a role in R-loop regulation.

Given that MCM8 interacted with DDX5 and DHX9, we hypothesized that MCM8 modulates the functions of these RNA helicases to promote R-loop resolution. We firstly evaluated the protein expression of the two interactors in WT and KO cells, and no obvious differences were detected in the expression and localization of DDX5 and DHX9 as indicated by the western blot and immunofluorescence assays (Appendix Figs. S3A–C). We then determined whether the interactions affected their localization at R-loops. The S9.6 antibody-based co-IP showed that MCM8 deficiency reduced the amount of DDX5 and DHX9 at R-loops in comparison with the controls (Fig. 6F,G). We further applied the in situ proximity-ligation assay (PLA), in which two specific antibodies binding to their targets in close proximity generate fluorescence foci, to evaluate the functional distribution of DDX5 at R-loops. A dramatic reduction in anti-DDX5-S9.6 PLA signal was found in KO cells compared to WT cells (Fig. 6H,I). To further determine that the resolution of a fraction of R-loops depends upon MCM8-DDX5 interaction, we conducted a comparative analysis of genome-wide R-loop distribution between *Mcm8* KO and DDX5-knockdown MEFs. Consistent with previous studies (Sessa et al, 2021; Villarreal et al, 2020), elevated R-loop signals were increased in DDX5-knockdown MEFs (Fig. EV5C,D). The genome-wide analysis in MCM8-deficient cells and DDX5-knockdown cells showed that R-loop peak signals were both gained in the promoters, introns and intergenic regions (Fig. EV5E). The percentage of R-loop gain peaks at the promoters in DDX5-knockdown cells was more than that in MCM8-deficient cells, suggesting that R-loop gain peaks after DDX5-knockdown were more likely to be located closer to the promoters (Fig. EV5E). MCM8-deficient cells had 870 gain peaks that overlapped with the gain peaks of DDX5-knockdown cells and the majority of these gain peaks were overlaid with the promoters (Fig. EV5F), as shown by the snapshots of the representative gene *Cdk4* (Fig. EV5G), suggesting that MCM8 and DDX5 coordinately control R-loop resolution at specific genomic loci, especially at promoters. Collectively, these results suggested that MCM8 deficiency reduced its interactors' localization at R-loops, which compromised their functions in preventing detrimental R-loop accumulation.

### MCM8 mutants with reduced interaction with DDX5 cause R-loop accumulation

To determine the interacting domain of MCM8 with DDX5, we performed co-IP in HEK293 cells after transfecting them with different MCM8 deletion mutants and found that amino acids 379–840 of MCM8, including the AAA+ core domain (containing amino acids 402–609 that were characterized as the MCM domain) and winged-helix domain, were essential for binding to DDX5 (Fig. 7A,B). The reported mutations of MCM8 in POI and cancer patients are mostly located within the AAA+ core domain (61%), suggesting the central role of the AAA+ core domain in the

functional integrity of MCM8 (Griffin and Trakselis, 2019). In addition, MCM8 has been highlighted as a core gene in the regulation of ovarian aging, and women with mutations in *MCM8* are predisposed to POI (Ruth et al, 2021). To determine the implications of the mechanism described above in POI pathogenesis, we examined the interaction between DDX5 and POI-causative MCM8 mutants lacking these domains (Fig. 7C). Both p.R309* and p.S492* identified in POI patients were truncated proteins from mutations c.925 C > T and c.1475 C > A (NM_032485) that led to the premature termination of translation at Arginine 309 and Serine 492, respectively (Heddar et al, 2020; Wang et al, 2020). As expected, both p.R309* and p.S492* showed decreased interaction with DDX5 due to their deletions leading to truncated proteins. In contrast, p.E341K (rs16991615), which is associated with age at natural menopause (He et al, 2009), showed no decrease in its interaction with DDX5 compared to WT (Fig. 7C). As expected, overexpression of WT MCM8 in KO HEK293 cells showed a drastic decrease in R-loop levels compared to KO cells (Fig. 7D,E). While a slight decrease in R-loop level was also detected after p.E341K overexpression, the introduction of p.R309* and p.S492* still showed a comparable level of R-loops in their nuclei when compared to KO cells (Fig. 7D,E). These results suggested that MCM8 mutants showing impaired interactions with DDX5 could induce genome instability by impairing R-loop resolution, thus revealing a novel pathogenesis of POI patients with *MCM8* mutations.

## Discussion

In this study, we demonstrated that MCM8 helped retain DDX5 and DHX9 at R-loops to facilitate R-loop resolution, thus maintaining genome stability, which might be involved in PGC rapid proliferation and reproductive reserve establishment; loss of MCM8 promoted R-loop accumulation by reducing DDX5 and DHX9 at R-loops, which might contribute to the DNA damage accumulation and proliferation defects in PGCs (Fig. 7F). Moreover, POI-causative mutants of MCM8 that impaired the interaction with DDX5 resulted in increased R-loop levels. Our findings reveal a novel function of MCM8 in maintaining genomic stability in mitotic germ cells and provide a new mechanistic interpretation for premature reproductive aging in individuals with MCM8 deficiency.

It was reported that MCM8-deficient spermatocytes exhibited impaired meiotic recombination, and MCM8-deficient cells showed inefficient HR (Lutzmann et al, 2012; Nishimura et al, 2012). Thus, the loss of meiotic germ cells is considered to be a leading cause of infertility in *Mcm8*-null mice and individuals. However, carriers with *MCM8* loss-of-function homozygous variants display gonadal dysgenesis, presenting as invisible or very small ovaries and primary amenorrhea in POI patients (AlAsiri et al, 2015; Bouali et al, 2017; Tenenbaum-Rakover et al, 2015) and as Sertoli-cell-only syndrome instead of spermatogenic arrest at the meiotic stage in NOA patients (Kherraf et al, 2022). Similarly, obvious ovarian atrophy has been observed in newborn *Mcm8* KO female mice, and dramatically decreased germ cells have been observed in newborn *Mcm8* KO male mice when meiosis had not yet been initiated (Saitou and Miyauchi, 2016). These clues strongly indicate that mitotic germ cells are lost prior to meiosis in *Mcm8* KO embryos. Consistent with these observations, we found that the massive loss

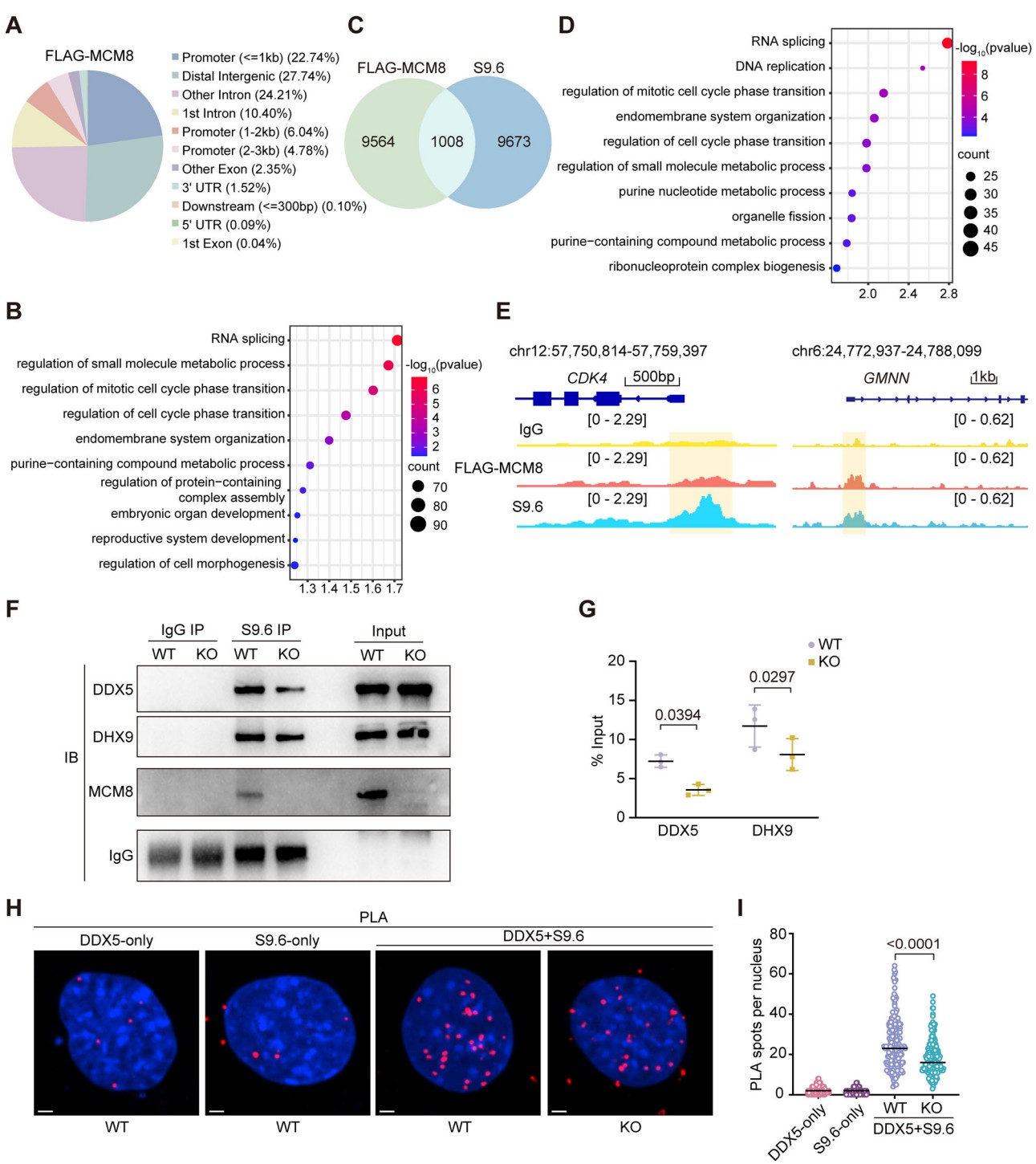

**Figure 6. MCM8 deficiency reduced DDX5 and DHX9 recruitment at R-loops.**

(A) The genomic distribution of FLAG CUT&Tag peaks in HEK293 cells after transfection with *FLAG-MCM8*. UTR, untranslated region. (B) The GO enrichment terms for MCM8-binding sites. (C) Venn diagram of the overlap of MCM8-binding sites and R-loop peaks in HEK293 cells. (D) The enriched GO terms for the overlapping genes. (E) Snapshots of FLAG-MCM8 and R-loop signals of the representative genes *CDK4*, and *GMNN* by genome browser tracks in HEK293 cells. IgG (yellow), FLAG-MCM8 (orange) and S9.6 (light blue) CUT&Tag data were shown. IgG was used as a control. (F) Detection of DDX5 and DHX9 immunoprecipitated with the S9.6 antibody in WT and KO MEFs by western blot. IgG was recovered to indicate equal amounts of antibody used in each group. (G) Quantitative analysis of the co-immunoprecipitated proteins between WT and KO MEFs. The immunoprecipitated DDX5 and DHX9 were compared with the input. $n = 3$ independent replicates. (H) Representative images of the in situ PLA between S9.6 and DDX5 antibodies. Single antibody-negative controls are shown. Scale bar: 10 μm. (I) Quantification of the number of PLA spots per nucleus. At least 200 cells were scored per group. Three independent experiments were conducted. Data information: In (G), data are presented as the mean ± SD, and in (I), data are presented as the median with IQR. The statistical significance of the difference was analyzed by paired $t$ test (G) and two-tailed Mann–Whitney U test (I), and the *P*-values were shown. Source data are available online for this figure.

of PGCs before entering meiosis in *Mcm8* KO embryos was the key reason for germ cell depletion in *Mcm8* KO mice. Because of the high conservation of MCM8 between humans and mice, our findings support the hypothesis that infertility in individuals with *MCM8* loss-of-function variants is due to PGC deficiency.

Notably, PGCs in *Mcm8* KO mice were specifically lost at E9.5 when transcription and proliferation were restored and replication stress was elevated (Kagiwada et al, 2013; Percharde et al, 2017; Yang et al, 2022). Although it has also been reported that MCM8 is involved in the replication stress response in mitotic cells (Griffin et al, 2022; Lutzmann et al, 2012), the source of endogenous replication stress has not been determined. We found that actively dividing PGCs were susceptible to MCM8 deficiency. Although active proliferation occurs in both PGCs and somatic stem cells in embryos, their characteristics are distinct. Rapid proliferation in PGCs is limited to the period between E9.5 and E13.5 in mouse embryos, and the maximum number of PGCs lays the foundation of the reproductive reserve of both sexes. Of note, because oocytes are not renewable, germ cell loss due to PGC development defects cannot be mitigated or rescued in females later in life. However, a subset of the somatic stem cells expands to renew and maintain the stem cell pools during their lifetime. Therefore, it is hypothesized that mitotic germ cells are more dependent on MCM8 to resolve specific replication stress, thus safeguarding genome stability. Recently, we have reported that rapidly proliferating PGCs are faced with constitutive replication stress, including high levels of R-loops and frequent RF stalling (Yang et al, 2022). In this study, we found that MCM8 prevented R-loop accumulation in order to maintain PGC genome stability. In addition, MCM8 has been reported to protect persistently stalled RFs from degradation by recruiting RAD51 (Griffin et al, 2022). These findings indicate that MCM8 is a potential multifunctional factor for attenuating replication stress.

Preventing excessive formation and ensuring the efficient removal of R-loops are two crucial mechanisms to avoid R-loop-induced genome instability. A great number of R-loop regulators have been identified through high-throughput assays (Cristini et al, 2018; Lin et al, 2022; Wang et al, 2018), among which DDX5 and DHX9 are the most commonly identified R-loop regulators that repress R-loop accumulation by directly unwinding the DNA-RNA hybrids (Chakraborty and Grosse, 2011; Mersaoui et al, 2019). Both proteins were identified as MCM8 interactors in PGCs in our study, and MCM8 deficiency reduced their associations with R-loops, indicating that MCM8 enhanced their retention or localization at R-loops. This is in line with the previous studies of R-loop regulation by DDX5 or DHX9, in which their function and activity have been shown to be delicately modulated by multiple co-factors (Li et al, 2020; Sessa et al, 2021; Yuan et al, 2021). For example, a breast cancer-associated BRCA2 variant reduces interactions with DDX5 and results in a decrease of DDX5 at R-loops (Sessa et al, 2021). In contrast, SOX2 facilitates somatic cell reprogramming by interacting with DDX5/DHX9 and preventing them from resolving R-loops (Li et al, 2020). Thus, R-loop dynamics are modulated by different mechanisms depending on the context in which the R-loops form. Here, we demonstrate a novel function for MCM8 in the regulation of R-loop resolution, which may be involved in PGC development. There is a strong interdependence between MCM8 and MCM9 because they stabilize each other by forming the MCM8/9 complex that is involved in DNA synthesis and HR repair

(Lee et al, 2015; Nishimura et al, 2012). Devoid of PGCs were also observed in *Mcm9* mutant mice (Luo and Schimenti, 2015) and our study found MCM9 interacted with DDX5 and DHX9; therefore, MCM8/9 complex may be involved in R-loop resolution.

MCM8 is a homolog of the MCM2-7 proteins that form the CMG complex together with CDC45 and GINS to allow the priming of eukaryotic DNA synthesis (Gozuacik et al, 2003). It has been reported that the MCM2-7 complex maintains genome stability by protecting the RF from unscheduled R-loops (Vijayr-aghavan et al, 2016) or preventing the formation of R-loops in the co-directional orientation (Hamperl et al, 2017). In our study, MCM8 deficiency induced the accumulation of aberrant R-loops in PGCs. Furthermore, there was an enrichment of R-loop signals particularly at the promoter region in MCM8-deficient cells. Interestingly, the results showed only 24% of R-loop peaks were overlapped between WT and KO MEFs, and more unique peaks in the KO cells were found. Usually, transcription and replication in the nucleus are coordinated spatially and temporally to minimize conflict between the two processes in the higher eukaryotes (Hamperl and Cimprich, 2016). On the one hand, defective transcription termination induces a redistribution of replication initiation factors in the G1-phase (Gros et al, 2015). On the other hand, dysregulated origin firing increases R-loop formation (Hamperl et al, 2017; Lang et al, 2017). Since MCM8 depletion could reduce chromatin loading of some replication initiation proteins (Gambus and Blow, 2013), we speculated that a large number of unscheduled R-loop signals and loss of shared signals with WT in KO cells were relevant to the repositioned RNA polymerases or the changed distribution of replication origins. Additionally, 10% of the MCM8-binding sites co-occurred with R-loop signals under physiological conditions. As for the low percentage of the MCM8-binding sites being associated with R-loop signals, it is likely that the immunoprecipitation sequences encompass all MCM8-binding sites involved in either replication fork elongation, HR or R-loop resolution in the CUT&Tag assay. Moreover, co-transcriptional R-loops generated across all the cell cycle phases have been detected all over the genome and their levels are associated with the global transcriptional activity. Given that MCM8 predominantly functions in the S-phase (Gozuacik et al, 2003; Maiorano et al, 2005) and prefers binding to R-loops in the promoter regions, MCM8 may be only responsible for resolving a specific subset of S-phase R-loops. The MCM domain can unwind the DNA-RNA hybrid structures while translocating along the single-stranded DNA in vitro (Shin and Kelman, 2006), which raises the possibility that replicative helicases may resolve R-loops directly. We found that MCM8 functioned at R-loop-containing sites by interacting with the R-loop regulators, and this extended our understanding of the role of the MCM family in R-loop homeostasis. However, whether MCM8 can unwind DNA-RNA hybrids directly requires further exploration. Our current results demonstrate that MCM8 maintains R-loop balance by interacting with R-loop regulators, thus diversifying its functions in genome stability maintenance.

More recently, increasing evidence has linked R-loops to both meiotic and mitotic germ cells, revealing the physiological function of R-loop dynamics at different stages of germ cell development. During meiosis, both the normal formation and timely removal of R-loops are important for meiotic DSB repair (Yang et al, 2021). The accumulation of R-loops induced by the deletion of *Rnaseh1*

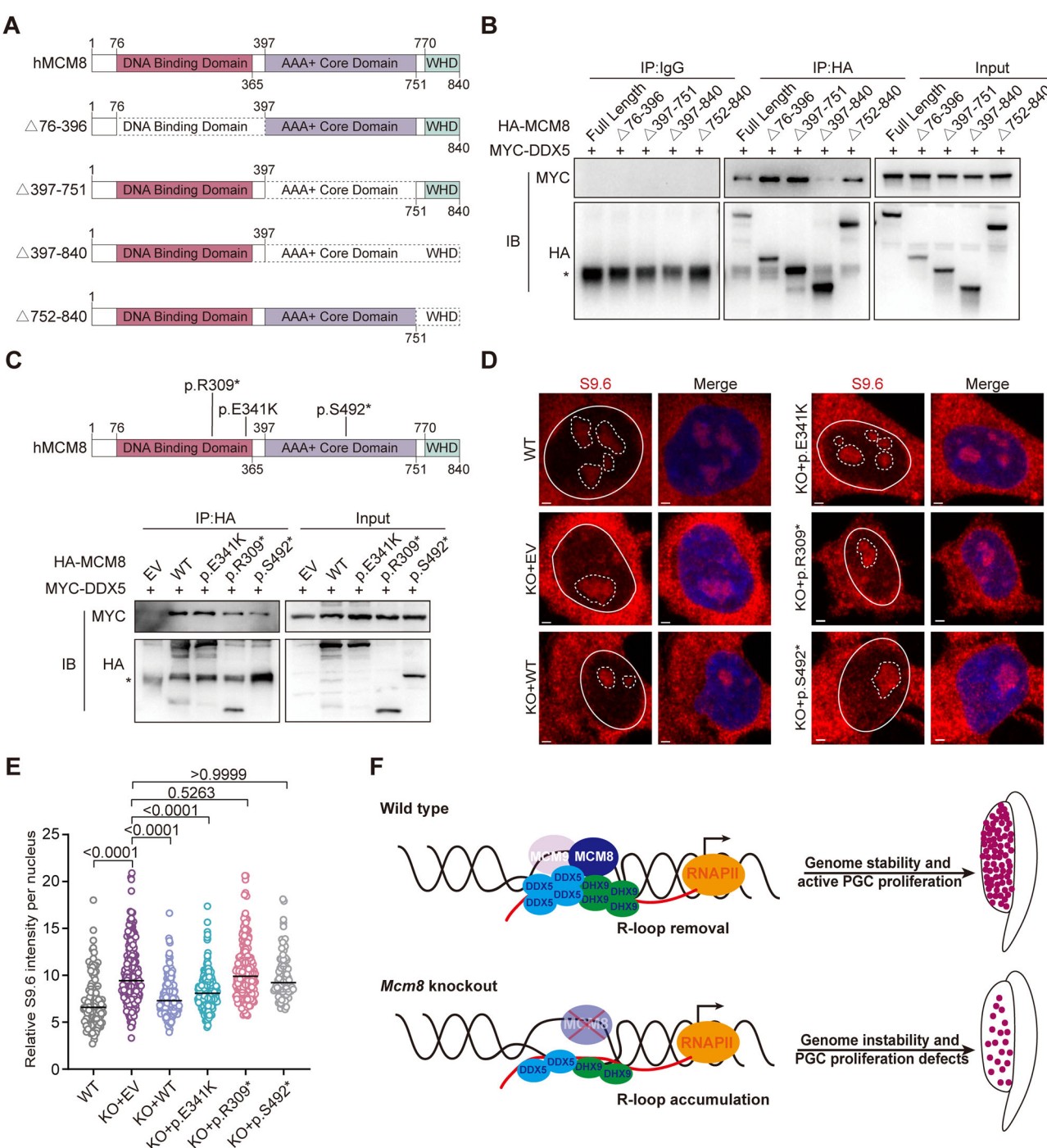

**Figure 7. MCM8 mutants showed reduced interaction with DDX5 and increased R-loops.**

(A) Schematic diagrams of full-length and defective MCM8 expression constructs. WHD, winged-helix domain. (B) Co-immunoprecipitation analysis of the interaction between MYC-DDX5 and different HA-MCM8 mutants with domain deletions. Three independent experiments were conducted. (C) Top: Overview of different POI-causative mutations in MCM8. Bottom: Detection by western blot of MYC-DDX5 co-immunoprecipitated proteins from *MCM8* knockout HEK293 cells overexpressing an empty vector (EV), WT, p.E341K, p.R309*, or p.S492*. Three independent experiments were conducted. (D) Representative images of S9.6 immunostaining to evaluate the R-loop levels in WT HEK293 and in *MCM8* knockout HEK293 cells after transfection of EV, WT, p.E341K, p.R309*, or p.S492*. Scale bars: 2 μm. (E) Quantification of S9.6 nuclear intensity after subtracting the nucleolar signal circled with the broken lines. Cells transfected with plasmid were scored per group. Three independent experiments were conducted. (F) Working model. R-loop formation is enhanced in actively dividing PGCs due to hypertranscription. In WT PGCs, MCM8, probably together with MCM9, helps retain DDX5 and DHX9 at R-loops through its interaction with them, which facilitates R-loop resolution, thus maintaining genome stability and ensuring the active proliferation of PGCs. In *Mcm8*[−/−] PGCs, the impaired interaction of MCM8 with DDX5 and DHX9 leads to reduced R-loop removal. Unscheduled R-loop accumulation increases genome instability and ultimately leads to PGC proliferation defects. Data information: Asterisk (*) indicates a non-specific band of IgG heavy chain. In (E), data are presented as the median with IQR. The statistical significance of the difference was analyzed by a Kruskal–Wallis test followed by Dunn's multiple comparison test (E), and the *P*-values were shown. Source data are available online for this figure.

caused transcriptional dysfunction of meiotic genes (Jiang et al, 2022) and impaired meiotic recombination by altering the recruitment of RAD51 and DMC1, resulting in male infertility (Liu et al, 2023). For mitotic germ cells, we have reported that increased R-loop levels is a crucial source of endogenous genome threats in rapidly proliferating PGCs, and this feature underlies why PGCs are more dependent on the FA pathway to safeguard their genome stability (Yang et al, 2022; Yu et al, 2023). In fact, impaired proliferation of PGCs was also observed in other ubiquitously expressed DNA repair gene knockout mouse models (Luo and Schimenti, 2015; Messiaen et al, 2013). These findings suggest that PGCs could be considered as a group of special cells that are more dependent on DNA repair factors to respond to endogenous genome threats, thus maintaining a high level of genome stability. In this study, we further found that PGCs lacking MCM8 had R-loop accumulation and our results from somatic cellular models suggested that MCM8 might also resolve R-loops by interacting with DDX5 and DHX9 in PGCs.

In addition, DDX5 knockdown resulted in R-loop accumulation, and the gain peaks overlapped in DDX5 knockdown and MCM8 knockout cells were much more enriched at the promoters, suggesting a coordinate regulation of R-loops at promoters of specific genes by DDX5 and MCM8. However, it's unknown about the function of DDX5 in PGCs. A conditional knockout (cKO) strategy will help elucidate the specific role of DDX5 in PGC development due to embryonic lethality of *Ddx5*-null mice. Interestingly, recent studies demonstrated that DDX5 played essential transcriptional and post-transcriptional roles in the maintenance and function of spermatogonia (Legrand et al, 2019; Xia et al, 2021). Collectively, both MCM8 and DDX5 play crucial roles in the development of germ cells at different stages.

The molecular basis of accelerated reproductive aging caused by MCM8 variants is attributed to genome instability caused by HR defects in meiotic germ cells (Lutzmann et al, 2012; Yatsenko and Rajkovic, 2015). However, in our study MCM8 mutants identified in POI patients, such as p.R309* and p.S492*, showed decreased interactions with DDX5 and led to the accumulation of R-loops, which might result in DNA damage and PGC proliferation defects. Thus, we revealed a new explanation for the infertility induced by these MCM8 mutants. In addition, we also showed that p.E341K (rs16991615), which had normal interaction with DDX5, did not show equally efficient R-loop removal as WT MCM8. This suggests that the single nucleotide polymorphism of the *MCM8* gene associated with age at natural menopause also impairs R-loop removal, perhaps by changing the interaction with proteins other than DDX5. Therefore, our study provides a novel interpretation for the pathogenesis of POI patients with *MCM8* mutations.

In summary, we show here that MCM8 coordinates R-loop regulation by enhancing retention of R-loop resolving factors to protect genome stability from accumulated R-loops, which may be required for PGC rapid proliferation and reproductive reserve establishment. DNA damage induced by increased R-loops after disabling MCM8 may contribute to PGC proliferation defects and infertility. Our findings extend our understanding of the function of MCM8 in germ cell development and the pathogenesis of MCM8-related infertility, and also give insights into the regulatory network involved in the maintenance of genome stability in mitotic germ cells.

# Methods

## Mice

*Mcm8*$^{+/-}$ mice were generated by Gempharmatech Company using the CRISPR-Cas9 technology in the C57BL/6 genetic background. *Fancd2*$^{+/K559R}$ mutant mice and *Stella-EGFP* reporter mice were generated by Cyagen Bioscience as described previously (Yang et al, 2022). All the founder mice were crossed with WT ICR mice to obtain the mice with a C57BL/6J × ICR mixed genetic background that were used in this study. *Mcm8*$^{+/-}$ female and male mice were then mated to obtain *Mcm8*$^{-/-}$ embryos or mice. *Mcm8*$^{+/-}$ and *Fancd2*$^{+/K559R}$ mice were crossbred to obtain double mutant embryos. All mice were kept under specific pathogen-free conditions at 22–26 °C with humidity at 40–70% and under a constant 12 h light/dark cycle. Genomic DNA extracted from the tail or embryonic head was used for genotyping by PCR. The PCR primers were as follows: *Mcm8*-F1 (CAG TTC ACA TTC CCA TCA GCA GTG), *Mcm8*-R1 (GCA AGT TGT GTC CAA CTT TTC CTA TG), *Mcm8*-F2 (CTG CAT GTG GAG AGA TTC AGA GC), *Mcm8*-R2 (GGA CAG TGC TCT GCC TAC CAA CTG), *Fancd2*$^{K559R}$-F (ATT CCA CAG TGA TTC TCA AGT CCT), *Fancd2*$^{K559R}$-R (GAT CTC CCA AGT TGC AAA GTA GAC), *Stella*-F1 (CCT TTC ATT AGT GTG AAG GTG CTG), *Stella*-R1 (ACA AAA GCT CAT TGG CCT AGT CTT), *Stella*-F2 (AGG AAG TCT GGT TAT TGA AGC AGT), and *Stella*-R2 (CTT TAA CAG AGA GAA GTT CGT GGC). All animal experiments were approved and conducted according to the ethical guidelines of the Animal Care and Research Committee of Shandong University (Approval No. 2022-21).

## Antibodies

The primary antibodies used for immunoprecipitation (IP) and western blotting (WB) were rabbit anti-MCM8 (5 μg for IP, 1:1000 dilution for WB; Proteintech 16451-1-AP), rabbit anti-DDX5 (5 μg for IP, 1:1000 dilution for WB; Abcam ab21696), rabbit anti-DHX9 (5 μg for IP, 1:900 dilution for WB; Abcam ab26271), rabbit anti-HA (5 μg for IP, 1:1000 dilution for WB; Cell Signaling Technology 3724), mouse anti-MYC (1:1000 dilution for WB; Cell Signaling Technology 2276), mouse anti-FLAG (1:1000 dilution for WB; Sigma-Aldrich F1804), mouse anti-S9.6 (5 μg for IP; Kerafast ENH001), normal rabbit IgG (5 μg for IP; Cell Signaling Technology 3700), normal mouse IgG (5 μg for IP; Proteintech B900620), and mouse anti-β-actin (1:5000 dilution for WB; Proteintech 66009-1-Ig).

The primary antibodies used for immunostaining were goat anti-STELLA (1:50 dilution; Novus Biologicals AF2566), goat anti-DDX4 (1:50 dilution; Novus Biologicals AF2030), rabbit anti-FOXL2 (1:250 dilution; Abcam ab246511), rabbit anti-SOX9 (1:200 dilution; Millipore AB5535), rabbit anti-cleaved PARP1 (1:100 dilution; Cell Signaling Technology 94885), rabbit anti-Ki67 (1:200 dilution; Abcam ab15580), rabbit anti-Cyclin B1 (1:200 dilution; Cell Signaling Technology 4138), rabbit anti-53BP1 (1:500 dilution; Novus Biologicals NB100-304), rabbit anti-γH2AX (1:50 dilution; Cell Signaling Technology 9718), rabbit anti-pATM (1:100 dilution; Cell Signaling Technology 5883), rabbit anti-pSer2 Pol II (1:100 dilution; Abcam Ab193468), rabbit anti-pSer5 Pol II (1:100

dilution; Abcam Ab193467), rabbit anti-5mC (1:100 dilution; Abcam Ab214727), rabbit anti-H3K9me2 (1:100 dilution; Millipore 07-441), rabbit anti-H3K27me3 (1:100 dilution; Millipore 07-449), rabbit anti-S9.6 (1:200 dilution; Kerafast Kf-Ab01137-23.0), goat anti-S9.6 (1:400 dilution; Kerafast Kf-Ab01137-24.1), mouse anti-Fibrillarin (1:500 dilution; Abcam ab4566), rabbit anti-DDX5 (1:400 dilution; Abcam ab21696), rabbit anti-DHX9 (1:400 dilution; Abcam ab26271), and mouse anti-HA (1:200 dilution; Cell Signaling Technology 2367). Primary antibodies were detected with Alexa Fluor 488-, 568-, or 637-conjugated secondary antibodies (1:800 dilution; Invitrogen).

The primary antibodies used for CUT&Tag were mouse anti-S9.6 (1 µg for $10^5$ cells; Kerafast ENH001), mouse anti-FLAG (1 µg for $10^5$ cells; Sigma-Aldrich F1804) and anti-mouse IgG (1 µg for $10^5$ cells; Abcam ab37355). The primary antibodies used for PLA were mouse anti-S9.6 (1:6000 dilution; Kerafast ENH001) and rabbit anti-DDX5 (1:9000 dilution; Abcam ab21696). The primary antibodies used for dot blot were mouse anti-S9.6 (1:500 dilution; Kerafast ENH001) and mouse anti-dsDNA (1:500 dilution; Santa Cruz sc-58749).

## Generation of MEFs

$Mcm8^{+/-}$ female mice were mated with $Mcm8^{+/-}$ males. When copulation plugs were examined, the time of embryo development was defined as E0.5. Pregnant female mice were euthanized by $CO_2$ inhalation at E13.5, and embryos were transferred to PBS-filled culture plates in a clean environment. After saving a piece of each fetus tissue for genotyping, the head, tail, limbs, heart, and liver were discarded, and the fetus trunk was collected for generating MEFs. The fetus trunk was minced thoroughly and digested in 0.05% trypsin (Invitrogen) in a 37 °C water bath for 4 min. Dissociated cells were spun down and resuspended and cultured in Dulbecco's modified eagle medium (DMEM) containing 10% FBS and 1% streptomycin/penicillin. Infection with lentivirus SV40gp6 (Genechem) was used for cell immortalization. After 48 h, MEFs were treated with puromycin for 3 days. Selected MEFs were cultured and used for the experiments.

## Generation of the MCM8 knockout cell line

*MCM8* knockout human embryonic kidney 293 (HEK293) cells were generated using the CRISPR/Cas9 technology. Briefly, the MCM8 targeting plasmid was constructed by inserting an MCM8-sgRNA into the PsPca9(BB)-2A-Puro (PX459) plasmid. MCM8-sgRNA sequences were as follows: MCM8-sg1F(CAC CGC TCT AAA AGC CCC ACT ATC T), MCM8-sg1R (AAA CAG ATA GTG GGG CTT TTA GAG C), MCM8-sg2F (CAC CGA GAT GTT GGC TAG AAC TGA G), and MCM8-sg2R (AAA CCT CAG TTC TAG CCA ACA TCT C). After transfection of the targeting plasmid for 48 h, the HEK293 cells were selected with puromycin (0.5 mg/ml) for 4 days. Selected HEK293 cells were seeded in 96-well plates at a limiting dilution, and single clones were cultured. Genotypes were confirmed using Sanger sequencing of genomic DNA from single clones of HEK293 cells. Finally, an *MCM8* knockout clone with the c.254-233_486+150del1027bp mutation was selected and used for all subsequent experiments.

## Cell culture and treatment

The P19 cell line was purchased from the Cell Resource Center, IBMS, CAMS/PUMC, China, and maintained in MEMα medium with 10% FBS and 1% streptomycin/penicillin. MEFs were derived from E13.5 $Mcm8^{+/+}$ and $Mcm8^{-/-}$ embryos as described above. MEFs and HEK293 cells were cultured in DMEM with 10% FBS and 1% streptomycin/penicillin and passaged every 2 days.

Infection with RNase H1-adenovirus (VectorBuilder) was used for removing endogenous R-loops by cleaving the RNA moiety of the DNA-RNA hybrid. Infection with sh*Ddx5*-adenovirus (Vector-Builder) (targeted sequence: CTA ACT CCT CAG AGG ATT ATA) was used for downregulation of DDX5. After the MEFs were expanded, the cells were plated in a 6-well plate and infected with adenovirus at 60% confluence. Twelve hours later, the culture medium was replaced with fresh medium. After 48 h, cells were reseeded in a 6-well plate or on glass coverslips (Nest) for the DNA fiber assay or for immunostaining. For inhibiting transcription in cells, 40 µM cordycepin (Selleck, S3610) was added to the medium for 2 h before collection or fixation. For EU incorporation assay, 1 mM EU (Invitrogen, C10330) was added to the medium for 1 h before fixation.

The full-length cDNA and different deletions of *MCM8* were purchased from YOUBAO Biology and were cloned into pcDNA-3.1-FLAG-N or pcDNA-3.1-HA-N for transduction. The p.E341K, p.R309*, and p.S492* MCM8 mutant overexpression plasmids were generated by YOUBAO Biology and confirmed by sequencing. The mutations were identified by sequence alignment with the NCBI Reference Sequence (NM_032485). After HEK293 cells were expanded, they were plated in a 6-well plate or 10 cm dish and transfected with plasmids at 60% confluence using Lipo3000 (Invitrogen). After 48 h, the cells were collected for protein extraction or immunostaining.

## Genital ridge section preparation and immunostaining

Pregnant female mice were euthanized at E11.5, E13.5, or E15.5, and embryos were fixed in 4% PFA at 4 °C overnight. For EdU incorporation experiment, pregnant females were intraperitoneally injected with 100 mg/kg EdU reagent for 1 h before being euthanized. For EU incorporation assay, genital ridges were cultured in the medium (MEMα, 10% Knockout Serum Replacement, 1.5 µM 2-O-α-D-glucopyranosyl-L-ascorbic acid and 1% streptomycin/penicillin) containing 1 mM EU for 1 h, followed by fixation with 4% PFA for 1 h at room temperature. Genital ridges or embryonic gonads were dissected and then embedded in OCT (Invitrogen) for sectioning at 10 µm thickness with a freezing microtome (Thermo).

For immunostaining, the sections or the cells fixed on glass coverslips were immersed in PBS twice and then permeabilized and blocked in 25% donkey serum containing 0.3% Triton X-100 at room temperature for 1 h. The sections were then incubated with the primary antibody diluted in 1% bovine serum albumin containing 0.3% Triton X-100 at 4 °C overnight. After rinsing in PBS-T (PBS containing 0.1% Triton X-100) three times, the sections were incubated with the relevant Alexa Fluor secondary antibodies diluted in PBS-T supplemented with Hoechst 33342 at room temperature for 1 h. After rinsing, the sections were mounted with an antifade reagent (Invitrogen). For EdU

incorporation experiment, the sections were washed with PBS and incubated with 2 mg/mL glycine solution, permeabilized twice with 0.5% Triton X-100, and incubated with freshly prepared EdU reaction solution for 30 min at room temperature following the manufacturer's instructions (Ribo Bioscience, C10371). For EU incorporation assay, the sections were washed with PBS, permeabilized once with 0.5% Triton X-100, incubated with Click-it EU reaction mixture and finally washed once with Click-it reaction rinse buffer following the manufacturer's instructions (Invitrogen, C10330). For 5mC immunostaining, the sections were incubated with 2 N HCl at 37 °C for DNA denaturation. The sections were then subjected to standard immunofluorescence staining as described above. Samples were photographed by a Z-axis scan using a Dragonfly spinning disc confocal microscope (ANDOR Technology).

## Hematoxylin and eosin staining

Ovaries, testes, and epididymides of $Mcm8^{-/-}$ and $Mcm8^{+/+}$ mice were collected and fixed in 4% PFA or Bouin's solution at 4 °C overnight and then dehydrated and embedded in paraffin. The samples were cut into sections of 5 μm thickness for standard H&E staining. Histological images were obtained using an Olympus BX53 microscope.

## Alkaline phosphatase staining

Pregnant female mice were sacrificed at E8.5, E9.5, and E11.5, and embryos were fixed in 4% PFA for 1 h. Embryos were then washed with PBS and 25 mM Tris-Maleate buffer (pH 9.0), and immersed in freshly prepared working buffer [(25 mM Tris-Maleate buffer (pH 9.0), 0.5 mM $MgCl_2$, 0.4 mg/mL 1-Naphthyl phosphate disodium salt, and 1 mg/mL Fast Red TR salt (Sigma)] in the dark for 15–30 min at 37 °C, and the staining was terminated by adding an excess volume of PBS. Embryos were washed with PBS followed by incubation in 40% glycerol for 1 h and 80% glycerol overnight. Images were obtained using a stereoscope.

## PGC sorting by fluorescence-activated cell sorting

Homozygous *Stella-EGFP* male mice were mated with WT ICR females. Genital ridges were isolated from embryos at E12.5 using a stereoscope and then transferred to cold Leibovitz's L-15 medium with 10% FBS and 1% streptomycin/penicillin. The collected genital ridges were dissociated into single cells with Accutase (Gibco, A1110501) for 22 min at 37 °C, and the digestion was quenched with an excessive volume of Leibovitz's L-15 medium containing 10% FBS. The cell suspension was filtered through a 70 μm cell strainer (Miltenyi Biotec) and centrifuged at $300 \times g$ and 4 °C for 8 min, and the supernatant was removed. The cell pellets were resuspended in DPBS supplemented with 0.5% bovine serum albumin and 2 mM EDTA (pH 8.0). The EGFP-positive PGCs were sorted on a FACSAria III (BD Biosciences) cell sorter and collected in MEMα medium containing 15% KnockOut Serum Replacement and 1% strepto-mycin/penicillin. For the IP experiment, the sorted cells were centrifuged at $300 \times g$ for 8 min, and the cell pellets were frozen and preserved at −80 °C.

## IP assay

Cells cultured in 10 cm dishes were washed with PBS twice and incubated with ice-cold Pierce IP lysis buffer (Invitrogen) containing protease and phosphatase inhibitor cocktails at 4 °C for 30 min on a shaker. The sorted PGCs were removed from the −80 °C freezer and incubated with ice-cold lysis buffer as described above. After sonication, the cell lysates were cleared by centrifugation at $12,000 \times g$ for 10 min, and the supernatants were retained as the cell extracts. The cell extract was mixed with the primary antibody and rotated at 4 °C overnight. The mixture was then incubated with protein A/G beads and rotated at 4 °C for 2 h. Protein-bound beads were subsequently collected using a magnetic frame and washed three times with IP lysis buffer. After the final wash, the supernatants were removed and 1 × SDS sample buffer was added. The samples were boiled at 100 °C for 10 min, and the supernatants were collected as immunoprecipitated proteins for MS or WB.

For MS, immunoprecipitated proteins from the IgG and MCM8 groups were separated by 8% SDS-polyacrylamide gel electrophoresis (PAGE) and stained with Coomassie brilliant blue (Beyotime Biotechnology). After washing overnight, protein gel slices were cut off for acquiring all the potential interactors of MCM8. And then in-gel digestion was performed. The gel slices were washed in 100 mM ammonium bicarbonate/100% acetonitrile (volume ratio 1:1) and dehydrated with 100% acetonitrile for 5 min. The proteins were reduced with 10 mM dithiothreitol and incubated at 56 °C for 1 h. The gel slices were then dehydrated in 100% acetonitrile before being incubated with 55 mM iodoacetamide at room temperature for 45 min in the dark. After being washed and dehydrated again, the gel slices were incubated with 10 ng/μl trypsin in 50 mM ammonium bicarbonate on ice for 1 h, followed by digestion with trypsin at 37 °C overnight. Peptides were extracted and dried and finally dissolved in 2% acetonitrile/0.1% formic acid for MS.

## MS analysis

Peptides were separated on an EASY-nLC 1000 UPLC system with a reversed-phase analytical column (15 cm length, 75 μm pore size) with a gradient of 0.55 μl/min with 0.1% formic acid in 2% acetonitrile (solvent A) and 0.1% formic acid in 98% acetonitrile (solvent B): 0–22 min, from 6% to 35% B; 22–26 min, from 35% to 80% B; 26–30 min, hold at 80% B. The separated peptides were subjected to NSI Ion Source followed by tandem mass spectrometry in Q Exactive Plus (Thermo Scientific). MS scans were acquired on the Orbitrap using an *m/z* window from 350 to 1800 and a resolution of 70,000. The 20 most intense precursor ions were selected for the detection of MS/MS spectra in the ion trap using collision-induced dissociation with a normalized collision energy of 28%, and the fragments were acquired in the Orbitrap at a resolution of 17,500. The automatic gain control was set at 5E4.

The MS/MS data files were processed for protein identification using Proteome Discoverer 2.4, and tandem mass spectra were searched against the *Mus musculus* database. The enzyme digestion method was set as Trypsin/P. Carbamydomethylation of cysteines was defined as a fixed modification, and oxidation of methionines was specified as a variable modification. The mass error tolerance was set to 10 ppm for precursor ions and 0.02 Da for fragment matching. The peptide confidence was high, and the score of peptide ion was greater than 20. The mass spectrometry

proteomics data have been deposited to the ProteomeXchange Consortium via the PRIDE partner repository with the dataset identifier PXD050727. The list of the proteins identified by mass spectrometry is included in the supplementary data (Dataset EV1).

## WB

Cells were collected and lysed in SDS lysis buffer supplemented with a protein inhibitor cocktail and boiled at 100 °C for 10 min. Cell extracts or immunoprecipitated proteins were loaded onto 8% or 10% SDS-PAGE gels. After electrophoresis, the proteins were separated and transferred onto polyvinylidene fluoride membranes (Millipore Corporation) and blocked in 5% skim milk for 1 h at room temperature before being incubated with the corresponding primary antibody at 4 °C overnight. After washing three times with TBS-T (TBS, pH 7.4 with 0.1% Tween-20), the membranes were incubated with horseradish peroxidase-conjugated secondary antibodies against rabbit or mouse IgG (Proteintech) for 1 h at room temperature. The membranes were treated with an ECL reagent (Millipore), and the blots were visualized using a ChemiDoc MP System (Bio-Rad).

## RNA-seq

Total RNA was extracted from WT and KO MEFs using TRIzol reagent (Invitrogen) following the manufacturer's instructions with three biological repeats. RNA integrity was assessed using the RNA Nano 6000 Assay Kit of the Bioanalyzer 2100 system (Agilent Technologies, CA, USA). Total RNA for each sample was used to prepare RNA libraries using NEBNext Ultra™ RNA Library Prep Kit for Illumina following manufacturer's protocols, and library quality was assessed on the Agilent Bioanalyzer 2100 system. Raw data (150 bp paired-end reads) were generated by the Illumina NovaSeq 6000.

## RNA-seq data processing and AS analysis

Raw reads were processed to remove adapters and perform quality control, and clean data with high quality were mapped to the mus_musculus_Ensembl_106 assembly using Hisat2 (V2.0.5). For gene expression quantification, reads numbers mapped to each gene were counted by featureCounts v1.5.0-p3, and number of Fragments Per Kilobase of transcript sequence per Millions base pairs sequenced (FPKMs) were calculated. Differentially expressed genes (DEG) between WT and KO group were measured by DESeq2 (1.20.0), and DEGs found in each pair of KO and WT sample were measured by edgeR, which were defined as above two-fold changes, with an adjusted $P$ value of <0.05. GO enrichment analysis was performed using the clusterProfiler R package. AS events were analyzed by rMATS (4.1.0) software and were classified into five types, including skipped exon (SE), retained intron (RI), mutually exclusive exons (MXE), alternative 5′ splice site (A5SS), and alternative 3′ splice site (A3SS). A false discovery rate (FDR) < 0.05 were categorized as differential alternative splicing events.

## DNA fiber assay

Cells grown on plates were treated with 25 μM 5-iodo-2′-deoxyuridine (IdU; Sigma-Aldrich, I7125) for 20 min and washed

with PBS. Cells were then labeled with 250 μM 5-chloro-2′-deoxyuridine (CldU; Sigma-Aldrich, C6891) for 20 min, collected, and resuspended at a concentration of $5 \times 10^5$–$1 \times 10^6$ cells/ml in cold PBS. As described previously (Yang et al, 2022), 2 μl of cell suspension was used to prepare slides for the DNA fiber assay. After being completely air-dried, the slides were fixed with methanol/acetic acid (at a 3:1 ratio) for 10 min. Fixed slides were washed twice in distilled water and immersed in 2 N HCl at 37 °C. Immunofluorescent staining was performed as described above using anti-BrdU antibodies (rat and mouse). DNA fiber images were captured on a confocal microscope (ANDOR Technology).

## Dot blot

Cells were permeabilized with cytoskeleton (CSK) buffer [10 mM PIPES pH 6.8, 100 mM NaCl, 3 mM $MgCl_2$, 0.3 M sucrose, and 1× protease inhibitor cocktail (Complete, EDTA-free; Sigma)] supplemented with 0.5% Triton X-100, followed by SDS/proteinase K buffer digestion at 37 °C overnight. Then nucleic acids were extracted by phenol-chloroform and precipitated by ethanol. A total of 2 μg DNA samples for each group were incubated with or without RNase H (New England Biolabs, M0297) at 37 °C for 4 h, and blotted onto NC membrane. After air-dried, the membranes were blocked with 5% skim milk before being immunoblotting with the primary antibody at 4 °C overnight. After washing, the membranes were incubated with horseradish peroxidase-conjugated secondary antibodies. The blots on membranes were collected by a ChemiDoc MP System (Bio-Rad) after treating with an ECL reagent (Millipore), and quantified using ImageJ software.

## PLA

Cells grown on glass coverslips (Nest) were permeabilized by CSK buffer supplemented with 0.5% Triton X-100, and then fixed with 4% PFA at room temperature for 15 min and washed three times with PBS. In situ PLA was carried out following the manufacturer's directions (Duolink, Sigma, DUO92008). Briefly, cells were blocked for 1 h at room temperature using Duolink Blocking Solution and then incubated with the primary antibodies for 1 h at 37 °C. After washing in PBS for 10 min, the cells were incubated with the secondary antibody mixture (Duolink In Situ PLA probes anti-mouse plus and anti-rabbit minus) for 1 h at 37 °C and washed with 1× buffer A. The samples were then incubated with the ligation mixture prepared according to the manufacturer's protocol for 30 min at 37 °C and washed with 1× buffer A. The samples were then incubated with the amplification mixture in the dark for 100 min at 37 °C and washed twice with 1× buffer B for 10 min and once with 0.01× buffer B for 1 min. Cells were incubated with Hoechst 33342 to label the nuclei before being mounted onto slides. Images were acquired with a confocal microscope (ANDOR Technology).

## CUT&Tag assay

The CUT&Tag experiments were performed according to the manufacturer's protocol using the Hyperactive Universal CUT&Tag Assay Kit for Illumina (Vazyme Biotech, TD904). In brief, $1 \times 10^5$ cells were collected, washed once with wash buffer at room

temperature, and resuspended with 100 µl wash buffer. Cells were immobilized on 10 µl activated concanavalin A-coated magnetic beads at room temperature for 10 min. The bead-bound cells were then incubated in 50 µl antibody buffer with 1 µg primary antibody or no antibody (as a negative control) at 4 °C overnight. The next day, the primary antibody buffer was removed and the cells were incubated in 50 µl DIG-wash buffer with 0.5 µl anti-mouse antibody for 1 h at room temperature with slow rotation. After washing three times with DIG-wash buffer, the cells were incubated in 98 µl DIG-300 buffer with 2 µl pA/G-Tnp for 1 h at room temperature with slow rotation. After washing three times with DIG-300 buffer, the cells were incubated in 40 µl DIG-300 buffer with 10 µl 5 × TTBL at 37 °C for 1 h. Then 2 µl 10% SDS were added and further incubated at 55 °C for 10 min to stop the fragmentation. Then, the genomic DNA then was transferred to pre-activated DNA Extract Beads Pro and incubated 20 min at room temperature. After washing with B&W buffer twice and air-dried, the samples were dissolved in nuclease-free water. The genomic DNA was amplified with barcoded i5 and i7 primers (Vazyme Biotech, TD202) using the recommended PCR program (72 °C for 3 min; 98 °C for 3 min; 12 cycles of 98 °C for 10 s, 60 °C for 5 s, and 72 °C for 1 min; and holding at 12 °C). VAHTS DNA Clean Beads (Vazyme Biotech, N411) were used to clean up the PCR library products, and the CUT&Tag libraries were sequenced on an Illumina NovaSeq 6000 platform.

### CUT&Tag data analysis

Low-quality reads and PCR adapters were removed, and high-quality mapped reads in MEFs were mapped to the mm10 reference genome using bowtie2 software (v2.2.4), respectively. The track files were generated using The BamCoverage command in deepTools (v3.5.1), and all peak calling was performed with MACS2 (v2.2.7.1) without using the "no-antibody" track as the input track. The annotation of CUT&Tag peaks was performed with the ChIPseeker R package, and the genomic distribution of the peaks was analyzed. Heatmaps and metaplots were made for the specific peaks and protein-coding genes using deepTools. Differential R-loop analysis was performed using MAnorm to obtain the shared and unique peaks. To quantify the R-loop signal of expressed genes in the samples, we used the peak set of expressed genes from the RNA-sequencing analysis in MEFs. The fold change was calculated using the average normalized coverage (rpkm) of the regions by using R/Bioconductor packages. Then enrichment around the TSS were created using deepTools. R-loop gain peaks were determined based on the FC > 2 and $P$ value < 0.05. The enriched peaks were visualized in IGV software. GO analysis were performed by ClusterProfiler R packages.

For observation of co-localized peaks between MCM8-binding sites and R-loop signals, clean reads in HEK293 cells were aligned to the hg38 reference genome using bowtie2 software (v2.2.4). All peak calling was performed with MACS2 software (v2.2.7.1). Then, peak annotation was performed with ChIPSeeker R package and significative peaks were determined based on the FC > 2 and $P$ value < 0.05 (Dataset EV3). The co-localized peaks were determined by Intervene software (Dataset EV3). GO analysis were performed by ClusterProfiler R packages. The track files of normalized reads were generated with bamCoverage function of deepTools (v3.5.1) for visualization.

### Statistical analysis

Graphs for all figures and statistical analyses were constructed in GraphPad Prism 8 (GraphPad Software). The number of replicates, mean, and error bars are shown in the figure legends. Data are presented as mean ± SD, mean ± SEM or median with interquartile range (IQR). When the numbers of data were more than 20, a normality test was used to determine whether they were normally distributed. The quantitative data between groups were analyzed by unpaired two-tailed Student's $t$ test, paired $t$ test, one-way ANOVA followed by Tukey's multiple comparison test, one-way ANOVA followed by Dunnett's multiple comparison test, two-tailed Mann–Whitney U test, or Kruskal–Wallis test followed by Dunn's multiple comparison test. $P < 0.05$ was considered statistically significant.

## Data availability

The bulk RNA sequencing data are available in Gene Expression Omnibus (GEO) database with the accession number GSE254004. The CUT&Tag sequencing data are available in Gene Expression Omnibus (GEO) database with the accession number GSE254352. Microscope images are available in the Bioimage Archive with the accession number S-BIAD1029 and in the source data. Protein interaction IP-MS data are available in PRIDE PXD050727.

The source data of this paper are collected in the following database record: biostudies:S-SCDT-10_1038-S44318-024-00134-0.

## Peer review information

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

## Acknowledgements

The authors thank Prof. Ping Zheng from Kunming Institute of Zoology, Chinese Academy of Sciences for providing help in technical support of DNA fiber assay and dot blot assay. Research was funded by grants from the National Key Research and Development Program of China (2022YFC2703800 and 2021YFC2700100), National Natural Science Foundation for Distinguished Young Scholars (82125014), National Natural Science Foundation of China (32170867, 32370906, 32200697 and 82371645), Basic Science Center Program of NSFC (31988101), Natural Science Foundation of Shandong Province for Distinguished Young Scholars (ZR2023JQ031) and Grand Basic Projects (ZR2021ZD33), Shandong Provincial Key Research and Development Program (2020ZLYS02), CAMS Innovation Fund for Medical Sciences (2021-I2M-5-001), and Taishan Scholars Program for Young Experts of Shandong Province (tsqn202211370).

## Author contributions

**Canxin Wen**: Data curation; Formal analysis; Validation; Investigation; Writing—original draft. **Lili Cao**: Formal analysis; Investigation; Visualization. **Shuhan Wang**: Formal analysis; Investigation; Visualization. **Weiwei Xu**: Investigation; Visualization. **Yongze Yu**: Investigation; Visualization. **Simin Zhao**: Investigation; Visualization. **Fan Yang**: Formal analysis; Validation. **Zi-Jiang Chen**: Resources; Supervision; Funding acquisition; Project administration. **Shidou Zhao**: Conceptualization; Funding acquisition; Investigation; Methodology; Writing—review and editing. **Yajuan Yang**: Conceptualization; Resources; Funding acquisition; Methodology; Writing—review and editing. **Yingying Qin**: Conceptualization; Supervision; Funding acquisition; Project administration; Writing—review and editing.

Source data underlying figure panels in this paper may have individual authorship assigned. Where available, figure panel/source data authorship is listed in the following database record: biostudies:S-SCDT-10_1038-S44318-024-00134-0.

## Disclosure and competing interests statement

The authors declare no competing interests.

# Expanded View Figures

**Figure EV1.  MCM8 deficiency caused germ cell depletion.**

(**A**) Top: Schematic diagram of the *Mcm8* gene targeting strategy, in which a 1479 bp segment including Exon 7 of the *Mcm8* gene was deleted. The black scissors refer to the targeted gRNA regions and the arrows indicate the primers for genotyping. Bottom: Schematic diagram of the wild-type MCM8 protein and the predicted mutant protein in *Mcm8*$^{-/-}$ mice. MCM domain: minichromosome maintenance domain. (**B**) Genotyping results of the *Mcm8*$^{+/+}$, *Mcm8*$^{+/-}$, and *Mcm8*$^{-/-}$ mice. (**C**) Expression validation of MCM8 in the testes of the adult *Mcm8*$^{+/+}$ and *Mcm8*$^{-/-}$ mice by western blot. β-Actin was used as the loading control. (**D**) Representative images of *Mcm8*$^{+/+}$ and *Mcm8*$^{-/-}$ mice at 3 months. (**E**) Quantification of body weight to observe the growth of *Mcm8*$^{+/+}$ and *Mcm8*$^{-/-}$ mice at 3 months. $n = 10/10/10/10$. (**F**) The genotype and sex ratios of the offspring generated by mating *Mcm8*$^{+/-}$ male and female mice. $n = 185$ pups. (**G**) Cumulative pup number from 5 months of mating female or male *Mcm8*$^{+/+}$/*Mcm8*$^{-/-}$ mice with WT mice, respectively. $n = 6/6/6/6$. (**H**) Gross morphology of ovaries (left) and the ovary/body weight ratio (right) in adult *Mcm8*$^{+/+}$ and *Mcm8*$^{-/-}$ females. Scale bar, 1 mm. $n = 9/9$. (**I**) The gross morphology of the testes (left) and the testis/body weight ratio (right) in adult *Mcm8*$^{+/+}$ and *Mcm8*$^{-/-}$ males. Scale bar, 1 mm. $n = 9/9$. (**J**) Hematoxylin and eosin staining of *Mcm8*$^{+/+}$ and *Mcm8*$^{-/-}$ paraffin sections of ovaries from PD3, PD21, and 3 M females. 3 M, 3-month-old. Scale bars, 100 μm. (**K**) Hematoxylin and eosin staining of *Mcm8*$^{+/+}$ and *Mcm8*$^{-/-}$ paraffin sections of testes and epididymis from PD3 and 3 M males. 3 M, 3-month-old. Scale bars: 50 μm. (**L**) Immunostaining of STELLA (a marker for PGCs) in the genital ridges from WT, *Mcm8*$^{-/-}$, *Fancd2*$^{K559R/K559R}$, and *Mcm8*$^{-/-}$*Fancd2*$^{K559R/K559R}$ embryos and qualification of PGC numbers from E11.5 embryos with the indicated genotypes. Scale bars: 50 μm. $n = 5/5/5/5$ embryos. Data information: In (**E, H, I, L**), data are presented as the mean ± SD, and the dots indicate individual mice. The statistical significance of the difference was analyzed by unpaired two-tailed Student's *t*-test (**E, H, I**) and with one-way ANOVA followed by Dunnett's multiple comparison test (**L**), and the *P*-values were shown. Source data are available online for this figure.

                                                                   

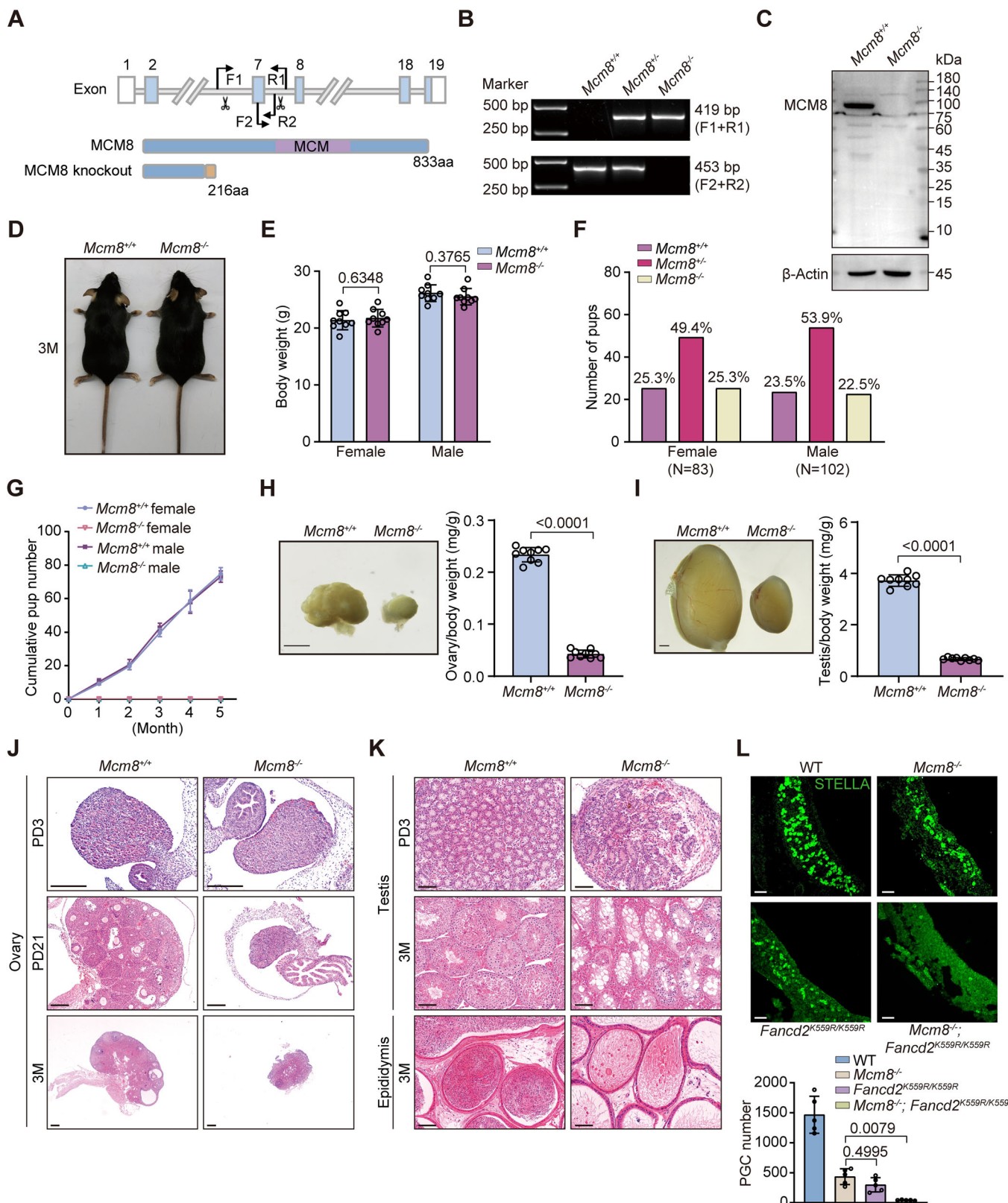

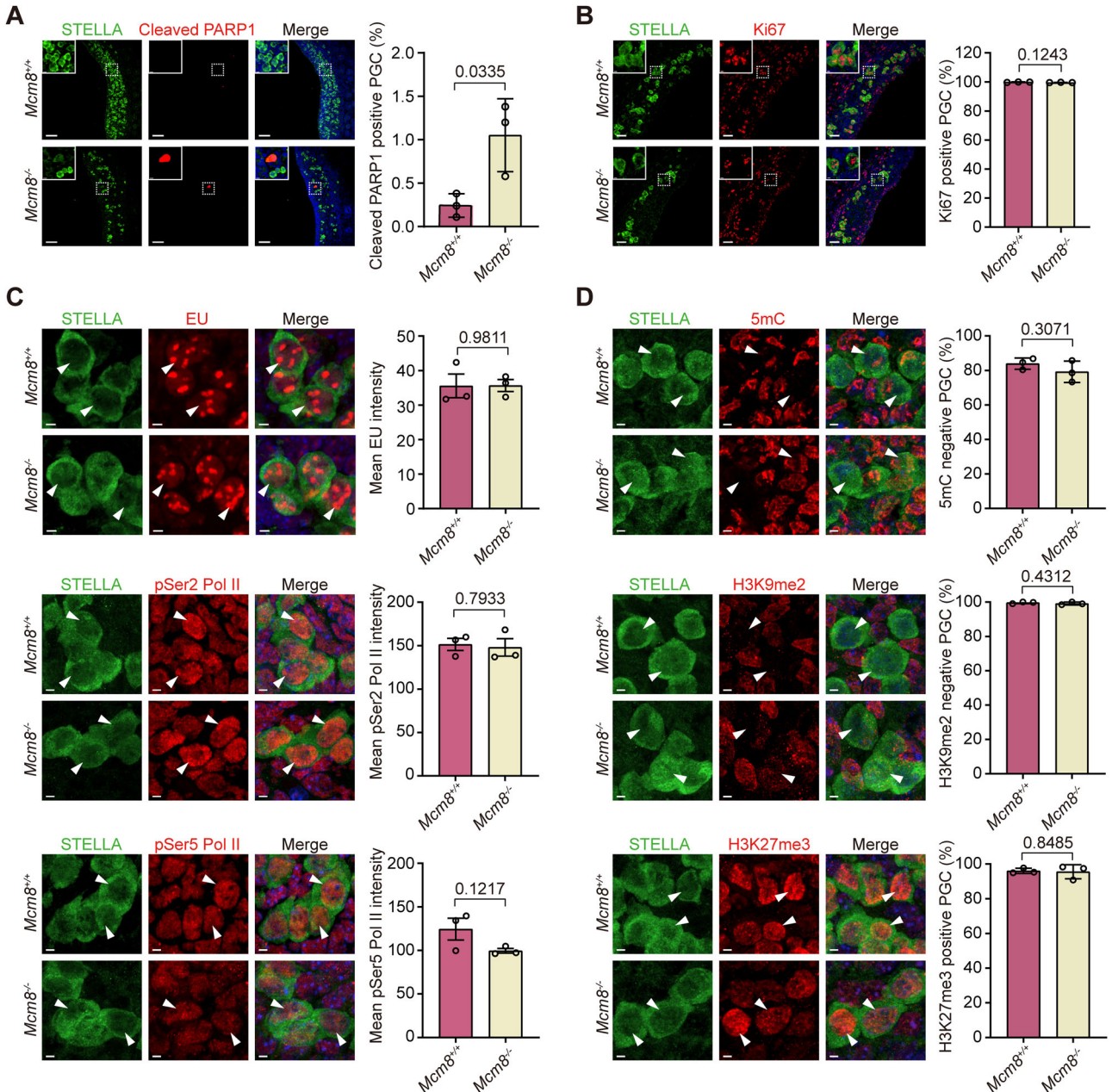

**Figure EV2. Primordial germ cells undergone transcription activation and epigenetic reprogramming in *Mcm8⁻/⁻* embryos.**

(A) Representative images of cleaved PARP1 immunostaining and percentage of the apoptotic PGCs (cleaved PARP1 positive) in E11.5 *Mcm8⁺/⁺* and *Mcm8⁻/⁻* genital ridges. Scale bars: 50 μm. Scale bars: 7 μm in the enlarged images. n = 3/3 embryos. (B) Representative images of Ki67 immunofluorescence staining and percentage of the PGCs actively progressing through the cell cycle (Ki67 positive) in E11.5 *Mcm8⁺/⁺* and *Mcm8⁻/⁻* genital ridges. Scale bars: 20 μm. Scale bars: 3 μm in the enlarged images. n = 3/3 embryos. (C) Representative images of EU incorporation and the phosphorylation of Ser2 and Ser5 within the C-terminal domain (CTD) of RNA polymerase II (Pol II) immunostaining in E11.5 *Mcm8⁺/⁺* and *Mcm8⁻/⁻* genital ridges to observe transcriptional output. Quantification of mean EU, pSer2 Pol II and pSer5 Pol II signal intensity in PGCs. Scale bars: 3 μm. n = 3/3 embryos. (D) Representative images of 5mC, H3K9me2 and H3K27me3 immunostaining in E11.5 *Mcm8⁺/⁺* and *Mcm8⁻/⁻* genital ridges to observe epigenetic modification. Quantification of percentage of the PGCs stained negative for 5mC and H3K9me2, and the PGCs stained positive for H3K27me3. Scale bars: 3 μm. n = 3/3 embryos. Data information: STELLA positivity indicates PGCs. In (A, B, D), data are presented as the mean ± SD, and in (C), data are presented as the mean ± SEM, and the dots indicate individual embryos. Dotted lines indicate positions of the enlarged images (A, B). Arrowheads indicate representative cells (C, D). The statistical significance of the difference was analyzed by unpaired two-tailed Student's *t*-test (A–D), and the *P*-values were shown. Source data are available online for this figure.

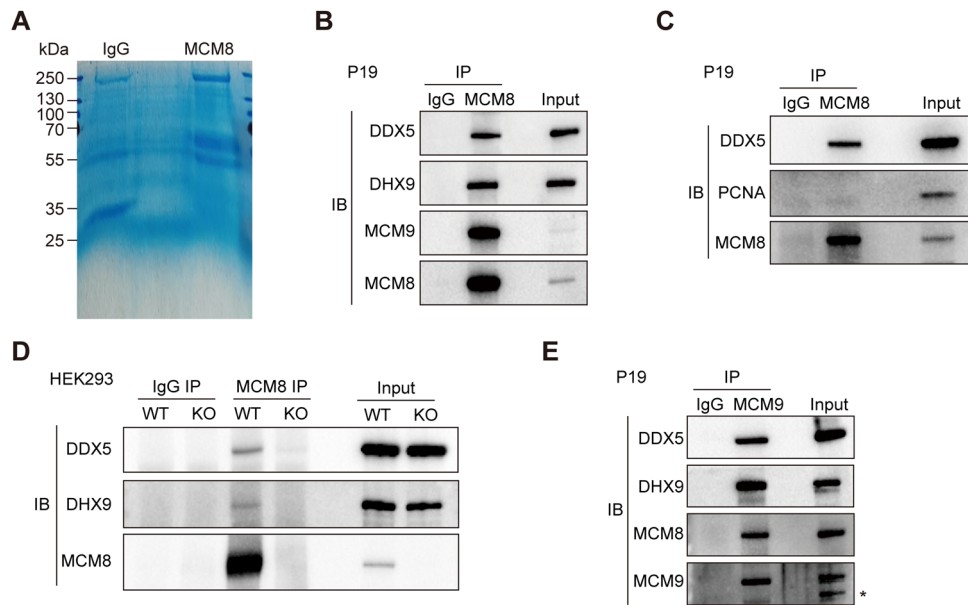

**Figure EV3.  MCM8 and MCM9 specifically interacted with DDX5 and DHX9.**

(A) Representative image of 10% SDS-PAGE gel stained with Coomassie blue. MCM8, MCM8 antibody precipitated proteins in P19 cells; IgG, IgG antibody precipitated proteins in P19 cells. (B) Co-immunoprecipitation of endogenous MCM8 from P19 cell lysates, followed by immunoblot of DDX5, DHX9, MCM9 and MCM8. IgG was used as a negative control. (C) Co-immunoprecipitation of endogenous MCM8 from P19 cell lysates, followed by immunoblot of PCNA, DDX5 and MCM8. IgG was used as a negative control. (D) Co-immunoprecipitation of endogenous MCM8 from WT or KO HEK293 cell lysates, followed by immunoblot of DDX5, DHX9 and MCM8. IgG was used as a negative control. (E) Co-immunoprecipitation of endogenous MCM9 from P19 cell lysates, followed by immunoblot of DDX5, DHX9, MCM8 and MCM9. IgG was used as a negative control. Asterisk (*) indicated a non-specific band detected by anti-MCM9 antibody. Source data are available online for this figure.

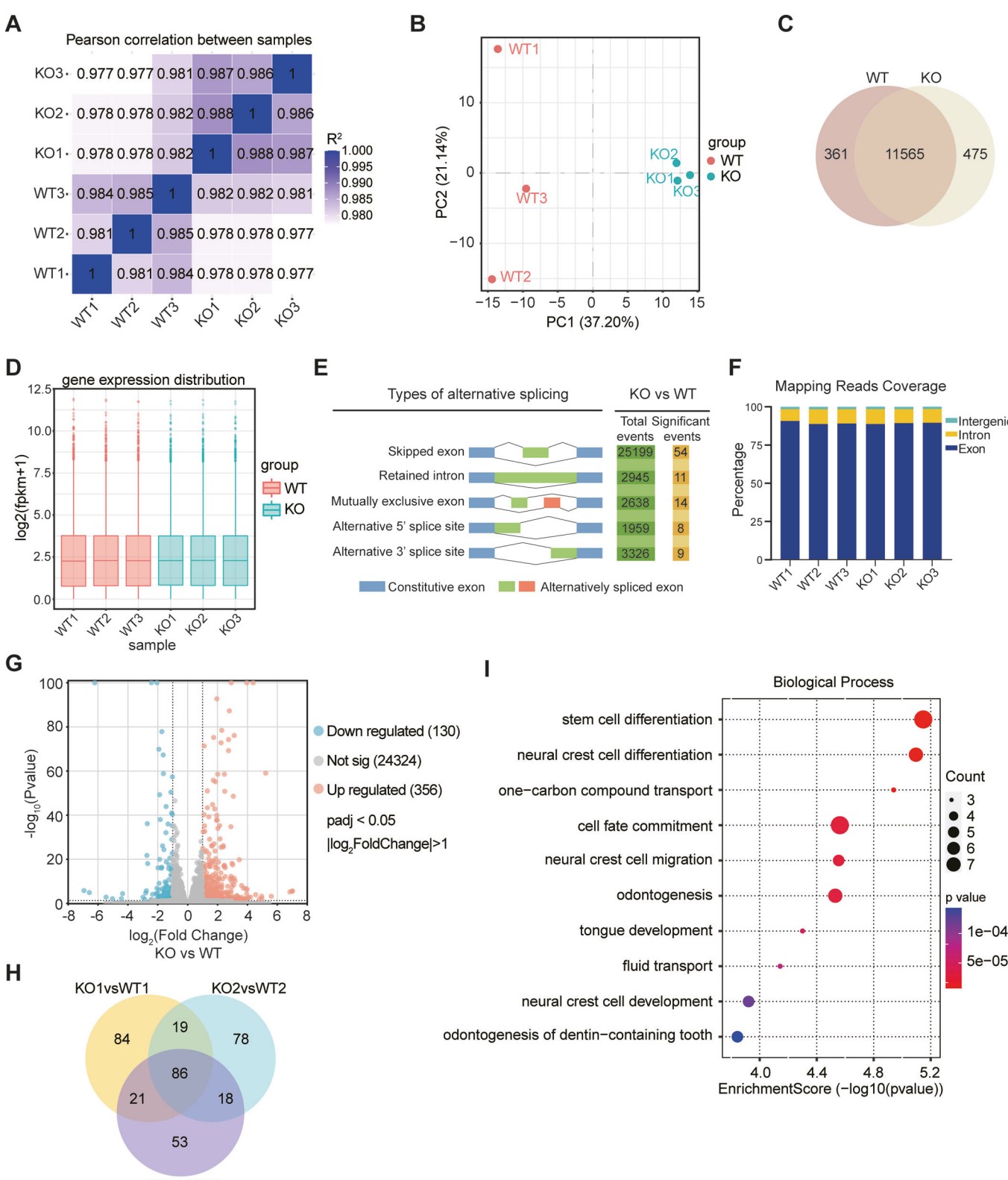

◀    **Figure EV4.   Global transcriptional activity and alternative splicing events in *Mcm8* KO MEFs were not significantly changed.**

(A) Pearson correlation coefficient analysis showing positive correlations between WT and KO groups. $n = 3/3$. (B) Principal component analysis (PCA) of RNA transcriptomes from WT and KO MEFs. (C) Venn diagram of the co-expressed genes between the WT and KO MEFs (fpkm>1). (D) Gene expression distribution of each sample in WT and KO group. (E) Schematic diagram of alternative splicing (AS) types and summary of AS analysis performed in WT and KO group. The numbers of total and different AS events in each category upon MCM8 deletion are indicated. FDR < 0.05 were considered significant. (F) Mapping reads coverage of each sample in WT and KO group. The percentages of reads aligned to exon, intron, and intergenic regions were shown. (G) Volcano plot of differentially expressed genes (DEGs) from RNA-seq in KO group compared to WT group (fpkm>0). |log2FC|>1 and an adjusted *p* value < 0.05 were considered significant. (H) Venn diagram showing 86 genes common to the DEGs from each KO sample compared to WT controls. (I) GO enrichment analysis of biological processes for the 86 genes showing the top 10 enriched terms. Source data are available online for this figure.

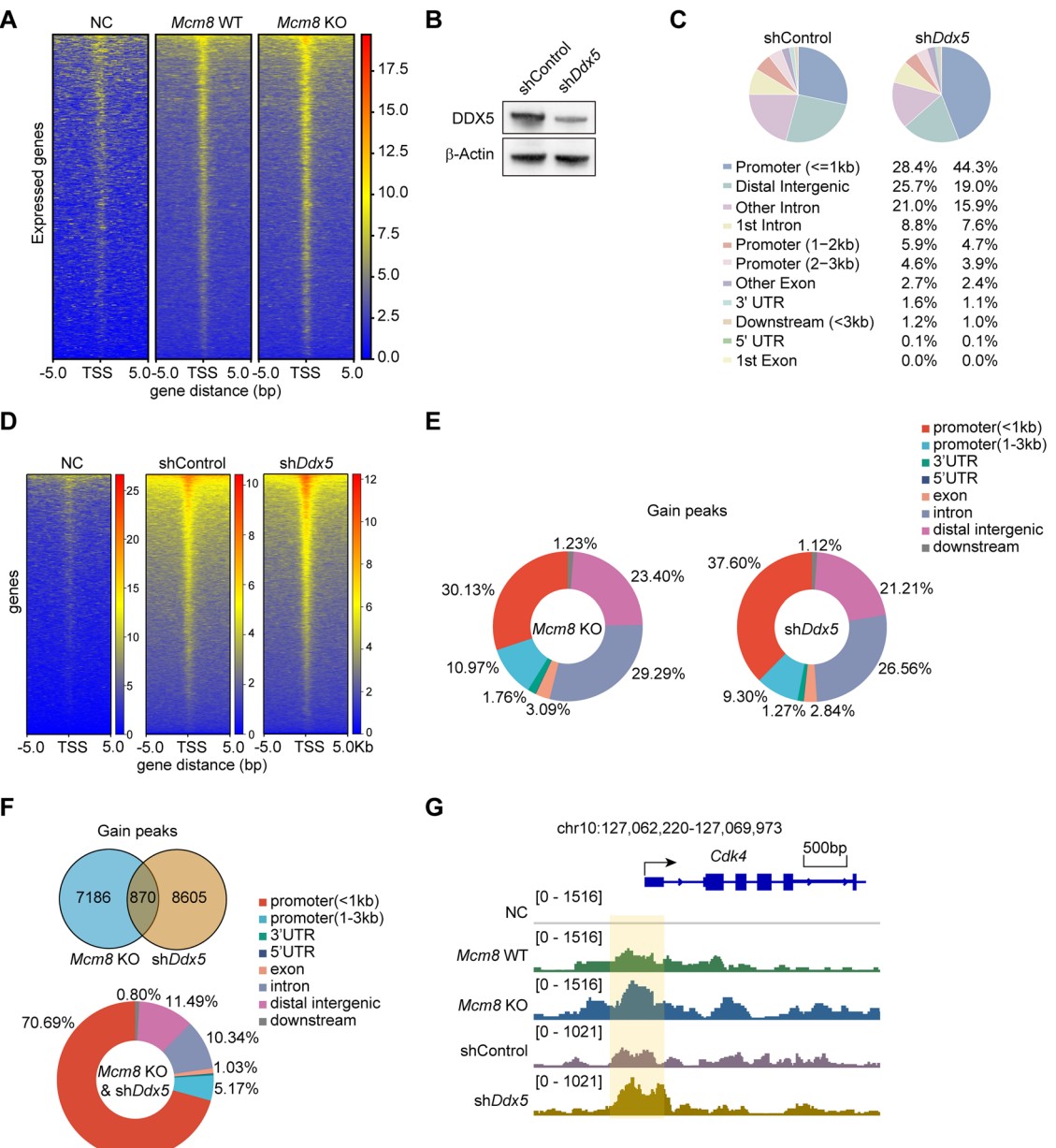

**Figure EV5.  R-loop gain peaks were overlapped in DDX5-knockdown and MCM8-deficient cells.**

(A) Heatmaps showing the distribution of the R-loop signal of all expressed genes (rpkm>1) in WT and KO MEFs. R-loop signal was sorted based on gene expression level obtained from the RNA sequencing analysis of MEFs from high to low. The region −5 kb/+5 kb around the TSS is individually displayed. TSS, transcription start site. (B) Confirmation of DDX5 knockdown in MEFs using sh*Ddx5* adenovirus by western blot. β-Actin was used as the loading control. (C) The genomic distribution of R-loop CUT&Tag peaks in shControl MEFs and sh*Ddx5* MEFs. UTR, untranslated region. (D) Heatmap plots of genes with R-loop signals across the 5 kb window around TSS in the NC, shControl and sh*Ddx5* group. NC, negative control that represents a group without primary antibody in the CUT&Tag assay. (E) The genomic distribution of the elevated R-loop CUT&Tag peaks (termed gain peaks) in *Mcm8* KO MEFs relative to WT (left) and sh*Ddx5* group relative to shControl (right). (F) Top: Venn diagrams showing the overlaps among shared peaks with gain in R-loop signal upon *Mcm8* KO and *Ddx5* knockdown condition. Bottom: The genomic distribution of the overlapped gain peaks. (G) Snapshots of R-loop signals in the representative gene *Cdk4* by genome browser tracks in MEFs. NC (gray), *Mcm8* WT (blue), *Mcm8* KO (green), shControl (purple) and sh*Ddx5* (brown) CUT&Tag data are shown. Source data are available online for this figure.

