## [Peer Review File · The EMBO Journal]

MCM8 interacts with DDX5 to promote R-loop resolution

Canxin Wen, Lili Cao, Shuhan Wang, Weiwei Xu, Yongze Yu, Simin Zhao, Fan Yang, Zi-Jiang Chen, Shidou Zhao, Yajuan Yang, and Yingying Qin

Corresponding author(s): Yingying Qin (qinyingying@sdu.edu.cn), Shidou Zhao (shidouzhao@sdu.edu.cn), Yajuan Yang (YangYJ0204@sdu.edu.cn)

Review Timeline:

Submission Date:	12th Aug 23
Editorial Decision:	2nd Oct 23
Revision Received:	31st Jan 24
Editorial Decision:	1st Mar 24
Revision Received:	17th May 24
Accepted:	21st May 24

Editor: Hartmut Vodermaier

Transaction Report:

EMBOJ-2023-115128 Pre-Decision Discussion

Dear Dr. Qin,

Thank you for submitting your manuscript on MCM8 and R-loop resolution to The EMBO Journal. We have now received comments from three expert referees, copied below for your information. As you will see, all reviewers consider this proposed MCM8 role in PGC maintenance potentially interesting. However, especially referees 1 and 3 also raise a considerable number of major concerns and question whether key conclusions of the study are decisively supported.

In order to find out whether these crucial issues might be adequately addressed through further work and satisfactory clarifications, I would like to give you an opportunity to consider the referee reports, and to provide a tentative response letter answering in detail to each of the referees' points. Based on this response, which we may share and discuss with the referees, we would then decide whether it would be warranted to invite a formal revision of the study for EMBO Journal publication.

I would appreciate if you could get back to us with your revision plan by early next.

Yours sincerely,

Hartmut Vodermaier

Referee #1 (Report for Author)

The manuscript by Wen et al. explored the function of Mcm8, a key gene known to function in homologous recombination repair and implicated in premature ovarian insufficiency (POI), during primordial germ cell (PGC) development in mice. The authors showed that an Mcm8 knockout (KO) leads to substantial loss of PGCs and infertility in both males and females. The authors provided multiple lines of evidence showing that MCM8 interacts with R-loop regulators such as DDX5 and DHX9 to promote the resolution of R-loops, in particular, those around gene promoters, during rapid proliferation of PGCs, and the failure in such processes leads to DNA damage accumulation and impaired PGC proliferation. Furthermore, the authors showed that Mcm8 mutants causative for POI display weaker interactions with DDX5 and leads to increased R-loop formation. This is a potentially interesting manuscript that provides an insight into the role of MCM8 and the significance of R-loop resolution during PGC development, which has a relevance in better understanding the etiology of POI.

Several concerns regarding the data and presentations in the current version are as follows:

Major concerns:

1. In figures showing immunofluorescence (IF) staining data (Fig 1A, B, 2A, 2C, 2E, EV4A, etc), the authors should provide more magnified, clear images so that the readers can evaluate the authors' statements more precisely.
2. In Page 6, the authors stated that "Compared to PGCs in WT littermates and single mutants of Mcm8 KO or ubiquitination-defective Fancd2, significantly fewer PGCs were detected in the double mutants (1467.0 {plus minus} 307.4 vs. 436.0 {plus minus} 130.6 vs. 300.6 {plus minus} 117.8 vs. 43.2 {plus minus} 6.4, $P < 0.001$), suggesting that MCM8 has functions in PGC development that are independent of the FA pathway (Fig. EV4B)." According to Hill and Crossan, 2019, the DNA damage response (DDR) pathway is important in PGCs, especially in the context of the FA pathway. Furthermore, the FA pathway is also crucial for R-loop resolution (García-Rubio et al., 2015). Given these findings, which DDR pathway do the authors believe the results related to R-loop accumulation in Mcm8 KO are more closely associated with?
3. In Page 7, the authors stated that "MCM8 deficiency resulted in a higher proportion of apoptosis when compared to the WT group (1.05 {plus minus} 0.42% vs. 0.24 {plus minus} 0.14%, $P < 0.05$), but the number of apoptotic PGCs was less than 10 per embryo (Fig. 2A and B)." Why then does the number of PGCs decrease so dramatically in Mcm8 KO mice?
4. In Page 8, the authors stated that "We removed 25 ribosomal proteins that were always found among the nonspecific binding proteins in the IP-MS assay and analyzed the remaining 84 binding proteins. We noted that the MCM8 interactors were mainly involved in RNA splicing processes, as indicated by Gene Ontology (GO) analysis (Fig. 3A)." The GO term "RNA splicing processes" is a also broad term, and many proteins related to this process are expected to be abundant in the nucleus, similar to "ribosomal proteins." This reviewer is concerned that this observation may also be attributed to noise. Therefore, conducting a comparative analysis for MCM8-IP in cells less abundant for MCM8 as a negative control may further support the authors' claim.
5. Fig 3B: Related to the above point, this analysis should be presented alongside an appropriate negative control to confirm that the "MCM8 interactome" significantly overlaps with "Human and mouse R-loop regulators."
6. Fig 3F, 5A, 7D: I do not think it really possible to make quantitative comparison of the IF signals between different specimens with the method employed by the authors, as the staining levels and intensities can easily be changed and very difficult to control between different samples. To overcome this point, for example, the authors should place the cells with two different genotypes on the same slide and then compare the signals. Why were S9.6 signals so

strong only in single puncta in the nuclei?

7. Fig 4B and Fig 4C: Judging from the pattern observed in the "non-Antibody" track in Fig 4B, the signal-to-noise (S/N) ratio of the IP tracks may not be optimal. Therefore, in this case, it is crucial to mention in detail in the "CUT&Tag assay" section that peak calling was performed using the "non-Antibody" track as the input track in MACS2.

8. Fig 4C-E: This reviewer wonders why the peaks in the WT condition are not entirely encompassed by those in the KO condition. Given the authors' primary claim that the KO condition leads to more frequent R-loops, one would expect the KO to exhibit additional peaks in addition to those found in the WT. However, Fig 4C shows many unique peaks in the WT. What are the properties of these peaks specific to WT cells?

9. In Page 9: "actively proliferating PGCs (EGFP positive) at E12.5..." The authors should explain the marker they used.

10. In Page 9: ".....as indicated by the normal transcriptional upregulation of KO PGCs when compared to WT PGCs (Fig. EV5A and B)." This reviewer does not think that to draw this conclusion is not really possible only with these ambiguous IF images. The authors should perform some different, more quantitative experiment (e.g., RNA-seq experiment with appropriate spike-ins) to make this conclusion.

11. In Page 10, the authors stated that "Two genes, Chchd7 and Armc10 are shown as examples (Fig. 4F). The biological processes of these two genes with increased R-loop signals were enriched in the positive regulation of developmental growth according to the GO analysis (Fig. 4G)." Here, the meaning of enrichment in developmental growth appears unclear to this reviewer, at least in its current context. Is this term specifically related to the mitotic proliferation of PGCs? It would be beneficial if the authors could clarify the significance of this term. Additionally, the rationale for selecting these two genes is not clearly explained.

12. Fig 5A and D: Is GFP-RNH1 fusion protein? If so, why does the majority of the proteins locate in the cytoplasm?

13. In Page 11, the authors stated that "We found that 16.9% (1657/9799) of the MCM8 binding sites co-occurred with R-loop signals (Fig. 6C), with the majority (81.3%, 1347/1657) being located in the promoter regions." From this reviewer's perspective, the observed percentage of 16.9% appears to be quite low to suggest a direct link between MCM8 and R-loop resolution. Could it be that MCM8 is only responsible for a specific subset of R-loops? It would be valuable for the authors to discuss the possible reasons behind this observation or provide additional evidence to support their findings in the text.

14. In Page 11, the authors stated that "R-loop levels are positively associated with

transcriptional activity (Wahba et al, 2016)." Based on the authors' data, is it also observed that genes with higher transcriptional activities exhibit more pronounced differences between the KO and WT conditions? If so, this could serve as further confirmation of the data quality.

15. Fig 6H and I: As in other IF figure panels, the data are hardly visible in H and the very minor difference shown in I is not really convincing.

16. In Page 12 and Fig 7, it would be better for the authors to define what p. R309* and p.S492* stand for, for the clarity of the readers.

17. What might be the phenotypes in germ cells in Ddx5 or Dhx9 knockouts? This information is useful to include for discussion to understand a broader picture of their relevance in germ cells.

18. In Page 14, the authors stated that "...actively dividing PGCs were susceptible to MCM8 deficiency; therefore, it was hypothesized that mitotic germ cells are more dependent on MCM8 to resolve specific replication stress, thus safeguarding genome stability. Recently, we reported that rapidly proliferating PGCs are faced with constitutive replication stress, including high levels of R-loops and frequent RF stalling (Yang et al., 2022)." There are many actively dividing cell types during development and therefore, further discussion may be useful why the authors think that PGCs are specifically susceptible to MCM8 loss and R-loop mis-resolution.

Minor points:

1. In Page 7, the authors stated that "Because PGCs are sensitive to DNA damage, we reasoned that PGCs had an increased requirement for MCM8 in genome stability maintenance under physiological conditions and that unrepaired DNA damage would result in apoptosis or cell cycle arrest in MCM8-deficient PGCs." In this sentence, the authors should consider including a citation to support the statement that "PGCs are sensitive to DNA damage."

2. In Page 8, the authors stated that "We removed 25 ribosomal proteins that were always found among the nonspecific binding proteins in the IP-MS assay and analyzed the remaining 84 binding proteins." Is there any evidence or citation to validate the exclusion of ribosomal proteins in this IP-MS analysis?

3. In Page 10, "... and KO embryos (Fig. 4A)" should be "KO MEFs."

4. In Page 13, the authors stated that "Furthermore, overexpression of WT MCM8 in KO HEK293 cells showed a drastic decrease in R-loop levels compared to KO cells." Please add a citation to the relevant figure.

5. In Page 17, the authors stated that "In addition, we also showed that p.E341K (rs16991615), which had normal interaction with DDX5, did not show equally efficient R-loop removal as WT

MCM8." Please add a citation to the relevant figure.

Reference

1. M. L. García-Rubio et al., The Fanconi Anemia Pathway Protects Genome Integrity from R-loops. *PLOS Genetics* 11, e1005674 (2015).
2. R. J. Hill, G. P. Crossan, DNA cross-link repair safeguards genomic stability during premeiotic germ cell development. *Nature Genetics* 51, 1283-1294 (2019).

Referee #2 (Report for Author)

MCM8 and MCM9 form a hexameric helicase, which functions in homologous recombination (HR) in mitosis and meiosis. The manuscript by Wen et al. describes a novel role of MCM8 in R-loop resolution in primordial germ cells (PGCs). The authors initially observed a significant reduction of PGCs in MCM8-KO embryos (Fig. 1). PGCs lacking MCM8 exhibited proliferation defects during the S phase (Fig. 2). Mass spectrometry analysis revealed that MCM8 interacted with numerous RNA processing proteins (Fig. 3). The authors confirmed the interactions of MCM8 with DDX5 and DHX9, both of which are involved in R-loop resolution. PGCs and MEFs derived from MCM8-KO mice accumulated R-loops, as detected by the S9.6 antibody (Fig. 3F, G and Fig. 5A, B). This R-loop accumulation was further validated using CUT&TAG (Fig. 4). MCM8-KO MEFs displayed reduced fork speed due to the accumulation of R-loops (Fig. 5). MCM8 was identified within genes related to mitotic progression and was found to bind to R-loops (Fig. 6). Notably, in MCM8-KO MEFs, the association of DDX5 and DHX9 with R-loops was reduced. Finally, the authors determined that DDX5 interacted with the AAA+ core domain of MCM8. Moreover, the p.R309* and p.S492* mutations found in patients were unable to suppress R-loop accumulation in HEK293 MCM8-KO cells (Fig. 7).

The presented findings are interesting and potentially explain the reason why PGCs before entering meiosis were lost in MCM8-KO mice. However, one point should be clarified before consideration for publication.

It has been reported that MCM8 forms both MCM8/9 dimer and hexamer (Gambus and Blow, *Cell Cycle*, 2013; McKinzev et al., *NAR*, 2023; Weng et al. *eLife*, 2023; Acharya et al. *BioRxiv*, 2023). Furthermore, studies with MCM9-KO mice have demonstrated a decrease in PGCs (Hartford et al. *PNAS*, 2011; Luo and Schimenti, *Genesis*, 2015). However, the authors favour the idea that MCM8 functions in R-loop resolution without MCM9, as shown in Fig. 7F and discussed in Discussion. The authors should provide clarification regarding whether MCM9 is dispensable for R-loop resolution and whether MCM8 operates autonomously from MCM9.

Minor points

- Reconsider the figure order. Fig. 5 A, B should directly follow Fig. 3.
- Show control HEK283 WT cells in Figure 7D, E.

- HROB is required for MCM8/9 helicase activity and works with MCM8/9 on in ICL repair and HR (Hustedt et al., G&D, 2019; Huang et al. Nat. Commun., 2020). This point should be mentioned in Introduction.

Referee #3 (Report for Author)

This manuscript reports phenotypes associated with knock out (KO) of the MCM8 DNA helicase in mouse primordial germ cells (PGCs) showing reduction of the pool of Primordial Germ Cells (PGCs) and generation of DNA damage, very likely DNA double strand breaks, resulting in infertility, confirming the original report (Lutzman et al., 2012). Authors also analyzed the global phosphorylation level of RNAPol II Ser2 and 5, as well as markers of epigenetic reprogramming in MCM8 KO, such as 5mC, H3K9me2 and H3K27me3, and found no significant differences, suggesting that MCM8 KO did not alter global transcription nor epigenetic reprogramming in PGCs. They next obtained the MCM8 protein interactome by MCM8 immunoprecipitation in P19 teratocarcinoma mouse cells and identified mostly proteins involved in RNA splicing processes. Amongst these proteins are proteins implicated in R-loop resolution, a DNA:RNA hybrid structure with a displaced single-stranded DNA that can be generated by conflicts between transcription and replication machinery, as well as a result of reduced mRNA processing (e.g.: splicing defects that slow down the transcription process). Because it was previously reported that PGCs display a high R-loops level, the authors explored whether compromised interaction between MCM8 and R-loops regulators may be responsible for the observed PGCs reduction and DNA damage upon MCM8 KO. For that, authors first confirmed physical interaction between MCM8 and two previously characterized R-loops regulators, the RNA helicases Ddx5 and Dhx9 in P19 as well as in PGCs. Then, they analyzed R-loop formation in either wild-type or MCM8 KO PGCs and show increased R-loops levels in MCM8 KO PGCs and mouse embryonic fibroblasts (MEFs). They further analyzed R-loops distribution genome-wide in MEFs and report a similar distribution between wild-type and MCM8 KO cells, as well as the appearance of additional R-loops compared to wild-type cells. Further, they asked whether R-loops accumulation is responsible for replication stress generated in MCM8 KO cells and for that authors measured replication fork progression in wild-type or MCM8 KO MEFs and report reduced replication fork speed which is somehow restored upon treatment with RnaseH1, an enzyme that degrades the RNA moiety of the R-loop. Finally, authors analyzed Ddx5 and Dhx9 retention at R-loops upon immunoprecipitation with the S9.6 antibody and show that their recovery is reduced in cells MCM8 KO. They conclude that MCM8 contributes to target Ddx5 and Dhx9 to R-loops and as such preserve genome integrity and PGCs proliferation.

Overall the study is well executed and reports interesting findings, in part confirming previous observation on the role of the MCM8 DNA helicase in DNA replication and germ cells metabolism. The observed interaction between MCM8 and Ddx5/Dhx9 is novel but somehow surprising. However, I found the claim that MCM8 KO increases the frequency of R-loops in PGCs needs to be strengthened. Also, I do not understand why the authors have not explored a potential role for MCM8 in regulating splicing, as strongly suggested by the MCM8 interactome,

and they instead focused on a possible role of MCM8 on R-loops metabolism. Mutations that affect splicing indirectly generate replication stress and consequent R-loop increase, as originally shown (Paulsen et al., 2009 Mol Cell; PMID: 19647519; Gomez-Gonzalez et al., 2009 Mol Cell Biol; PMID: 19651896). Further, from data shown in Fig. 3B, the Numa1 protein appears to be the most abundant MCM8 interactor. Numa1 is a protein involved in the correct orientation of the mitotic spindle but it has also recently shown to localize to gene promoter regions (Ray et al., Nature 2022; PMID: 36171374). Numa1 depletion generates DNA damage and genomic instability, phenotypes similar to those observed in MCM8 KO cells, and consistent with the increase in the fraction of cells in the G2 phase of the cell cycle shown in Fig. 2F. Hence, I do not see the logic of focusing on Ddx5 and especially Dhx9, this latter being not the most abundant MCM8 interactor compared to at least two other Ddxs involved in R-loops resolution such as Ddx21 and Ddx41, unless authors have reasons to believe that these latter do not contribute to the genomic instability in PGCs upon MCM8 KO. Finally, although the authors claim a function for MCM8 for R-loops metabolism in PGCs, the critical data have been mostly obtained using P19, MEFs and HEK293T cells. Here below are comments that may help the authors to improve the manuscript.

1. Fig. 2A-B, the difference in the number of cells showing cleaved PARP-1 staining between wild-type and MCM8 KO is very weak. The quantification shown in panel B does not match the data shown in panel A. Same applies to Fig. 2E-F for Cyclin B1 and EdU. I do not understand how authors can identify G1 and G2 cells using EdU (S-phase) and Cyclin B1 (mitosis) staining. From these data one cannot say that cells are arrested in the cell cycle. If one believes these data, what appears is that there is an increase in the number of G2 cells, but this does not mean that cells are arrested. Checking activation of the DNA damage response (ATM and/or ATR, Chk1/Chk2) would have been more convincing.
2. In Fig. 2G-H authors show that MCM8 KO PGCs display a strong increase in the number of 53BP1 foci compared to wild-type PGCs and conclude that DNA damage accumulation leads to decreased PGCs proliferation. However, authors did not provide additional data to prove this statement. Should this be the case, then proliferation should be rescued by inhibiting the DNA damage response (probably the ATM pathway, not shown in the manuscript).
3. Fig. 3, it is indeed curious that the MCM8 interactome showed a great enrichment of RNA splicing factors. Why the authors have not explored a potential role of MCM8 in splicing? What about the best hit, Numa1? Did the authors identified MCM9 in the proteome, which is a stable MCM8 partner? I could not find the list of the proteins identified by mass spectrometry. A western blot and silver stain of the immunoprecipitated material must be shown.
4. Fig. 3C-E, 6F, 7B-C, do the western blot signals belong to the same gel? This is an important point in order to compare the relative signal intensity. In the immunoprecipitation experiments, can the authors show a protein that does not interact with MCM8? Can they immunoprecipitated MCM9?
5. Fig. 3F-G, it is important to include a treatment with RNaseH1 to show the specificity of the S9.6 signal. Also, the use of RNaseH1-dead mutant has been now shown to be a more reliable tool to detect R-loops than the S9.6 antibody by immunofluorescence (Crossley et al., JCB 2021;

PMID: 34232287).

6. It would have been nice to have a DDX5 knock out (or downregulation) control aside both immunofluorescence and Chip data to compare with. Ideally, one wished to see how Ddx5 or Dhx9 affects R-loops distribution genome-wide and compare it with that observed in MCM8 KO cells.

7. In Fig. 5C authors analyze replication dynamics by DNA fiber stretching and conclude that replication speed is slower in MCM8 KO cells, in line with previous reports demonstrating a function of MCM8 in contributing to replication fork progression (Maiorano et al., Cell 2005; Gambus et al., Cell Cycle 2013; Griffin et al., 2022). However, according to these data, it appears that MCM8 depletion reduced the length of both IdU and CldU tracks, suggesting an effect on both the initiation and elongation steps of DNA synthesis. However, I am not convinced that this is due to R-loops accumulation, as claimed by the authors. The statistical test here and elsewhere is inappropriate (see other points below). Significance must be recalculated using a Man-Whitney test. Further, images shown in panel D are not representative of the quantification shown in panel E. Concerning the possible increase in R-loops upon MCM8 KO, this could be due to reduce DNA synthesis rate generating transcription-replication conflicts. Hence it is expected that inhibition of transcription should reduce R-loops formation, thus rescuing the effect of MCM8 KO. Fig. 5D-E, the reduction of 53BP1 foci in MCM8 KO cells treated with RNaseH1 is weak and quantification does not match the data.

8. Fig 6, a western blot showing the expression of Flag-MCM8 must be shown. Panel E, a control including RnaseH1 treatment and ChiP with a non-specific antibody (Flag alone as a control for Flag-MCM8 Chip and non-specific for the S9.6 Chip) must be included to ascertain the specificity of the immunoprecipitated material. Panel F, it is critical here to show that equal amounts of IgGs were recovered in all lanes, since based on this experiment authors conclude that Ddx5 and Dhx9 retention is reduced on R-loops upon MCM8 KO.

9. Fig. 7, authors complement R-loops generated by MCM8 KO in HEK293T cells by expressing either wild-type or divers MCM8 mutants. Here again, the images provided in Fig. 7C do not match the quantification shown in panel E. The differences in S9.6 signals observed in these images are very weak.

10. Fig. EV6, a quantification of the Ddx5 and Dhx9 immunofluorescence signals obtained must be shown.

Other points

1. Page numbers are missing

2. Introduction, second page, MCM2-7 protein are not degraded in S-phase, they are removed from chromatin by ongoing DNA synthesis (Todorov et al., J. Cell Biol. 1995). Next page, Ddx5 and Dhx9 are not "the" R-loops removing factors but are R-loops-removing factors. A general introduction to factors known so far in R-loops resolution (including other Ddxs helicases) is missing.

3. Fig. 3F-G, it is unclear what the white arrows indicate (whether cytoplasmic R-loops or cells displaying nuclear R-loops); this has not been described in the figure legend.

4. Fig. 7F, what is the evidence that Ddx5 and Dhx9 function as dimers (or multimers) in R-loops resolution? It is entirely possible that resolution of a fraction of R-loops depends upon MCM8-

Ddx5 and/or Dhx9 interaction. It is for this reason that it would have been nice to have a picture of R-loops distribution in Ddx or Dhx9 knock down cells.

5. A Mann-Whitney U test is more appropriate to determine significance in several experiments (Fig. 3G; 5B-C; 7E).

Dear Dr. Vodermaier,

Thank you for your consideration of our manuscript 'EMBOJ-2023-115128'. We appreciate the referees for their valuable comments on our work. The comments of the referees have been responded to point by point. We would appreciate it if you could give us an opportunity to revise the manuscript through our tentative response.

Below you will find our point-by-point responses and revision plans to the referees' comments.

Referee #1

Major concerns:

1. In figures showing immunofluorescence (IF) staining data (Fig 1A, B, 2A, 2C, 2E, EV4A, etc), the authors should provide more magnified, clear images so that the readers can evaluate the authors' statements more precisely.

Response: Thank you for your suggestion. We are sorry for the compressed IF images with poor resolution in the initially submitted manuscript. All of our original IF images are clear and of high resolution. We will provide more magnified and clear images throughout the manuscript as required.

2. In Page 6, the authors stated that "Compared to PGCs in WT littermates and single mutants of Mcm8 KO or ubiquitination-defective Fancd2, significantly fewer PGCs were detected in the double mutants (1467.0 {plus minus} 307.4 vs. 436.0 {plus minus} 130.6 vs. 300.6 {plus minus} 117.8 vs. 43.2 {plus minus} 6.4, $P < 0.001$), suggesting that MCM8 has functions in PGC development that are independent of the FA pathway (Fig. EV4B)." According to Hill and Crossan, 2019, the DNA damage response (DDR) pathway is important in PGCs, especially in the context of the FA pathway. Furthermore, the FA pathway is also crucial for R-loop resolution (García-Rubio et al., 2015). Given these findings, which DDR pathway do the authors believe the results related to R-loop accumulation in Mcm8 KO are more closely associated with?

Response: Thank you for your comments. We have reviewed the DNA repair proteins involved in suppression and resolution of R-loops. Besides the FA pathway, a number of DNA repair proteins participate in the regulation of R-loop homeostasis, including DNA-damage checkpoint proteins such as ATR and ATM, and homologous recombination (HR) proteins such as BRCA1, BRCA2 and RAD51, etc. Generally, they suppress or remove R-loops by promoting the recruitment of the RNA-DNA helicases. For instance, while activation of ATR and ATM promotes the recruitment of the RNA-DNA helicase SETX or DDX19 to unwind R-loops (*Yuce and West. Mol*

Cell Biol. 2013. PubMed: 23149945; Hodroj, et al. EMBO J. 2017. PubMed: 28314779), BRCA2 recruits RNA-DNA helicase DDX5 to resolve R-loops (Sessa G, et al. EMBO J. 2021. PMID: 33634895). In this study, we first reveal that MCM8 also recruits DDX5 to resolve R-loops and it regulates PGC development that is independent of the FA pathway. Because MCM8 is well known for its involvement in homologous recombination (HR), it is assumed that its role in R-loop resolution may be associated with the HR pathway. However, it needs to be determined in the future study.

3. In Page 7, the authors stated that "MCM8 deficiency resulted in a higher proportion of apoptosis when compared to the WT group (1.05 {plus minus} 0.42% vs. 0.24 {plus minus} 0.14%, $P < 0.05$), but the number of apoptotic PGCs was less than 10 per embryo (Fig. 2A and B)." Why then does the number of PGCs decrease so dramatically in *Mcm8* KO mice?

Response: Both excessive apoptosis and proliferation defects can result in PGC loss. However, during a limited time window between E9.5 and E13.5 in mice, PGCs undergo very rapid proliferation to expand the germ cell pool. Although the apoptotic *Mcm8*^{-/-} PGCs were few (less than 10 per embryo) at E11.5, PGCs at S-phase were decreased and PGCs at G2-phase were significantly increased in *Mcm8*^{-/-} embryos, indicating that impaired cell expansion due to proliferation defects was the leading cause of a dramatic PGC loss in *Mcm8*^{-/-} embryos.

4. In Page 8, the authors stated that "We removed 25 ribosomal proteins that were always found among the nonspecific binding proteins in the IP-MS assay and analyzed the remaining 84 binding proteins. We noted that the MCM8 interactors were mainly involved in RNA splicing processes, as indicated by Gene Ontology (GO) analysis (Fig. 3A)." The GO term "RNA splicing processes" is also a broad term, and many proteins related to this process are expected to be abundant in the nucleus, similar to "ribosomal proteins." This reviewer is concerned that this observation may also be attributed to noise. Therefore, conducting a comparative analysis for MCM8-IP in cells less abundant for MCM8 as a negative control may further support the authors' claim.

Response: Thank you for your suggestion. It was reported that ribosomal proteins including ribosomal protein large/small subunits (RPL/RPS) were among the most common contaminating proteins in the antibody-based affinity purification–mass spectrometry experiments (Mellacheruvu D, et al. *Nat Methods. 2013. PMID: 23921808; Mohammed H, et al. Nat Protoc. 2016. PMID: 26797456*). Accordingly, we have excluded RPL/RPS proteins from our IP-MS data in which 13.7% (14/102) and 19.6% (37/189) purified binders were removed from the IgG interactome and

MCM8 interactome, respectively. According to your suggestion, we will include a comparative analysis for MCM8-IP in MCM8 KO/knockdown cells as a negative control to further support our claim.

5. Fig 3B: Related to the above point, this analysis should be presented alongside an appropriate negative control to confirm that the "MCM8 interactome" significantly overlaps with "Human and mouse R-loop regulators."

Response: Thank you for your suggestion. We had tried to downregulate MCM8 in P19 cells by utilizing siRNA, shRNA adenovirus and antisense oligonucleotides (LNA-GapmeRs), but they all failed to downregulate MCM8 efficiently. Alternatively, we will provide MCM8 interactome from MCM8 WT and KO MEF/HEK293 cells (negative control) to further confirm that the MCM8 interactome significantly overlaps with "Human and mouse R-loop regulators".

6. Fig 3F, 5A, 7D: I do not think it really possible to make quantitative comparison of the IF signals between different specimens with the method employed by the authors, as the staining levels and intensities can easily be changed and very difficult to control between different samples. To overcome this point, for example, the authors should place the cells with two different genotypes on the same slide and then compare the signals. Why were S9.6 signals so strong only in single puncta in the nuclei?

Response: Thank you for your suggestion. At present, it is a common approach to evaluate the R-loop levels between different specimens by quantitative comparison of the IF signals of the S9.6 antibody, which is known to recognize DNA-RNA hybrids (*Bhatia V, et al. Nature. 2014. PMID: 24896180; Wiedemann EM, et al. Cell Rep. 2016. PMID: 27974207*). To minimize the variability of staining levels and intensities between samples on different slides, we rigorously controlled the experimental conditions, including pairing WT and KO embryos from the same pregnant mice, seeding WT and KO cells simultaneously, fixing samples with the same duration, and using the same exposure intensity and time when capturing the images. Therefore, increased IF signals of S9.6 in our study could provide supportive evidence for R-loop accumulation in MCM8-deficient cells. To give a more convincing conclusion, we will use dot-blot with the S9.6 antibody to quantify R-loop levels.

Besides common R-loops in the genome, the S9.6 antibody also recognizes DNA-RNA hybrids within the mitochondria and the nucleoli. The single puncta strong signals in the nuclei were generated by the abundant rRNA-derived DNA-RNA hybrids in the nucleoli (*Shen W, et al. Nucleic Acids Res. 2017. PMID: 28977560*). In this study, we also detected the nucleolus with an anti-Fibrillarin antibody (Fig. 5A), which was consistent with the previous studies (*Mersaoui SY, et al. EMBO J. 2019. PMID: 31267554; Gong D, et al. Stem Cell Reports. 2023. PMID: 36931280*).

7. Fig 4B and Fig 4C: Judging from the pattern observed in the "non-Antibody" track in Fig 4B, the signal-to-noise (S/N) ratio of the IP tracks may not be optimal. Therefore, in this case, it is crucial to mention in detail in the "CUT&Tag assay" section that peak calling was performed using the "non-Antibody" track as the input track in MACS2.

Response: Thank you for your comment. Compared to ChIP-seq datasets, CUT&Tag profiles have extremely low background noise levels (*Kaya-Okur HS, et al. Nat Commun. 2019. PMID: 31036827*), which is consistent with the CUT&Tag profiles of the S9.6 antibody in our samples (Fig. 4F). It has been reported that for large genomes the reads in the no-antibody negative control are too sparsely distributed to be useful for data analysis (*Kaya-Okur HS, et al. Nat Protoc. 2020. PMID: 32913232*). Consistently, there was no significant enrichment in our "no-antibody" group. Therefore, we performed peak calling without using the "no-antibody" track as the input track. But, it was inappropriate to show the genomic distribution of signals in "no-antibody" group in Fig. 4B in the previous manuscript. Accordingly, we will amend our image and add detailed information in the "CUT&Tag assay" section.

8. Fig 4C-E: This reviewer wonders why the peaks in the WT condition are not entirely encompassed by those in the KO condition. Given the authors' primary claim that the KO condition leads to more frequent R-loops, one would expect the KO to exhibit additional peaks in addition to those found in the WT. However, Fig 4C shows many unique peaks in the WT. What are the properties of these peaks specific to WT cells?

Response: As the reviewer concerned, only 24% of R-loop peaks were overlapped between WT and KO MEFs, and more unique peaks in the KO cells were found. These results were consistent across three independently repeated experiments, and we attempt to provide a reasonable explanation for these interesting results based on the current knowledge of R-loop regulation. The higher eukaryotes coordinate transcription and replication in the nucleus spatially and temporally to minimize conflict between the two processes. On the one hand, it has been reported that defective transcription termination induces a redistribution of replication initiation factors (MCM2-7 complexes) in the G1-phase (*Gros J, et al. Mol Cell. 2015. PMID: 26656162*). On the other hand, dysregulated origin firing increases R-loop formation (*Hamperl S, et al. Cell. 2017. PMID: 28802045; Lang KS, et al. Cell. 2017. PMID: 28802046*). Since MCM8 depletion in human cells reduced chromatin loading of replication initiation MCM2-7 complexes (*Volkening M, et al. Mol Cell Biol. 2005. PMID: 15684404*), we speculated that a large number of unscheduled R-loop signals (KO unique peaks) and loss of shared signals with WT in KO cells were relevant to

the repositioned RNA polymerases or the changed distribution of replication origins. We will add these contents in the discussion section.

9, In Page 9: "actively proliferating PGCs (EGFP positive) at E12.5..." The authors should explain the marker they used.

Response: We apologize for the inaccurate statement. This sentence should be "actively proliferating PGCs (STELLA-EGFP positive) at E12.5...". STELLA is used as a marker of PGC, and we have successfully constructed STELLA-EGFP reporter mice to isolate and enrich PGCs by fluorescence-activated cell sorting (*Yang Y, et al. Proc Natl Acad Sci U S A. 2022. PMID: 35969748*).

10. In Page 9: ".....as indicated by the normal transcriptional upregulation of KO PGCs when compared to WT PGCs (Fig. EV5A and B)." This reviewer does not think that to draw this conclusion is not really possible only with these ambiguous IF images. The authors should perform some different, more quantitative experiment (e.g., RNA-seq experiment with appropriate spike-ins) to make this conclusion.

Response: Thanks for your valuable suggestion. It was not precise to conclude normal transcriptional upregulation in KO PGCs based on no obvious differences found in global IF signals between WT and KO PGCs. We agree that it is more accurate to quantitatively analyze the global transcription activity by RNA-seq. However, the very small number of remaining PGCs in E11.5 KO embryos makes it very difficult to collect them for RNA-seq using a sorting-based method. To give more convincing evidence, we will apply the 5-ethynyl uridine (EU) incorporation assay to quantify the transcriptional activity by directly imaging nascent RNA. Meanwhile, quantitative analysis of phosphorylation on Ser5 and Ser2 of RNA Pol II indicating the transcription initiation and elongation respectively will be added.

11. In Page 10, the authors stated that "Two genes, *Chchd7* and *Armc10* are shown as examples (Fig. 4F). The biological processes of these two genes with increased R-loop signals were enriched in the positive regulation of developmental growth according to the GO analysis (Fig. 4G)." Here, the meaning of enrichment in developmental growth appears unclear to this reviewer, at least in its current context. Is this term specifically related to the mitotic proliferation of PGCs? It would be beneficial if the authors could clarify the significance of this term. Additionally, the rationale for selecting these two genes is not clearly explained.

Response: In the first sentence, *Chchd7* and *Armc10* are the representative genes with snapshots of R-loop signals for the subsets with shared peaks in Fig. 4D and the KO-only peaks in Fig. 4E, respectively. *Chchd7* was associated with growth including height, reproduction and muscle formation (*Lettre G, et al. Nat Genet. 2008. PMID:*

18391950; Mota LFM, et al. *Genomics*. 2022. PMID: 35671870), and *Armc10* was related to axon regeneration and neurite outgrowth (Xie L, et al. *Sci Transl Med*. 2023. PMID: 37556559) in previous studies. For the second sentence, we apologize for the wrong statement of “The biological processes of these two genes”, which should be “The biological processes of these genes”. In fact, we conducted the GO analysis of all the genes with increased R-loop signals (including the increased signals in shared peaks and the signals in KO-only peaks) in KO MEFs and we found an enrichment of genes related to positive regulation of developmental growth. The R-loop accumulation may impair the expression of these genes and thus the PGC proliferation. The most crucial point we determined by the CUT&Tag assay was the accumulated R-loops after MCM8 deficiency, which led to genome instability and impaired proliferation of PGCs. Therefore, we will revise them in our manuscript.

12. Fig 5A and D: Is GFP-RNH1 fusion protein? If so, why does the majority of the proteins locate in the cytoplasm?

Response: Yes, GFP-RNH1 is a fusion protein. The longest transcript of RNH1 that encoded two isoforms due to the use of alternative translation initiation codons was overexpressed in MEFs. The longer isoform derived from the upstream start codon is a mitochondrial protein, whereas the shorter isoform derived from the downstream start codon is a nuclear protein. In this study, the nuclear isoform was capable of removing the nuclear DNA-RNA hybrids, as indicated by the decrease of R-loop levels in cells upon overexpressing GFP-RNH1 (Fig. 5A and 5B). The reason for the RNH1-GFP showing high signals in the cytoplasm is that the longer isoform is enriched in mitochondria (Shen W, et al. *Nucleic Acids Res*. 2017. PMID: 28977560).

13. In Page 11, the authors stated that "We found that 16.9% (1657/9799) of the MCM8 binding sites co-occurred with R-loop signals (Fig. 6C), with the majority (81.3%, 1347/1657) being located in the promoter regions." From this reviewer's perspective, the observed percentage of 16.9% appears to be quite low to suggest a direct link between MCM8 and R-loop resolution. Could it be that MCM8 is only responsible for a specific subset of R-loops? It would be valuable for the authors to discuss the possible reasons behind this observation or provide additional evidence to support their findings in the text.

Response: Thank you for your insightful question. MCM8 has long been well-known as a DNA helicase to drive replication fork elongation and promote homologous repair. Here, we uncovered a previously unappreciated role of MCM8 in R-loop resolution. Two possible scenarios may explain the results of a low percentage of the MCM8 binding sites being associated with R-loop signals. First, the immunoprecipitation sequences encompass all MCM8-binding sites involved in either

replication fork elongation, HR or R-loop resolution in the CUT&Tag assay; second, co-transcriptional R-loops generated across all the cell cycle phases have been detected all over the genome and their levels are associated with the global transcriptional activity. Given that MCM8 predominantly functions in the S-phase (Gozuacik D, et al. *Nucleic Acids Res.* 2003. PMID: 12527764; Maiorano D, et al. *Cell.* 2005. PMID: 15707891) and prefers binding to R-loop in the promoter regions, MCM8 may be only responsible for resolving a specific subset of S-phase R-loops. Thus, it is not surprising to find that only part of the MCM8 binding sites relate to R-loops. These contents will be added to the discussion section.

14. In Page 11, the authors stated that "R-loop levels are positively associated with transcriptional activity (Wahba et al, 2016)." Based on the authors' data, is it also observed that genes with higher transcriptional activities exhibit more pronounced differences between the KO and WT conditions? If so, this could serve as further confirmation of the data quality.

Response: Thanks for your suggestion. Based on our currently available data, we will analyze whether the R-loop levels exhibit more pronounced differences between the KO and WT in the genes with higher transcriptional activities, especially in the cycle-regulated genes mentioned in the manuscript. Alternatively, we will conduct the RNA-seq of WT and KO MEFs, and try to integrate the transcriptome with the CUT&Tag data to provide further evidence for the relevance between transcriptional activities and R-loop levels.

15. Fig 6H and I: As in other IF figure panels, the data are hardly visible in H and the very minor difference shown in I is not really convincing.

Response: Thanks for your comment. We will optimize the experimental conditions and repeat the experiments in Fig 6H and I.

16. In Page 12 and Fig 7, it would be better for the authors to define what p. R309* and p.S492* stand for, for the clarity of the readers.

Response: Thanks for your comment. Both p.R309* and p.S492* identified in POI patients were the truncated proteins from mutations c.925C>T and c.1475C>A (NM_032485) that led to the premature termination of translation at Arginine 309 and Serine 492, respectively (Heddar A, et al. *J Clin Endocrinol Metab.* 2020. PMID: 32242235; Wang F, et al. *Mol Genet Genomic Med.* 2020. PMID: 32652893). We will supplement the information of the two mutations.

17. What might be the phenotypes in germ cells in Ddx5 or Dhx9 knockouts? This information is useful to include for discussion to understand a broader picture of their

relevance in germ cells.

Response: Thanks for your comment. Lack of DDX5 or DHX9 both lead to embryonic lethality. The Mouse Genome Informatics (MGI) database reports that *Ddx5* null mice die around E11.5 displaying blood vessel malformations and *Dhx9* knockouts die in embryonic stages with massive apoptotic cells in embryonic ectodermal cells. Thus, a conditional deleting strategy is helpful for elucidating the independent roles of these two factors in germ cell development. Interestingly, recent studies demonstrated that DDX5 played essential transcriptional and post-transcriptional roles in the maintenance and function of spermatogonia (*Legrand JMD. Nat Commun. 2019. PMID: 31123254; Xia Q, et al. Cell Prolif. 2021. PMID: 33666296*). In line with these findings, we also found a progressive loss of spermatogonia in *Mcm8*-cKO male mice at around PD3-PD10 by using the *Ddx4*-Cre recombinase (*unpublished data*). Collectively, both MCM8 and DDX5 play crucial roles in the development of germ cells at different stages. We will include these contents in the discussion section.

18. In Page 14, the authors stated that "...actively dividing PGCs were susceptible to MCM8 deficiency; therefore, it was hypothesized that mitotic germ cells are more dependent on MCM8 to resolve specific replication stress, thus safeguarding genome stability. Recently, we reported that rapidly proliferating PGCs are faced with constitutive replication stress, including high levels of R-loops and frequent RF stalling (Yang et al., 2022)." There are many actively dividing cell types during development and therefore, further discussion may be useful why the authors think that PGCs are specifically susceptible to MCM8 loss and R-loop mis-resolution.

Response: Although active proliferation occurs in both PGCs and somatic stem cells in embryos, their characteristics are distinct. In mouse embryos, the period for rapid proliferation in PGCs is limited to between E9.5 and E13.5, and the maximum number of PGCs lays the foundation of the reproductive reserve of both sexes. Of note, because oocytes are not renewable, germ cell loss due to PGC development defects cannot be mitigated or rescued in females in later life. However, a subset of the somatic stem cells expands to renew and maintain the stem cell pools during their lifetime. Although reduced proliferation affects the establishment of the initial stem cell pool, the remaining somatic stem cells can maintain the pool through compensatory proliferation to mitigate the adverse effects on the individual.

Our previous study has shown that increased R-loop levels originating from global upregulation of transcription output is a crucial source of endogenous DNA threats in rapidly proliferating PGCs, and this feature underlies why PGCs are more dependent on the FA pathway to safeguard their genome stability (*Yang Y, et al. Proc Natl Acad Sci U S A. 2022. PMID: 35969748*). In fact, the specific loss of germ cells was also

observed in other ubiquitously expressed DNA repair gene knockout mouse models, such as *Mcm9* and *Rad54* (Luo Y, et al. *Genesis*. 2015. PMID: 26388201; Messiaen S, et al. *Cell Death Dis*. 2013. PMID: 23949223). These findings suggest that PGCs could be considered as a group of special cells that are more dependent on DNA repair factors to respond to endogenous genome threats, thus maintaining a high level of genome stability. When DNA damage occurs, they will be more sensitive and thus initiate more strict monitoring mechanisms, such as p53 pathway activation and cell cycle regulation to repair the DNA damage, thus resulting in the severe loss of PGCs. In this scenario, we showed that PGCs lacking MCM8 were more susceptible to R-loop accumulation-induced DNA damage, and their impaired proliferation caused an insufficient reproductive reserve. We will expand our discussion to address this issue.

Minor points:

1. In Page 7, the authors stated that "Because PGCs are sensitive to DNA damage, we reasoned that PGCs had an increased requirement for MCM8 in genome stability maintenance under physiological conditions and that unrepaired DNA damage would result in apoptosis or cell cycle arrest in MCM8-deficient PGCs." In this sentence, the authors should consider including a citation to support the statement that "PGCs are sensitive to DNA damage."

Response: Thanks for your suggestion. There are several previous studies supporting the statement that "PGCs are sensitive to DNA damage (Hill RJ, Crossan GP. *Nat Genet*. 2019. PMID: 31367016; Luo Y, et al. *PLoS Genet*. 2014. PMID: 25010009)". We will add these citations in our manuscript.

2. In Page 8, the authors stated that "We removed 25 ribosomal proteins that were always found among the nonspecific binding proteins in the IP-MS assay and analyzed the remaining 84 binding proteins." Is there any evidence or citation to validate the exclusion of ribosomal proteins in this IP-MS analysis?

Response: Thanks for your suggestion. As we mentioned in above question 4, there is evidence showing nonspecific binding of ribosomal proteins in the immunoprecipitation assay (Mellacheruvu D, et al. *Nat Methods*. 2013. PMID: 23921808; Mohammed H, et al. *Nat Protoc*. 2016. PMID: 26797456). We will revise the statement and include citations in our manuscript.

3. In Page 10, "... and KO embryos (Fig. 4A)" should be "KO MEFs."

Response: We are sorry for this mistake. The "embryos" should be changed to "MEFs.". We will correct this sentence.

4. In Page 13, the authors stated that "Furthermore, overexpression of WT MCM8 in

KO HEK293 cells showed a drastic decrease in R-loop levels compared to KO cells." Please add a citation to the relevant figure.

Response: Thanks for your suggestion. Fig. 7D and E are relevant to this sentence, and we will add the citation.

5. In Page 17, the authors stated that "In addition, we also showed that p.E341K (rs16991615), which had normal interaction with DDX5, did not show equally efficient R-loop removal as WT MCM8." Please add a citation to the relevant figure.

Response: Thanks for your suggestion. Fig. 7C-E are relevant to this sentence, and we will add the citation.

Referee #2

It has been reported that MCM8 forms both MCM8/9 dimer and hexamer (Gambus and Blow, *Cell Cycle*, 2013; McKinzey et al., *NAR*, 2023; Weng et al. *eLife*, 2023; Acharya et al. *BioRxiv*, 2023). Furthermore, studies with MCM9-KO mice have demonstrated a decrease in PGCs (Hartford et al. *PNAS*, 2011; Luo and Schimenti, *Genesis*, 2015). However, the authors favour the idea that MCM8 functions in R-loop resolution without MCM9, as shown in Fig. 7F and discussed in Discussion. The authors should provide clarification regarding whether MCM9 is dispensable for R-loop resolution and whether MCM8 operates autonomously from MCM9.

Response: Thank you for your insightful question. The MCM8/9 complex not only participates in DNA synthesis but also exerts a variety of biological functions to orchestrate HR repair (Lee KY, et al. *Nat Commun.* 2015. PMID: 26215093; Park J, et al. *Mol Cell Biol.* 2013. PMID: 23401855). But deficiency of each of them didn't generate identical phenotypes among cells or mice even in the same background (Nishimura K, et al. *Mol Cell.* 2012. PMID: 22771115; Lutzmann M, et al. *Mol Cell.* 2012. PMID: 22771120; Luo & Schimenti. *Genesis.* 2015. PMID: 26388201). For example, almost complete loss of MCM9 was detected in *Mcm8* knockout cells, whereas much MCM8 remained in *Mcm9* knockout cells (Nishimura K, et al. *Mol Cell.* 2012. PMID: 22771115). *Mcm8*-null male mice were devoid of post-meiotic cells and were completely sterile, but *Mcm9*-null males produced functional spermatozoa and were fertile (Lutzmann M, et al. *Mol Cell.* 2012. PMID: 22771120). Based on these findings, we reasoned that they could exert complex-independent biological functions. To further validate this hypothesis, we will perform the co-immunoprecipitation (co-IP) assay using the S9.6 antibody to find whether MCM9 is enriched in the immunoprecipitated proteins. Furthermore, we will perform the co-IP assay using anti-MCM9 antibody in P19 cells to detect whether MCM9 interacts with R-loop regulators DDX5 and DHX9.

Minor points

- Reconsider the figure order. Fig. 5 A, B should directly follow Fig. 3.

Response: Thanks for your suggestion. We will change the figure order to make the manuscript more readable.

- Show control HEK293 WT cells in Figure 7D, E.

Response: Thanks for your suggestion. We will re-perform the immunofluorescence staining of R-loops and show control WT HEK293 cells alongside MCM8 KO HEK293 cells expressing either wild-type or mutant MCM8.

- HROB is required for MCM8/9 helicase activity and works with MCM8/9 on in ICL repair and HR (Hustedt et al., G&D, 2019; Huang et al. Nat. Commun., 2020). This point should be mentioned in Introduction.

Response: Thanks for your suggestion. We will add the introduction of HROB and its function related to MCM8/9 in Introduction.

Referee #3

Overall the study is well executed and reports interesting findings, in part confirming previous observation on the role of the MCM8 DNA helicase in DNA replication and germ cells metabolism. The observed interaction between MCM8 and Ddx5/Dhx9 is novel but somehow surprising. However, I found the claim that MCM8 KO increases the frequency of R-loops in PGCs needs to be strengthened. Also, I do not understand why the authors have not explored a potential role for MCM8 in regulating splicing, as strongly suggested by the MCM8 interactome, and they instead focused on a possible role of MCM8 on R-loops metabolism. Mutations that affect splicing indirectly generate replication stress and consequent R-loop increase, as originally shown (Paulsen et al., 2009 Mol Cell; PMID: 19647519; Gomez-Gonzalez et al., 2009 Mol Cell Biol; PMID: 19651896). Further, from data shown in Fig. 3B, the Numal protein appears to be the most abundant MCM8 interactor. Numal is a protein involved in the correct orientation of the mitotic spindle but it has also recently shown to localize to gene promoter regions (Ray et al., Nature 2022; PMID: 36171374). Numal depletion generates DNA damage and genomic instability, phenotypes similar to those observed in MCM8 KO cells, and consistent with the increase in the fraction of cells in the G2 phase of the cell cycle shown in Fig. 2F. Hence, I do not see the logic of focusing on Ddx5 and especially Dhx9, this latter being not the most abundant MCM8 interactor compared to at least two other Ddxs involved in R-loops resolution such as Ddx21 and Ddx41, unless authors have reasons to believe that these

latter do not contribute to the genomic instability in PGCs upon MCM8 KO. Finally, although the authors claim a function for MCM8 for R-loops metabolism in PGCs, the critical data have been mostly obtained using P19, MEFs and HEK293T cells. Here below are comments that may help the authors to improve the manuscript.

Response: Thank you for your insightful questions. MCM8 is a well-known factor involved in DNA damage repair and genome stability for mitotic cells (*Lee KY, et al. Nat Commun. 2015. PMID: 26215093; Lutzmann M, et al. Mol Cell. 2012. PMID: 22771120; Nishimura K, et al. Mol Cell. 2012. PMID: 22771115*). The maintenance of genome stability is essential for both mitotic germ cells and somatic cells. Because germ cells with a low mutation rate have super-stable genomes, they must rely on more robust DNA damage response (DDR) mechanisms to ensure their genome stability. Recently, specific loss of PGCs in ubiquitously expressed DDR gene knockout mouse models was attributed to a more pronounced accumulation of endogenous DNA damage, indicating that unique origins of endogenous genome threats may underpin higher demands for these DDR factors during PGC development (*Hill RJ, Crossan GP. Nat Genet. 2019. PMID: 31367016; Yang Y, et al. Proc Natl Acad Sci U S A. 2022. PMID: 35969748*).

Our recent study has shown that increased R-loop levels originating from global upregulation of transcription is a crucial source of endogenous DNA damage in rapidly proliferating PGCs, thus necessitating the functional FA pathway to safeguard genome stability and proliferation of PGCs (*Yang Y, et al. Proc Natl Acad Sci U S A. 2022. PMID: 35969748*). Interestingly, *Mcm8* KO mice also displayed profound PGC loss, without obvious abnormalities of somatic systems. Therefore, we focus on elucidating the role of MCM8 in the resolution of endogenous DNA damage in PGCs. Then, we found that MCM8 may be related to R-loop regulation through its interactors from IP-MS. Later, we verified the increase of R-loops after MCM8 deficiency in PGCs and confirmed the interaction of MCM8 with R-loop-resolving factors DDX5 and DHX9 in sorted WT PGCs. However, limited by the scarcity of the KO PGC population, we predominantly conducted functional experiments in P19, MEF, and HEK293 cells, and these consistent results indicated that MCM8 was capable of suppressing R-loops, a function not restricted to germ cells, but also universal in somatic cells. Because of the increased requirement for preserving genome stability, PGCs lacking MCM8 were more susceptible to R-loop-induced DNA damage, showing dramatically decreased proliferation.

As you mentioned, there may be a potential involvement of MCM8 in regulating splicing. We will perform RNA-seq in WT/KO MEF and analyze the alternative splicing events.

Among the R-loop regulators collected in the database (*Lin R, et al. Nucleic Acids Res. 2022. PMID: 34792163*), some are validated as the R-loop binding proteins by

biological experiments, whereas a large proportion of them including NuMA are obtained from high-throughput screening studies (*Cristini A, et al. Cell Rep. 2018. PMID: 29742442; Wu T, et al. Mol Cell Proteomics. 2021. PMID: 34478875; Mosler T, et al. Nat Commun. 2021. PMID: 34916496*), but lack verification. Recently, it has been reported that a fraction of NuMA in the interphase nucleus promotes gene transcription by limiting the polyADP-ribosylation of RNA polymerase II and repairs oxidative DNA breaks (*Swagat Ray, et al. Nature. 2022. PMID: 36171374*). In this study, NuMA depletion increased genome instability resulting from oxidative DNA breaks at regulatory genome elements during transcription activation. Therefore, the role of NuMA in transcription activation and DNA repair indicates potential reciprocal regulation between NuMA and MCM8, which needs further exploration.

In this study, we found that many interactors were related to R-loop regulation in MCM8 interactome, including DDX5, DHX9, DDX21 and DDX41 (*Lin R, et al. Nucleic Acids Res. 2022. PMID: 34792163*). For DDX5 and DHX9, multiple lines of evidence from experimental studies have verified that they repress R-loops by unwinding the DNA-RNA hybrids in both human and mouse cells, and their R-loop-resolving activities are regulated by multiple co-factors (*Sessa G, et al. EMBO J. 2021. PMID: 33634895; Yuan W, et al. Nucleic Acids Res. 2021. PMID: 34329467*). For DDX21 and DDX41, they have not been demonstrated in R-loop resolution in mouse cells. Therefore, DDX5 and DHX9 were selected as representative factors for verification in our study. Yet, it cannot be ruled out that MCM8 also suppresses R-loops by interacting with DDX21, DDX41 or other factors.

1. Fig. 2A-B, the difference in the number of cells showing cleaved PARP-1 staining between wild-type and MCM8 KO is very weak. The quantification shown in panel B does not match the data shown in panel A. Same applies to Fig. 2E-F for Cyclin B1 and EdU. I do not understand how authors can identify G1 and G2 cells using EdU (S-phase) and Cyclin B1 (mitosis) staining. From these data one cannot say that cells are arrested in the cell cycle. If one believes these data, what appears is that there is an increase in the number of G2 cells, but this does not mean that cells are arrested. Checking activation of the DNA damage response (ATM and/or ATR, Chk1/Chk2) would have been more convincing.

Response: Thank you for your comments. During the evaluation of PGC apoptosis, we also performed the cleaved-PARP1 immunofluorescence staining in a positive control group from pregnant mice treated with an intraperitoneal injection of mitomycin C (MMC). After MMC treatment, more cleaved-PARP1 positive PGCs were detected in E11.5 genital ridges. However, the apoptotic PGCs in the MMC-treated group were also few, which is consistent with the results in MCM8-deficient PGCs. Usually, only 1~2 apoptotic PGCs were found per genital ridge section, and the

absolute number was fewer than 10 per *Mcm8*^{-/-} embryo. Therefore, only the representative images with positive cleaved PARP-1 staining were selected to be displayed in the submitted manuscript. We will provide a representative image without apoptotic PGCs for the WT group to match the statistical data shown in panel B.

For cell cycle phase determination, we applied immunofluorescence staining of Cyclin B1 combining with EdU incorporation assay which is a reliable method for cell cycle analysis, especially when only a small amount of target cells could be available (*Eastman AE, et al. FEBS Lett. 2020. PMID: 32441778*). Cyclins are proteins that are distinguished by their steady accumulation in interphase followed by specific and rapid proteolysis at mitosis (*Pines J. Cell Growth Differ. 1991. PMID: 1648379*). Cyclin B1 that accumulates in G2-phase is predominantly cytoplasmic until just before mitosis, and then it translocates into the nucleus at the beginning of mitosis (*Pines J, et al. J Cell Biol. 1991. PMID: 1717476; Hagting A, et al. Curr Biol. 1999. PMID: 10395539*). Therefore, the G2-phase cells can be identified by the presence of high levels of Cyclin B1 in the cytoplasm (*Seki Y, et al. Development. 2007. PMID: 17567665*), whereas accumulated Cyclin B1 in the nucleus indicates the M-phase cells and the G1-phase cells are characterized by negative staining of both Cyclin B1 and EdU. Our results showed that the proportion of S-phase PGCs decreased and G2-phase PGCs increased, suggesting a prolonged cell cycle which was consistent with the longer doubling time of KO PGCs calculated from our data (shown in the table below). Accordingly, we will revise the statements of the cell cycle.

According to your suggestion, we will perform ATM and CHK2 immunofluorescence staining in PGCs to detect the DNA damage response to the DSB formation.

Phenotype	PGC number			Doubling time (h)
	E8.5	E9.5	E11.5	
WT	100±10	254±62	1861±592	16.7
KO	95±9	110±26	471±169	22.9

Note: T doubling = $48/\log_2(N_{E11.5}/N_{E9.5})$ h

2. In Fig. 2G-H authors show that MCM8 KO PGCs display a strong increase in the number of 53BP1 foci compared to wild-type PGCs and conclude that DNA damage accumulation leads to decreased PGCs proliferation. However, authors did not provide additional data to prove this statement. Should this be the case, then proliferation should be rescued by inhibiting the DNA damage response (probably the ATM pathway, not shown in the manuscript).

Response: Thank you for your comments. We know that the DDR is initiated upon sensing damage, and then complex signal transduction is activated to promote DNA repair, including prolonging the cell cycle to allow more time for repair or eliminating cells containing severe unrepaired DNA damage by apoptosis (*Lanz MC, et al. EMBO J. 2019. PMID: 31393028; Blackford AN, et al. Mol Cell. 2017. PMID: 28622525*). In our study, decreased PGC proliferation in MCM8 KO embryos may be mainly due to the prolonged cell cycle. Since effective DDR is essential for cell survival and proliferation, especially for PGCs, inhibition of the DDR pathway such as the ATM pathway may result in PGC death. Therefore, it may not be an efficient rescue strategy in this context. We will perform additional experiments, such as γ H2AX foci detection or comet assay to strengthen our statement that the DNA damage accumulates in MCM8 KO PGCs.

3. Fig. 3, it is indeed curious that the MCM8 interactome showed a great enrichment of RNA splicing factors. Why the authors have not explored a potential role of MCM8 in splicing? What about the best hit, Numa1? Did the authors identified MCM9 in the proteome, which is a stable MCM8 partner? I could not find the list of the proteins identified by mass spectrometry. A western blot and silver stain of the immunoprecipitated material must be shown.

Response: Thank you for your insightful questions. As answered above, we will perform RNA-seq in WT/KO MEF and analyze the alternative splicing events to determine whether MCM8 is involved in RNA splicing.

NuMA is the top hit in our MCM8 interactome. As described above, the role of NuMA in DNA repair indicates potential reciprocal regulation between NuMA and MCM8, which needs further exploration.

MCM9 is a stable partner of MCM8, which has been confirmed by immunoprecipitation assay (*Nishimura K, et al. Mol Cell. 2012. PMID: 22771115*), and they form a complex to function in the replication and HR repair. However, MCM9 wasn't identified in the proteome, which was probably attributed to its low endogenous abundance under the physiological condition (*Beck M, et al. Mol Syst Biol. 2011. PMID: 22068332*). We will perform co-IP and western blot to verify their interactions.

We will also show the original images of gels with Coomassie brilliant blue staining and western blot of the immunoprecipitated material and add more details in the "IP assay" part of the Materials and methods. In detail, immunoprecipitated proteins from the IgG and MCM8 groups were loaded for electrophoresis and then Coomassie brilliant blue staining was performed. After washing the excessive dye, gel slices were cut off for subsequent mass spectrometry analysis. Therefore, we acquired all the potential interactors of MCM8 in P19 cells. The list of the proteins identified

by mass spectrometry will be included in the supplementary data.

4. Fig. 3C-E, 6F, 7B-C, do the western blot signals belong to the same gel? This is an important point in order to compare the relative signal intensity. In the immunoprecipitation experiments, can the authors show a protein that does not interact with MCM8? Can they immunoprecipitated MCM9?

Response: Thank you for your comments. The signals in Fig. 3C-E were used to validate the interactions qualitatively, and they belonged to the same gel and were generated by stripping and re-incubating with desired antibodies. In Fig. 6F, 7B-C, all compared groups belonged to the same gel. However, the signals of the Input or IP samples were generated separately.

In the immunoprecipitation experiments, we did find proteins that did not interact with MCM8, such as PCNA. We will show the data. As we mentioned in question 3, we will perform co-IP and western blot to verify the interaction between MCM8 and MCM9 in P19 cells.

5. Fig. 3F-G, it is important to include a treatment with RNaseH1 to show the specificity of the S9.6 signal. Also, the use of RNaseH1-dead mutant has been now shown to be a more reliable tool to detect R-loops than the S9.6 antibody by immunofluorescence (Crossley et al., JCB 2021; PMID: 34232287).

Response: Thank you for your comments. We agree with you that it's more reliable to detect R-loop by using RNase H1-dead mutant. Transfection of RNase H1-dead mutant is eminently suitable for detecting R-loops in cell lines *in vitro* (Chen L, et al. *Mol Cell* 2017. PMID: 29104020; Ginno PA, et al. *Mol Cell* 2012. PMID: 22387027; Legros P, et al. *PLoS Genet.* 2014. PMID: 25392932). However, it's hard to carry out in PGCs *in vivo*. Alternatively, we will include a group with RNase H1 treatment to show the specificity of the S9.6 signals in Fig. 3F-G.

6. It would have been nice to have a DDX5 knock out (or downregulation) control aside both immunofluorescence and Chip data to compare with. Ideally, one wished to see how Ddx5 or Dhx9 affects R-loops distribution genome-wide and compare it with that observed in MCM8 KO cells.

Response: Thank you for your suggestions. It has been reported that increased nuclear S9.6 staining (Mersaoui SY, et al. *EMBO J.* 2019. PMID: 31267554) and the enrichment of R-loop signals at promoters, transcription start sites and gene bodies are displayed in DDX5-depleted cells (Villarreal OD, et al. *Life Sci Alliance.* 2020. PMID: 32747416; Sessa G, et al. *EMBO J.* 2021. PMID: 33634895). As you suggested, we will perform immunofluorescence staining and compare the change of R-loop distribution genome-wide in DDX5-depleted cells with that observed in

MCM8 KO cells. If the data is inappropriate for combined analysis, we will perform the R-loop CUT&Tag assay in DDX5 KO or knockdown cells in order to obtain comparative results.

7. In Fig. 5C authors analyze replication dynamics by DNA fiber stretching and conclude that replication speed is slower in MCM8 KO cells, in line with previous reports demonstrating a function of MCM8 in contributing to replication fork progression (Maiorano et al., Cell 2005; Gambus et al., Cell Cycle 2013; Griffin et al., 2022). However, according to these data, it appears that MCM8 depletion reduced the length of both IdU and CldU tracks, suggesting an effect on both the initiation and elongation steps of DNA synthesis. However, I am not convinced that this is due to R-loops accumulation, as claimed by the authors. The statistical test here and elsewhere is inappropriate (see other points below). Significance must be recalculated using a Man-Whitney test. Further, images shown in panel D are not representative of the quantification shown in panel E. Concerning the possible increase in R-loops upon MCM8 KO, this could be due to reduce DNA synthesis rate generating transcription-replication conflicts. Hence it is expected that inhibition of transcription should reduce R-loops formation, thus rescuing the effect of MCM8 KO. Fig. 5D-E, the reduction of 53BP1 foci in MCM8 KO cells treated with RNaseH1 is weak and quantification does not match the data.

Response: Thank you for your comments. In the DNA fiber assay, IdU signal was used to indicate the active replication fork (RF), and the length of CldU tracks of progressing RFs was used to evaluate RF velocity. As you mentioned, slower replication fork elongation may also contribute to the observed short CldU length in MCM8 KO cells. Here, our findings suggest that the inability to resolve R-loops also contributed to the slower RF progression in MCM8 KO cells.

When increasing or decreasing RNase H1 activity in the nucleus alters phenotypes of interest, it could be reasoned that R-loops specifically contribute to these phenotypes. Therefore, we overexpressed RNase H1 to remove the R-loops and showed that the reduced replication progression and increased DNA damage levels could be partially recovered in KO MEF, supporting that R-loop accumulation contributed to the shorter RF velocity and increased DNA damage in KO MEFs.

As you suggested, reducing R-loop levels by global inhibition of transcription, such as adding Actinomycin D (*Mosler T, et al. Nat Commun. 2021. PMID: 34916496*), is an alternative approach to determine whether DNA damage derives from unresolved R-loops. However, globally suppressing transcription output may affect the expression of important genes, especially those responsible for basic functions, and thus the results may be nonspecific and difficult to interpret.

As you mentioned, the statistical method we used was improper and the Man-

Whitney test will be better. We will re-perform the statistical analysis as you suggested. We will also replace the representative image in Fig. 5D, in which cells with 53BP1 > 5 foci were counted.

8. Fig 6, a western blot showing the expression of Flag-MCM8 must be shown. Panel E, a control including RnaseH1 treatment and ChiP with a non-specific antibody (Flag alone as a control for Flag-MCM8 Chip and non-specific for the S9.6 Chip) must be included to ascertain the specificity of the immunoprecipitated material. Panel F, it is critical here to show that equal amounts of IgGs were recovered in all lanes, since based on this experiment authors conclude that Ddx5 and Dhx9 retention is reduced on R-loops upon MCM8 KO.

Response: Thank you for your suggestions. We will show the western blot of the expression of Flag-MCM8 related to Fig. 6. As you suggest, it's more precise to include a Flag alone control for Flag-MCM8 CHIP and a control of RnaseH1 treatment for the S9.6 ChIP. Therefore, we will perform these experiments to ascertain the specificity of the immunoprecipitated material and re-analyze the binding sites. In Fig. 6F, we added the same amounts of antibodies to do the immunoprecipitation assay but didn't show the recovered amount of IgGs in the original figure. We will supplement IgG signals detection to illustrate this point.

9. Fig. 7, authors complement R-loops generated by MCM8 KO in HEK293T cells by expressing either wild-type or divers MCM8 mutants. Here again, the images provided in Fig. 7C do not match the quantification shown in panel E. The differences in S9.6 signals observed in these images are very weak.

Response: Thank you for your comments. Because of the compression of IF images in the initially submitted manuscript, the S9.6 IF signals were weak and ambiguous. We will provide more clear and representative images to match the quantification data shown in panel E.

10. Fig. EV6, a quantification of the Ddx5 and Dhx9 immunofluorescence signals obtained must be shown.

Response: Thank you for your suggestion. We will quantify the DDX5 and DHX9 immunofluorescence signals.

Other points

1. Page numbers are missing

Response: We will add the page numbers.

2. Introduction, second page, MCM2-7 protein are not degraded in S-phase, they are

removed from chromatin by ongoing DNA synthesis (Todorov et al., J. Cell Biol. 1995). Next page, Ddx5 and Dhx9 are not "the" R-loops removing factors but are R-loops-removing factors. A general introduction to factors known so far in R-loops resolution (including other Ddx helicases) is missing.

Response: Thank you for your suggestions and we will modify the statement. In the introduction section, we wrote that "It is reported that MCM8 binds to chromatin and functions as a DNA helicase to drive replication fork (RF) progression after the core replicative helicase subunit MCM2 is degraded.". In the previous study (*Natsume T, et al. Genes Dev. 2017. PMID: 28487407*), the authors induced the rapid degradation of MCM2 by auxin-inducible degron (AID) technology to explore the possible role of MCM8-9 in replication progression and found that MCM8-9 acted as an alternative replicative helicase to promote DNA synthesis. Therefore, we will revise the sentence to "It is reported that MCM8 binds to chromatin and functions as a backup DNA helicase to promote replication fork (RF) progression when the core replicative helicase subunit MCM2 is deficient". In addition, for the sentence "Ddx5 and Dhx9, the R-loop resolving factors", we will remove "the". What's more, we will supplement the introduction of the known R-loop-resolving factors as you suggest.

3. Fig. 3F-G, it is unclear what the white arrows indicate (whether cytoplasmic R-loops or cells displaying nuclear R-loops); this has not been described in the figure legend.

Response: We are sorry for this missing information. The white arrows indicate representative cells displaying nuclear R-loops in Fig. 3F-G. We will add this in the figure legend.

4. Fig. 7F, what is the evidence that Ddx5 and Dhx9 function as dimers (or multimers) in R-loops resolution? It is entirely possible that resolution of a fraction of R-loops depends upon MCM8-Ddx5 and/or Dhx9 interaction. It is for this reason that it would have been nice to have a picture of R-loops distribution in Ddx or Dhx9 knock down cells.

Response: Thank you for your suggestion. It has been reported that DDX5 and DHX9 also work together in a complex to resolve R-loops (*Kim S, et al. Nucleic Acids Res. 2020. PMID: 32542338*). However, it was inaccurate in Fig. 7F to show DDX5 and DHX9 as dimers. We will amend our working model to show mutual interactions of the three proteins and the reduction of DDX5 and DHX9 at R-loops without MCM8. Furthermore, as we mentioned in above question 6, we will analyze the R-loop distribution in DDX5-depleted cells.

5. A Mann-Whitney U test is more appropriate to determine significance in several

experiments (Fig. 3G; 5B-C; 7E).

Response: Thank you for your suggestions. We will carefully check and modify the statistical analysis in several experiments (Fig. 3G; 5B-C; 7E) as you suggest.

Dr. Yingying Qin
Shandong University
44 Wenhua Xi Road
Jinan, Shandong 250021
China

2nd Oct 2023

Re: EMBOJ-2023-115128
MCM8 interacts with DDX5 to promote R-loop resolution in primordial germ cells

Dear Dr. Qin,

Thank you for sending me your detailed tentative responses and revision plan for your recent EMBO Journal submission on MCM8 in primordial germ cells. I have now had a chance to carefully consider them. I appreciate that your answers should potentially clarify most of the key concerns of the referees, and would therefore be willing to give you an opportunity to revise and resubmit a modified version of this study. Should you need an extended revision period, please let me know, and as always, competing manuscript published during the course of this revision will not affect our final decision on your study. But please be reminded that in light of our single-major-revision-round policy, it will be important to convince the critical referees with the additional data and clarifications at this stage.

Please also note the additional information and more detailed guidelines on how to prepare a revision below (and in our online Guide to Authors) - closely adhering to them shall greatly facilitate the editorial process at the time of resubmission.

Thank you again for the opportunity to consider this work, and I look forward to receiving your revised manuscript in due time.

Yours sincerely,

Hartmut Vodermaier

5) Point-by-point response letters should include the original referee comments in full together with your detailed responses to

them (and to specific editor requests if applicable), and also be uploaded as editable (e.g., .docx) text files.

9) Digital image enhancement is acceptable practice, as long as it accurately represents the original data and conforms to community standards. If a figure has been subjected to significant electronic manipulation, this must be clearly noted in the figure legend and/or the 'Materials and Methods' section. The editors reserve the right to request original versions of figures and the original images that were used to assemble the figure. Finally, we generally encourage uploading of numerical as well as gel/blot image source data; for details see: embopress.org/page/journal/14602075/authorguide#sourcedata

At EMBO Press, we ask authors to provide source data for the main manuscript figures. Our source data coordinator will contact you to discuss which figure panels we would need source data for and will also provide you with helpful tips on how to upload and organize the files.

In the interest of ensuring the conceptual advance provided by the work, we recommend submitting a revision within 3 months (31st Dec 2023). Please discuss the revision progress ahead of this time with the editor if you require more time to complete the revisions. Use the link below to submit your revision:

Link Not Available

Referee #1:

The manuscript by Wen et al. explored the function of Mcm8, a key gene known to function in homologous recombination repair and implicated in premature ovarian insufficiency (POI), during primordial germ cell (PGC) development in mice. The authors showed that an Mcm8 knockout (KO) leads to substantial loss of PGCs and infertility in both males and females. The authors provided multiple lines of evidence showing that MCM8 interacts with R-loop regulators such as DDX5 and DHX9 to promote the resolution of R-loops, in particular, those around gene promoters, during rapid proliferation of PGCs, and the failure in such processes leads to DNA damage accumulation and impaired PGC proliferation. Furthermore, the authors showed that Mcm8 mutants causative for POI display weaker interactions with DDX5 and leads to increased R-loop formation. This is a potentially interesting manuscript that provides an insight into the role of MCM8 and the significance of R-loop resolution during PGC development, which has a relevance in better understanding the etiology of POI.

Several concerns regarding the data and presentations in the current version are as follows:

Major concerns:

1. In figures showing immunofluorescence (IF) staining data (Fig 1A, B, 2A, 2C, 2E, EV4A, etc), the authors should provide more magnified, clear images so that the readers can evaluate the authors' statements more precisely.

2. In Page 6, the authors stated that "Compared to PGCs in WT littermates and single mutants of Mcm8 KO or ubiquitination-defective Fancd2, significantly fewer PGCs were detected in the double mutants (1467.0 {plus minus} 307.4 vs. 436.0 {plus minus} 130.6 vs. 300.6 {plus minus} 117.8 vs. 43.2 {plus minus} 6.4, $P < 0.001$), suggesting that MCM8 has functions in PGC development that are independent of the FA pathway (Fig. EV4B)." According to Hill and Crossan, 2019, the DNA damage response (DDR) pathway is important in PGCs, especially in the context of the FA pathway. Furthermore, the FA pathway is also crucial for R-loop resolution (García-Rubio et al., 2015). Given these findings, which DDR pathway do the authors believe the results related to R-loop accumulation in Mcm8 KO are more closely associated with?

3. In Page 7, the authors stated that "MCM8 deficiency resulted in a higher proportion of apoptosis when compared to the WT group (1.05 {plus minus} 0.42% vs. 0.24 {plus minus} 0.14%, $P < 0.05$), but the number of apoptotic PGCs was less than 10 per embryo (Fig. 2A and B)." Why then does the number of PGCs decrease so dramatically in Mcm8 KO mice?
4. In Page 8, the authors stated that "We removed 25 ribosomal proteins that were always found among the nonspecific binding proteins in the IP-MS assay and analyzed the remaining 84 binding proteins. We noted that the MCM8 interactors were mainly involved in RNA splicing processes, as indicated by Gene Ontology (GO) analysis (Fig. 3A)." The GO term "RNA splicing processes" is a also broad term, and many proteins related to this process are expected to be abundant in the nucleus, similar to "ribosomal proteins." This reviewer is concerned that this observation may also be attributed to noise. Therefore, conducting a comparative analysis for MCM8-IP in cells less abundant for MCM8 as a negative control may further support the authors' claim.
5. Fig 3B: Related to the above point, this analysis should be presented alongside an appropriate negative control to confirm that the "MCM8 interactome" significantly overlaps with "Human and mouse R-loop regulators."
6. Fig 3F, 5A, 7D: I do not think it really possible to make quantitative comparison of the IF signals between different specimens with the method employed by the authors, as the staining levels and intensities can easily be changed and very difficult to control between different samples. To overcome this point, for example, the authors should place the cells with two different genotypes on the same slide and then compare the signals. Why were S9.6 signals so strong only in single puncta in the nuclei?
7. Fig 4B and Fig 4C: Judging from the pattern observed in the "non-Antibody" track in Fig 4B, the signal-to-noise (S/N) ratio of the IP tracks may not be optimal. Therefore, in this case, it is crucial to mention in detail in the "CUT&Tag assay" section that peak calling was performed using the "non-Antibody" track as the input track in MACS2.
8. Fig 4C-E: This reviewer wonders why the peaks in the WT condition are not entirely encompassed by those in the KO condition. Given the authors' primary claim that the KO condition leads to more frequent R-loops, one would expect the KO to exhibit additional peaks in addition to those found in the WT. However, Fig 4C shows many unique peaks in the WT. What are the properties of these peaks specific to WT cells?
9. In Page 9: "actively proliferating PGCs (EGFP positive) at E12.5..." The authors should explain the marker they used.
10. In Page 9: ".....as indicated by the normal transcriptional upregulation of KO PGCs when compared to WT PGCs (Fig. EV5A and B)." This reviewer does not think that to draw this conclusion is not really possible only with these ambiguous IF images. The authors should perform some different, more quantitative experiment (e.g., RNA-seq experiment with appropriate spike-ins) to make this conclusion.
11. In Page 10, the authors stated that "Two genes, Chchd7 and Armc10 are shown as examples (Fig. 4F). The biological processes of these two genes with increased R-loop signals were enriched in the positive regulation of developmental growth according to the GO analysis (Fig. 4G)." Here, the meaning of enrichment in developmental growth appears unclear to this reviewer, at least in its current context. Is this term specifically related to the mitotic proliferation of PGCs? It would be beneficial if the authors could clarify the significance of this term. Additionally, the rationale for selecting these two genes is not clearly explained.
12. Fig 5A and D: Is GFP-RNH1 fusion protein? If so, why does the majority of the proteins locate in the cytoplasm?
13. In Page 11, the authors stated that "We found that 16.9% (1657/9799) of the MCM8 binding sites co-occurred with R-loop signals (Fig. 6C), with the majority (81.3%, 1347/1657) being located in the promoter regions." From this reviewer's perspective, the observed percentage of 16.9% appears to be quite low to suggest a direct link between MCM8 and R-loop resolution. Could it be that MCM8 is only responsible for a specific subset of R-loops? It would be valuable for the authors to discuss the possible reasons behind this observation or provide additional evidence to support their findings in the text.
14. In Page 11, the authors stated that "R-loop levels are positively associated with transcriptional activity (Wahba et al, 2016)." Based on the authors' data, is it also observed that genes with higher transcriptional activities exhibit more pronounced differences between the KO and WT conditions? If so, this could serve as further confirmation of the data quality.
15. Fig 6H and I: As in other IF figure panels, the data are hardly visible in H and the very minor difference shown in I is not really convincing.
16. In Page 12 and Fig 7, it would be better for the authors to define what p. R309* and p.S492* stand for, for the clarity of the readers.
17. What might be the phenotypes in germ cells in Ddx5 or Dhx9 knockouts? This information is useful to include for discussion to understand a broader picture of their relevance in germ cells.
18. In Page 14, the authors stated that "...actively dividing PGCs were susceptible to MCM8 deficiency; therefore, it was

hypothesized that mitotic germ cells are more dependent on MCM8 to resolve specific replication stress, thus safeguarding genome stability. Recently, we reported that rapidly proliferating PGCs are faced with constitutive replication stress, including high levels of R-loops and frequent RF stalling (Yang et al., 2022)." There are many actively dividing cell types during development and therefore, further discussion may be useful why the authors think that PGCs are specifically susceptible to MCM8 loss and R-loop mis-resolution.

Minor points:

1. In Page 7, the authors stated that "Because PGCs are sensitive to DNA damage, we reasoned that PGCs had an increased requirement for MCM8 in genome stability maintenance under physiological conditions and that unrepaired DNA damage would result in apoptosis or cell cycle arrest in MCM8-deficient PGCs." In this sentence, the authors should consider including a citation to support the statement that "PGCs are sensitive to DNA damage."
2. In Page 8, the authors stated that "We removed 25 ribosomal proteins that were always found among the nonspecific binding proteins in the IP-MS assay and analyzed the remaining 84 binding proteins." Is there any evidence or citation to validate the exclusion of ribosomal proteins in this IP-MS analysis?
3. In Page 10, "... and KO embryos (Fig. 4A)" should be "KO MEFs."
4. In Page 13, the authors stated that "Furthermore, overexpression of WT MCM8 in KO HEK293 cells showed a drastic decrease in R-loop levels compared to KO cells." Please add a citation to the relevant figure.
5. In Page 17, the authors stated that "In addition, we also showed that p.E341K (rs16991615), which had normal interaction with DDX5, did not show equally efficient R-loop removal as WT MCM8." Please add a citation to the relevant figure.

Reference

1. M. L. García-Rubio et al., The Fanconi Anemia Pathway Protects Genome Integrity from R-loops. *PLOS Genetics* 11, e1005674 (2015).
2. R. J. Hill, G. P. Crossan, DNA cross-link repair safeguards genomic stability during premeiotic germ cell development. *Nature Genetics* 51, 1283-1294 (2019).

Referee #2:

MCM8 and MCM9 form a hexameric helicase, which functions in homologous recombination (HR) in mitosis and meiosis. The manuscript by Wen et al. describes a novel role of MCM8 in R-loop resolution in primordial germ cells (PGCs). The authors initially observed a significant reduction of PGCs in MCM8-KO embryos (Fig. 1). PGCs lacking MCM8 exhibited proliferation defects during the S phase (Fig. 2). Mass spectrometry analysis revealed that MCM8 interacted with numerous RNA processing proteins (Fig. 3). The authors confirmed the interactions of MCM8 with DDX5 and DHX9, both of which are involved in R-loop resolution. PGCs and MEFs derived from MCM8-KO mice accumulated R-loops, as detected by the S9.6 antibody (Fig. 3F, G and Fig. 5A, B). This R-loop accumulation was further validated using CUT&TAG (Fig. 4). MCM8-KO MEFs displayed reduced fork speed due to the accumulation of R-loops (Fig. 5). MCM8 was identified within genes related to mitotic progression and was found to bind to R-loops (Fig. 6). Notably, in MCM8-KO MEFs, the association of DDX5 and DHX9 with R-loops was reduced. Finally, the authors determined that DDX5 interacted with the AAA+ core domain of MCM8. Moreover, the p.R309* and p.S492* mutations found in patients were unable to suppress R-loop accumulation in HEK293 MCM8-KO cells (Fig. 7).

The presented findings are interesting and potentially explain the reason why PGCs before entering meiosis were lost in MCM8-KO mice. However, one point should be clarified before consideration for publication.

It has been reported that MCM8 forms both MCM8/9 dimer and hexamer (Gambus and Blow, *Cell Cycle*, 2013; McKinzey et al., *NAR*, 2023; Weng et al. *eLife*, 2023; Acharya et al. *BioRxiv*, 2023). Furthermore, studies with MCM9-KO mice have demonstrated a decrease in PGCs (Hartford et al. *PNAS*, 2011; Luo and Schimenti, *Genesis*, 2015). However, the authors favour the idea that MCM8 functions in R-loop resolution without MCM9, as shown in Fig. 7F and discussed in Discussion. The authors should provide clarification regarding whether MCM9 is dispensable for R-loop resolution and whether MCM8 operates autonomously from MCM9.

Minor points

- Reconsider the figure order. Fig. 5 A, B should directly follow Fig. 3.
- Show control HEK293 WT cells in Figure 7D, E.
- HROB is required for MCM8/9 helicase activity and works with MCM8/9 on in ICL repair and HR (Hustedt et al., *G&D*, 2019; Huang et al. *Nat. Commun.*, 2020). This point should be mentioned in Introduction.

This manuscript reports phenotypes associated with knock out (KO) of the MCM8 DNA helicase in mouse primordial germ cells (PGCs) showing reduction of the pool of Primordial Germ Cells (PGCs) and generation of DNA damage, very likely DNA double strand breaks, resulting in infertility, confirming the original report (Lutzman et al., 2012). Authors also analyzed the global phosphorylation level of RNAPol II Ser2 and 5, as well as markers of epigenetic reprogramming in MCM8 KO, such as 5mC, H3K9me2 and H3K27me3, and found no significant differences, suggesting that MCM8 KO did not alter global transcription nor epigenetic reprogramming in PGCs. They next obtained the MCM8 protein interactome by MCM8 immunoprecipitation in P19 teratocarcinoma mouse cells and identified mostly proteins involved in RNA splicing processes. Amongst these proteins are proteins implicated in R-loop resolution, a DNA:RNA hybrid structure with a displaced single-stranded DNA that can be generated by conflicts between transcription and replication machinery, as well as a result of reduced mRNA processing (e.g.: splicing defects that slow down the transcription process). Because it was previously reported that PGCs display a high R-loops level, the authors explored whether compromised interaction between MCM8 and R-loops regulators may be responsible for the observed PGCs reduction and DNA damage upon MCM8 KO. For that, authors first confirmed physical interaction between MCM8 and two previously characterized R-loops regulators, the RNA helicases Ddx5 and Dhx9 in P19 as well as in PGCs. Then, they analyzed R-loop formation in either wild-type or MCM8 KO PGCs and show increased R-loops levels in MCM8 KO PGCs and mouse embryonic fibroblasts (MEFs). They further analyzed R-loops distribution genome-wide in MEFs and report a similar distribution between wild-type and MCM8 KO cells, as well as the appearance of additional R-loops compared to wild-type cells. Further, they asked whether R-loops accumulation is responsible for replication stress generated in MCM8 KO cells and for that authors measured replication fork progression in wild-type or MCM8 KO MEFs and report reduced replication fork speed which is somehow restored upon treatment with RnaseH1, an enzyme that degrades the RNA moiety of the R-loop. Finally, authors analyzed Ddx5 and Dhx9 retention at R-loops upon immunoprecipitation with the S9.6 antibody and show that their recovery is reduced in cells MCM8 KO. They conclude that MCM8 contributes to target Ddx5 and Dhx9 to R-loops and as such preserve genome integrity and PGCs proliferation.

Overall the study is well executed and reports interesting findings, in part confirming previous observation on the role of the MCM8 DNA helicase in DNA replication and germ cells metabolism. The observed interaction between MCM8 and Ddx5/Dhx9 is novel but somehow surprising. However, I found the claim that MCM8 KO increases the frequency of R-loops in PGCs needs to be strengthened. Also, I do not understand why the authors have not explored a potential role for MCM8 in regulating splicing, as strongly suggested by the MCM8 interactome, and they instead focused on a possible role of MCM8 on R-loops metabolism. Mutations that affect splicing indirectly generate replication stress and consequent R-loop increase, as originally shown (Paulsen et al., 2009 Mol Cell; PMID: 19647519; Gomez-Gonzalez et al., 2009 Mol Cell Biol; PMID: 19651896). Further, from data shown in Fig. 3B, the Numa1 protein appears to be the most abundant MCM8 interactor. Numa1 is a protein involved in the correct orientation of the mitotic spindle but it has also recently shown to localize to gene promoter regions (Ray et al., Nature 2022; PMID: 36171374). Numa1 depletion generates DNA damage and genomic instability, phenotypes similar to those observed in MCM8 KO cells, and consistent with the increase in the fraction of cells in the G2 phase of the cell cycle shown in Fig. 2F. Hence, I do not see the logic of focusing on Ddx5 and especially Dhx9, this latter being not the most abundant MCM8 interactor compared to at least two other Ddxs involved in R-loops resolution such as Ddx21 and Ddx41, unless authors have reasons to believe that these latter do not contribute to the genomic instability in PGCs upon MCM8 KO. Finally, although the authors claim a function for MCM8 for R-loops metabolism in PGCs, the critical data have been mostly obtained using P19, MEFs and HEK293T cells. Here below are comments that may help the authors to improve the manuscript.

1. Fig. 2A-B, the difference in the number of cells showing cleaved PARP-1 staining between wild-type and MCM8 KO is very weak. The quantification shown in panel B does not match the data shown in panel A. Same applies to Fig. 2E-F for Cyclin B1 and EdU. I do not understand how authors can identify G1 and G2 cells using EdU (S-phase) and Cyclin B1 (mitosis) staining. From these data one cannot say that cells are arrested in the cell cycle. If one believes these data, what appears is that there is an increase in the number of G2 cells, but this does not mean that cells are arrested. Checking activation of the DNA damage response (ATM and/or ATR, Chk1/Chk2) would have been more convincing.
2. In Fig. 2G-H authors show that MCM8 KO PGCs display a strong increase in the number of 53BP1 foci compared to wild-type PGCs and conclude that DNA damage accumulation leads to decreased PGCs proliferation. However, authors did not provide additional data to prove this statement. Should this be the case, then proliferation should be rescued by inhibiting the DNA damage response (probably the ATM pathway, not shown in the manuscript).
3. Fig. 3, it is indeed curious that the MCM8 interactome showed a great enrichment of RNA splicing factors. Why the authors have not explored a potential role of MCM8 in splicing? What about the best hit, Numa1? Did the authors identified MCM9 in the proteome, which is a stable MCM8 partner? I could not find the list of the proteins identified by mass spectrometry. A western blot and silver stain of the immunoprecipitated material must be shown.
4. Fig. 3C-E, 6F, 7B-C, do the western blot signals belong to the same gel? This is an important point in order to compare the relative signal intensity. In the immunoprecipitation experiments, can the authors show a protein that does not interact with MCM8? Can they immunoprecipitated MCM9?
5. Fig. 3F-G, it is important to include a treatment with RNaseH1 to show the specificity of the S9.6 signal. Also, the use of RNaseH1-dead mutant has been now shown to be a more reliable tool to detect R-loops than the S9.6 antibody by immunofluorescence (Crossley et al., JCB 2021; PMID: 34232287).
6. It would have been nice to have a DDX5 knock out (or downregulation) control aside both immunofluorescence and Chip data to compare with. Ideally, one wished to see how Ddx5 or Dhx9 affects R-loops distribution genome-wide and compare it with

that observed in MCM8 KO cells.

7. In Fig. 5C authors analyze replication dynamics by DNA fiber stretching and conclude that replication speed is slower in MCM8 KO cells, in line with previous reports demonstrating a function of MCM8 in contributing to replication fork progression (Maiorano et al., Cell 2005; Gambus et al., Cell Cycle 2013; Griffin et al., 2022). However, according to these data, it appears that MCM8 depletion reduced the length of both IdU and CldU tracks, suggesting an effect on both the initiation and elongation steps of DNA synthesis. However, I am not convinced that this is due to R-loops accumulation, as claimed by the authors. The statistical test here and elsewhere is inappropriate (see other points below). Significance must be recalculated using a Mann-Whitney test. Further, images shown in panel D are not representative of the quantification shown in panel E. Concerning the possible increase in R-loops upon MCM8 KO, this could be due to reduce DNA synthesis rate generating transcription-replication conflicts. Hence it is expected that inhibition of transcription should reduce R-loops formation, thus rescuing the effect of MCM8 KO. Fig. 5D-E, the reduction of 53BP1 foci in MCM8 KO cells treated with RNaseH1 is weak and quantification does not match the data.

8. Fig 6, a western blot showing the expression of Flag-MCM8 must be shown. Panel E, a control including RNaseH1 treatment and ChIP with a non-specific antibody (Flag alone as a control for Flag-MCM8 ChIP and non-specific for the S9.6 ChIP) must be included to ascertain the specificity of the immunoprecipitated material. Panel F, it is critical here to show that equal amounts of IgGs were recovered in all lanes, since based on this experiment authors conclude that Ddx5 and Dhx9 retention is reduced on R-loops upon MCM8 KO.

9. Fig. 7, authors complement R-loops generated by MCM8 KO in HEK293T cells by expressing either wild-type or diverse MCM8 mutants. Here again, the images provided in Fig. 7C do not match the quantification shown in panel E. The differences in S9.6 signals observed in these images are very weak.

10. Fig. EV6, a quantification of the Ddx5 and Dhx9 immunofluorescence signals obtained must be shown.

Other points

1. Page numbers are missing

2. Introduction, second page, MCM2-7 protein are not degraded in S-phase, they are removed from chromatin by ongoing DNA synthesis (Todorov et al., J. Cell Biol. 1995). Next page, Ddx5 and Dhx9 are not "the" R-loops removing factors but are R-loops-removing factors. A general introduction to factors known so far in R-loops resolution (including other Ddxs helicases) is missing.

3. Fig. 3F-G, it is unclear what the white arrows indicate (whether cytoplasmic R-loops or cells displaying nuclear R-loops); this has not been described in the figure legend.

4. Fig. 7F, what is the evidence that Ddx5 and Dhx9 function as dimers (or multimers) in R-loops resolution? It is entirely possible that resolution of a fraction of R-loops depends upon MCM8-Ddx5 and/or Dhx9 interaction. It is for this reason that it would have been nice to have a picture of R-loops distribution in Ddx or Dhx9 knock down cells.

5. A Mann-Whitney U test is more appropriate to determine significance in several experiments (Fig. 3G; 5B-C; 7E).

Dear Dr. Vodermaier,

Thank you very much for your consideration of our manuscript 'EMBOJ-2023-115128'. We appreciate the referees for their valuable comments on our work. The comments of the referees have been responded to point by point and all changes have been highlighted in blue color within the document.

Below you will find our point-by-point responses to the referees' comments.

Referee #1

Major concerns:

1. In figures showing immunofluorescence (IF) staining data (Fig 1A, B, 2A, 2C, 2E, EV4A, etc), the authors should provide more magnified, clear images so that the readers can evaluate the authors' statements more precisely.

Response: Thank you for your suggestion. We are sorry for the compressed IF images with poor resolution in the initially submitted manuscript. All of our original IF images are clear and of high resolution. We have provided more clear images (Fig 1A, 1B, 2A, EV2A, EV2B) and more magnified images (Fig 2A, EV2A, EV2B) in the revised manuscript.

2. In Page 6, the authors stated that "Compared to PGCs in WT littermates and single mutants of Mcm8 KO or ubiquitination-defective Fancd2, significantly fewer PGCs were detected in the double mutants (1467.0 ± 307.4 vs. 436.0 ± 130.6 vs. 300.6 ± 117.8 vs. 43.2 ± 6.4 , $P < 0.001$), suggesting that MCM8 has functions in PGC development that are independent of the FA pathway (Fig. EV4B)." According to Hill and Crossan, 2019, the DNA damage response (DDR) pathway is important in PGCs, especially in the context of the FA pathway. Furthermore, the FA pathway is also crucial for R-loop resolution (García-Rubio et al., 2015). Given these findings, which DDR pathway do the authors believe the results related to R-loop accumulation in Mcm8 KO are more closely associated with?

Response: Thank you for your comments. We have reviewed the DNA repair proteins involved in the suppression and resolution of R-loops. Besides the FA pathway, several DNA repair proteins participate in the regulation of R-loop homeostasis, including DNA-damage checkpoint proteins such as ATR and ATM, and homologous recombination (HR) proteins such as BRCA1, BRCA2 and RAD51, etc. Generally, they suppress or remove R-loops by promoting the recruitment of the RNA-DNA helicases. For instance, while activation of ATR and ATM promotes the recruitment of the RNA-DNA helicase SETX or DDX19 to unwind R-loops (Yüce Ö and West SC. *Mol Cell Biol.* 2013. PMID: 23149945; Hodroj D, et al. *EMBO J.* 2017. PMID:

28314779), BRCA2 recruits RNA-DNA helicase DDX5 to resolve R-loops (*Sessa G, et al. EMBO J. 2021. PMID: 33634895*). In this study, we firstly reveal that MCM8 also recruits DDX5 to resolve R-loops and it regulates PGC development that is independent of the FA pathway. Because MCM8 is well known for its involvement in homologous recombination (HR), it is assumed that its role in R-loop resolution may be associated with the HR pathway. However, it needs to be determined in the future study.

3. In Page 7, the authors stated that "MCM8 deficiency resulted in a higher proportion of apoptosis when compared to the WT group ($1.05 \pm 0.42\%$ vs. $0.24 \pm 0.14\%$, $P < 0.05$), but the number of apoptotic PGCs was less than 10 per embryo (Fig. 2A and B)." Why then does the number of PGCs decrease so dramatically in *Mcm8* KO mice? Response: Both excessive apoptosis and proliferation defects will result in PGC loss. During a limited time window between E9.5 and E13.5 in mice, PGCs undergo very rapid proliferation to expand the germ cell pool. Given that the apoptotic PGCs were few (less than 10 per embryo) at E11.5 in *Mcm8*^{-/-} embryos, PGCs at S-phase were decreased and PGCs at G2-phase were increased significantly, indicating that proliferation defect resulted from abnormal cell cycle was the leading cause for dramatic PGC loss. We have revised the statement in the manuscript.

4. In Page 8, the authors stated that "We removed 25 ribosomal proteins that were always found among the nonspecific binding proteins in the IP-MS assay and analyzed the remaining 84 binding proteins. We noted that the MCM8 interactors were mainly involved in RNA splicing processes, as indicated by Gene Ontology (GO) analysis (Fig. 3A)." The GO term "RNA splicing processes" is also a broad term, and many proteins related to this process are expected to be abundant in the nucleus, similar to "ribosomal proteins." This reviewer is concerned that this observation may also be attributed to noise. Therefore, conducting a comparative analysis for MCM8-IP in cells less abundant for MCM8 as a negative control may further support the authors' claim.

Response: Thank you for your suggestion. It was reported that ribosomal proteins including ribosomal protein large/small subunits (RPL/RPS) were among the most common contaminating proteins in the antibody-based affinity purification–mass spectrometry experiments (*Mellacheruvu D, et al. Nat Methods. 2013. PMID: 23921808; Mohammed H, et al. Nat Protoc. 2016. PMID: 26797456*). Accordingly, we have excluded RPL/RPS proteins from our IP-MS data in which 13.7% (14/102) and 19.6% (37/189) purified binders were removed from the IgG interactome and MCM8 interactome, respectively. According to your suggestion, we have included an analysis for MCM8-IP in MCM8 KO HEK293 cells (Fig EV3D) as a negative control

in the revised manuscript. The specificity of MCM8's interactions with DDX5 and DHX9 has been confirmed, thereby ruling out the possibility of mere noise.

5. Fig 3B: Related to the above point, this analysis should be presented alongside an appropriate negative control to confirm that the "MCM8 interactome" significantly overlaps with "Human and mouse R-loop regulators."

Response: Thank you for your suggestion. We tried to downregulate MCM8 in P19 cells by utilizing siRNA, shRNA adenovirus and antisense oligonucleotides (LNA-GapmeRs), however, none of them could downregulate MCM8 efficiently. Alternatively, we have conducted a comparative analysis for MCM8-immunoprecipitated proteins in MCM8 KO HEK293 cells as a negative control (Fig EV3D) and confirmed the endogenous interactions between MCM8 and DDX5, as well as DHX9, which further verified that MCM8 significantly interacted with R-loop regulators reported in the previous study (*Lin, et al. Nucleic Acids Res. 2022. PMID:34792163*).

6. Fig 3F, 5A, 7D: I do not think it really possible to make quantitative comparison of the IF signals between different specimens with the method employed by the authors, as the staining levels and intensities can easily be changed and very difficult to control between different samples. To overcome this point, for example, the authors should place the cells with two different genotypes on the same slide and then compare the signals. Why were S9.6 signals so strong only in single puncta in the nuclei?

Response: Thank you for your suggestion. At present, it is a common approach to evaluate the R-loop levels between different specimens by quantitative comparison of the IF signals of the S9.6 antibody, which is known to recognize DNA-RNA hybrids (*Bhatia V, et al. Nature. 2014. PMID: 24896180; Wiedemann EM, et al. Cell Rep. 2016. PMID: 27974207*). To minimize the variability of staining levels and intensities between samples on different slides, we rigorously controlled the experimental conditions, including pairing WT and KO embryos from the same pregnant mice, seeding WT and KO cells simultaneously, fixing samples with the same duration, and using the same exposure intensity and time when capturing the images. Therefore, increased IF signals of S9.6 in our study could provide supportive evidence for R-loop accumulation in MCM8-deficient cells. To give a more convincing conclusion, we have used a dot blot assay with the S9.6 antibody to quantify R-loop levels in WT and KO cells, with DDX5-downregulated cells as a positive control (Fig 4C, 4D) in the revised manuscript.

Besides common R-loops in the genome, the S9.6 antibody also recognizes DNA-RNA hybrids within the mitochondria and the nucleoli. The single puncta strong signals in the nuclei are generated by the abundant rRNA-derived DNA-RNA hybrids in the nucleoli (Shen W, et al. *Nucleic Acids Res.* 2017. PMID: 28977560). In this study, we also detected the nucleolus with an anti-Fibrillarin antibody (Fig 4A), which was consistent with the previous studies (Mersaoui SY, et al. *EMBO J.* 2019. PMID: 31267554; Gong D, et al. *Stem Cell Reports.* 2023. PMID: 36931280).

7. Fig 4B and Fig 4C: Judging from the pattern observed in the "non-Antibody" track in Fig 4B, the signal-to-noise (S/N) ratio of the IP tracks may not be optimal. Therefore, in this case, it is crucial to mention in detail in the "CUT&Tag assay" section that peak calling was performed using the "non-Antibody" track as the input track in MACS2.

Response: Thank you for your comment. Compared to ChIP-seq datasets, CUT&Tag profiles have extremely low background noise levels (Kaya-Okur HS, et al. *Nat Commun.* 2019. PMID: 31036827), which is consistent with the CUT&Tag profiles of the S9.6 antibody in our samples (Fig 5) in the revised manuscript. It has been reported that for large genomes the reads in the no-antibody negative control are too sparsely distributed to be useful for data analysis (Kaya-Okur HS, et al. *Nat Protoc.* 2020. PMID: 32913232). Consistently, there was no significant enrichment in our "no-antibody" group. Therefore, we performed peak calling without using the "no-antibody" track as the input track. However, it was inappropriate to show the genomic distribution of signals in the "no-antibody" (NC) group in the previous manuscript. Accordingly, we have amended this image (Fig 5B) to heatmap plots of genes with R-loop signals across the 5 kb window around the transcription start site (TSS), and added detailed information in the "CUT&Tag assay" section in the revised manuscript.

8. Fig 4C-E: This reviewer wonders why the peaks in the WT condition are not entirely encompassed by those in the KO condition. Given the authors' primary claim that the KO condition leads to more frequent R-loops, one would expect the KO to

exhibit additional peaks in addition to those found in the WT. However, Fig 4C shows many unique peaks in the WT. What are the properties of these peaks specific to WT cells?

Response: As the reviewer concerned, only 24% of R-loop peaks were overlapped between WT and KO MEFs, and more unique peaks in the KO cells were found. These results were consistent across three independently repeated experiments, and we attempt to provide a reasonable explanation for these interesting results based on the current knowledge of R-loop regulation. Usually, transcription and replication in the nucleus are coordinated spatially and temporally to minimize conflict between the two processes in the higher eukaryotes (*Hamperl & Cimprich. Cell. 2016. PMID: 27912056*). On the one hand, defective transcription termination induces a redistribution of replication initiation factors (MCM2-7 complexes) in the G1-phase (*Gros J, et al. Mol Cell. 2015. PMID: 26656162*). On the other hand, dysregulated origin firing increases R-loop formation (*Hamperl S, et al. Cell. 2017. PMID: 28802045; Lang KS, et al. Cell. 2017. PMID: 28802046*). Since MCM8 depletion in human cells reduced chromatin loading of replication initiation MCM2-7 complexes (*Volkening M, et al. Mol Cell Biol. 2005. PMID: 15684404*), we speculated that a large number of unscheduled R-loop signals (KO unique peaks) and loss of shared signals with WT in KO cells were relevant to the repositioned RNA polymerases or the changed distribution of replication origins. We have added these contents in the discussion section.

9, In Page 9: "actively proliferating PGCs (EGFP positive) at E12.5..." The authors should explain the marker they used.

Response: We apologize for the inaccurate statement. This sentence should be "actively proliferating PGCs (STELLA-EGFP positive) at E12.5...". STELLA is used as a marker of PGC, and we have successfully constructed STELLA-EGFP reporter mice to isolate and enrich PGCs by fluorescence-activated cell sorting (*Yang Y, et al. Proc Natl Acad Sci U S A. 2022. PMID: 35969748*).

10. In Page 9: ".....as indicated by the normal transcriptional upregulation of KO PGCs when compared to WT PGCs (Fig. EV5A and B)." This reviewer does not think that to draw this conclusion is not really possible only with these ambiguous IF images. The authors should perform some different, more quantitative experiment (e.g., RNA-seq experiment with appropriate spike-ins) to make this conclusion.

Response: Thanks for your valuable suggestion. It was not precise to conclude normal transcriptional upregulation in KO PGCs based on no obvious differences found in global IF signals between WT and KO PGCs. We agree that it is more accurate to quantitatively analyze the global transcription activity by RNA-seq. However, the

very small number of remaining PGCs in E11.5 KO embryos makes it very difficult to collect them for RNA-seq using a sorting-based method. To give more convincing evidence, we have applied the 5-ethynyl uridine (EU) incorporation assay (Fig EV2C) to quantify the transcriptional activity by directly imaging nascent RNA. Meanwhile, quantitative analysis of phosphorylation on Ser5 and Ser2 of RNA Pol II indicating the transcription initiation and elongation respectively have been added (Fig EV2C) in the revised manuscript.

11. In Page 10, the authors stated that "Two genes, *Chchd7* and *Armc10* are shown as examples (Fig. 4F). The biological processes of these two genes with increased R-loop signals were enriched in the positive regulation of developmental growth according to the GO analysis (Fig. 4G)." Here, the meaning of enrichment in developmental growth appears unclear to this reviewer, at least in its current context. Is this term specifically related to the mitotic proliferation of PGCs? It would be beneficial if the authors could clarify the significance of this term. Additionally, the rationale for selecting these two genes is not clearly explained.

Response: In the first sentence, *Chchd7* and *Armc10* are the representative genes with snapshots of R-loop signals for the subsets with shared peaks in Fig. 5D and the KO-only peaks in Fig. 5E of the revised manuscript, respectively. *Chchd7* was associated

with growth including height, reproduction and muscle formation (*Lettre G, et al. Nat Genet. 2008. PMID: 18391950*; *Mota LFM, et al. Genomics. 2022. PMID: 35671870*), and *Armc10* was related to axon regeneration and neurite outgrowth (*Xie L, et al. Sci Transl Med. 2023. PMID: 37556559*) in previous studies. For the second sentence, we apologize for the wrong statement of “The biological processes of these two genes”, which should be “The biological processes of these genes”. We conducted the GO analysis of all the genes with increased R-loop signals (including the increased signals in shared peaks and the signals in KO-only peaks) in KO MEFs and we found an enrichment of genes related to positive regulation of developmental growth. The R-loop accumulation may impair the expression of these genes and thus the PGC proliferation. The most crucial point we determined by the CUT&Tag assay was the accumulated R-loops after MCM8 deficiency, which led to genome instability and impaired proliferation of PGCs.

12. Fig 5A and D: Is GFP-RNH1 fusion protein? If so, why does the majority of the proteins locate in the cytoplasm?

Response: Yes, GFP-RNH1 is a fusion protein. The longest transcript of RNH1 that encoded two isoforms due to the use of alternative translation initiation codons was overexpressed in MEFs. The longer isoform derived from the upstream start codon is a mitochondrial protein, whereas the shorter isoform derived from the downstream start codon is a nuclear protein. In this study, the nuclear isoform was capable of removing the nuclear DNA-RNA hybrids, as indicated by the decrease of R-loop levels in cells upon overexpressing GFP-RNH1 (Fig. 4A and 4B in the revised manuscript). The reason for the RNH1-GFP showing high signals in the cytoplasm is that the longer isoform is enriched in mitochondria as reported previously (*Shen W, et al. Nucleic Acids Res. 2017. PMID: 28977560*).

13. In Page 11, the authors stated that "We found that 16.9% (1657/9799) of the MCM8 binding sites co-occurred with R-loop signals (Fig. 6C), with the majority (81.3%, 1347/1657) being located in the promoter regions." From this reviewer's perspective, the observed percentage of 16.9% appears to be quite low to suggest a direct link between MCM8 and R-loop resolution. Could it be that MCM8 is only responsible for a specific subset of R-loops? It would be valuable for the authors to discuss the possible reasons behind this observation or provide additional evidence to support their findings in the text.

Response: Thank you for your insightful question. MCM8 has long been well-known as a DNA helicase to drive replication fork elongation and promote homologous repair. Here, we uncovered a previously unappreciated role of MCM8 in R-loop resolution. Two possible scenarios may explain the results of a low percentage of the MCM8

binding sites being associated with R-loop signals. First, the immunoprecipitation sequences encompass all MCM8-binding sites involved in either replication fork elongation, HR or R-loop resolution in the CUT&Tag assay; second, co-transcriptional R-loops generated across all the cell cycle phases have been detected all over the genome and their levels are associated with the global transcriptional activity. Given that MCM8 predominantly functions in the S-phase (Gozuacik D, et al. *Nucleic Acids Res.* 2003. PMID: 12527764; Maiorano D, et al. *Cell.* 2005. PMID: 15707891) and prefers binding to R-loops in the promoter regions, MCM8 may be only responsible for resolving a specific subset of S-phase R-loops. Thus, it is not surprising to find that only part of the MCM8 binding sites relate to R-loops. These contents have been added to the discussion section.

14. In Page 11, the authors stated that "R-loop levels are positively associated with transcriptional activity (Wahba et al, 2016)." Based on the authors' data, is it also observed that genes with higher transcriptional activities exhibit more pronounced differences between the KO and WT conditions? If so, this could serve as further confirmation of the data quality.

Response: Thanks for your suggestion. Based on our currently available data, we have analyzed the distribution of R-loop signals of all the expressed genes between the KO and WT. The heatmaps (in Fig EV5A) were shown by sorting the R-loop signals based on gene expression level obtained from the RNA sequencing analysis of MEFs from high to low, which suggested that loss of MCM8 in MEFs led to a pronounced accumulation of R-loops in promoters of active genes.

15. Fig 6H and I: As in other IF figure panels, the data are hardly visible in H and the very minor difference shown in I is not really convincing.

Response: Thanks for your comment. We have optimized the experimental conditions and repeated the experiments in Fig 6H and I.

16. In Page 12 and Fig 7, it would be better for the authors to define what p. R309* and p.S492* stand for, for the clarity of the readers.

Response: Thanks for your suggestion. Both p.R309* and p.S492* identified in POI patients were truncated proteins from mutations c.925C>T and c.1475C>A (NM_032485) that led to the premature termination of translation at Arginine 309 and Serine 492, respectively (*Heddar A, et al. J Clin Endocrinol Metab. 2020. PMID: 32242235; Wang F, et al. Mol Genet Genomic Med. 2020. PMID: 32652893*). We have supplemented the information of the two mutations in the revised manuscript.

17. What might be the phenotypes in germ cells in Ddx5 or Dhx9 knockouts? This information is useful to include for discussion to understand a broader picture of their relevance in germ cells.

Response: Thanks for your comment. Lack of DDX5 or DHX9 both lead to embryonic lethality. The Mouse Genome Informatics (MGI) database reports that *Ddx5* null mice die around E11.5 displaying blood vessel malformations and *Dhx9* knockouts die in embryonic stages with massive apoptotic cells in embryonic ectodermal cells. Thus, a conditional deleting strategy will help elucidate the independent roles of these two factors in germ cell development. Interestingly, recent studies demonstrated that DDX5 played essential transcriptional and post-transcriptional roles in the maintenance and function of spermatogonia (*Legrand JMD. Nat Commun. 2019. PMID: 31123254; Xia Q, et al. Cell Prolif. 2021. PMID: 33666296*). Collectively, both MCM8 and DDX5 play crucial roles in the development of germ cells at different stages. We have included these contents in the discussion section.

18. In Page 14, the authors stated that "...actively dividing PGCs were susceptible to MCM8 deficiency; therefore, it was hypothesized that mitotic germ cells are more dependent on MCM8 to resolve specific replication stress, thus safeguarding genome stability. Recently, we reported that rapidly proliferating PGCs are faced with constitutive replication stress, including high levels of R-loops and frequent RF stalling (Yang et al., 2022)." There are many actively dividing cell types during development and therefore, further discussion may be useful why the authors think that PGCs are specifically susceptible to MCM8 loss and R-loop mis-resolution.

Response: Although active proliferation occurs in both PGCs and somatic stem cells in embryos, their characteristics are distinct. In mouse embryos, rapid proliferation of PGCs is limited to the period between E9.5 and E13.5, and the maximum number of PGCs lays the foundation of the reproductive reserve of both sexes. Of note, because oocytes are not renewable, germ cell loss due to PGC development defects cannot be mitigated or rescued in females in later life. However, a subset of the somatic stem

cells expands to renew and maintain the stem cell pools during their lifetime. Although reduced proliferation affects the establishment of the initial stem cell pool, the remaining somatic stem cells can maintain the pool through compensatory proliferation to mitigate the adverse effects on the individual.

Our previous study has shown that increased R-loop levels originating from global upregulation of transcription output is a crucial source of endogenous DNA threats in rapidly proliferating PGCs, and this feature underlies why PGCs are more dependent on the FA pathway to safeguard their genome stability (Yang Y, et al. *Proc Natl Acad Sci U S A*. 2022. PMID: 35969748). The specific loss of germ cells was also observed in other ubiquitously expressed DNA repair gene knockout mouse models, such as *Mcm9* and *Rad54* (Luo Y, et al. *Genesis*. 2015. PMID: 26388201; Messiaen S, et al. *Cell Death Dis*. 2013. PMID: 23949223). These findings suggest that PGCs could be considered as a group of special cells that are more dependent on DNA repair factors to respond to endogenous genome threats, thus maintaining a high level of genome stability. When DNA damage occurs, they will be more sensitive and thus initiate more strict monitoring mechanisms, such as p53 pathway activation and cell cycle regulation to repair the DNA damage, thus resulting in the severe loss of PGCs. In this scenario, we showed that PGCs lacking MCM8 were more susceptible to R-loop accumulation-induced DNA damage, and their impaired proliferation caused an insufficient reproductive reserve. We have expanded our discussion to address this issue in the revised manuscript.

Minor points:

1. In Page 7, the authors stated that "Because PGCs are sensitive to DNA damage, we reasoned that PGCs had an increased requirement for MCM8 in genome stability maintenance under physiological conditions and that unrepaired DNA damage would result in apoptosis or cell cycle arrest in MCM8-deficient PGCs." In this sentence, the authors should consider including a citation to support the statement that "PGCs are sensitive to DNA damage."

Response: Thanks for your suggestion. There are several previous studies supporting the statement that "PGCs are sensitive to DNA damage (Hill RJ, Crossan GP. *Nat Genet*. 2019. PMID: 31367016; Luo Y, et al. *PLoS Genet*. 2014. PMID: 25010009)". We have added these citations in the revised manuscript.

2. In Page 8, the authors stated that "We removed 25 ribosomal proteins that were always found among the nonspecific binding proteins in the IP-MS assay and analyzed the remaining 84 binding proteins." Is there any evidence or citation to validate the exclusion of ribosomal proteins in this IP-MS analysis?

Response: Thanks for your suggestion. As we mentioned in question 4 above, there is

evidence showing nonspecific binding of ribosomal proteins in the immunoprecipitation assay (Mellacheruvu D, et al. *Nat Methods*. 2013. PMID: 23921808; Mohammed H, et al. *Nat Protoc*. 2016. PMID: 26797456). We have revised the statement and included citations in the revised manuscript.

3. In Page 10, "... and KO embryos (Fig. 4A)" should be "KO MEFs."

Response: We apologize for this mistake. The "embryos" should be changed to "MEFs.". We have corrected this sentence.

4. In Page 13, the authors stated that "Furthermore, overexpression of WT MCM8 in KO HEK293 cells showed a drastic decrease in R-loop levels compared to KO cells." Please add a citation to the relevant figure.

Response: Thanks for your suggestion. Fig. 7D and 7E were relevant to this sentence, and we have added the citation.

5. In Page 17, the authors stated that "In addition, we also showed that p.E341K (rs16991615), which had normal interaction with DDX5, did not show equally efficient R-loop removal as WT MCM8." Please add a citation to the relevant figure.

Response: Thanks for your suggestion. Fig. 7C-E were relevant to this sentence, and we have revised our statements and added the relevant citations.

Referee #2

It has been reported that MCM8 forms both MCM8/9 dimer and hexamer (Gambus and Blow, *Cell Cycle*, 2013; McKinzey et al., *NAR*, 2023; Weng et al. *eLife*, 2023; Acharya et al. *BioRxiv*, 2023). Furthermore, studies with MCM9-KO mice have demonstrated a decrease in PGCs (Hartford et al. *PNAS*, 2011; Luo and Schimenti, *Genesis*, 2015). However, the authors favour the idea that MCM8 functions in R-loop resolution without MCM9, as shown in Fig. 7F and discussed in Discussion. The authors should provide clarification regarding whether MCM9 is dispensable for R-loop resolution and whether MCM8 operates autonomously from MCM9.

Response: Thank you for your insightful question. There is a strong interdependence between MCM8 and MCM9 because they stabilize each other by forming the MCM8/9 complex which not only participates in DNA synthesis but also exerts a variety of biological functions to orchestrate HR repair (Nishimura K, et al. *Mol Cell*. 2012. PMID: 22771115; Lee KY, et al. *Nat Commun*. 2015. PMID: 26215093; Park J, et al. *Mol Cell Biol*. 2013. PMID: 23401855). But deficiency of each of them didn't generate identical phenotypes among cells or mice even in the same background (Nishimura K, et al. *Mol Cell*. 2012. PMID: 22771115; Lutzmann M, et al. *Mol Cell*.

2012. PMID: 227711201). For example, *Mcm8*-null male mice were devoid of post-meiotic cells and were completely sterile, but *Mcm9*-null males produced functional spermatozoa and were fertile (*Lutzmann M, et al. Mol Cell. 2012. PMID: 22771120*), suggesting that their functions exhibited differences. As you mentioned, in order to determine the probability of MCM9 interaction with MCM8 during the R-loop resolution process, we performed the co-immunoprecipitation (co-IP) assay using an anti-MCM9 antibody in P19 cells and discovered that MCM9 not only interacted with MCM8 but also interacted with R-loop regulator DDX5 and DHX9 (Fig EV3E), indicating the probability of a functional MCM8/9 complex in the R-loop resolution process. We have amended our statements and working model in the revised manuscript.

Minor points

- Reconsider the figure order. Fig. 5 A, B should directly follow Fig. 3.

Response: Thanks for your suggestion. We have changed the figure order to make the manuscript more readable.

- Show control HEK293 WT cells in Figure 7D, E.

Response: Thanks for your suggestion. We have re-performed the immunofluorescence staining of R-loops and show control WT HEK293 cells alongside MCM8 KO HEK293 cells expressing either wild-type or mutant MCM8 (Fig 7D, 7E) in the revised manuscript.

- HROB is required for MCM8/9 helicase activity and works with MCM8/9 on in ICL repair and HR (Hustedt et al., G&D, 2019; Huang et al. Nat. Commun., 2020). This point should be mentioned in Introduction.

Response: Thanks for your suggestion. We have added the introduction of HROB and its function related to MCM8/9 in the Introduction section.

Referee #3

Overall the study is well executed and reports interesting findings, in part confirming previous observation on the role of the MCM8 DNA helicase in DNA replication and germ cells metabolism. The observed interaction between MCM8 and Ddx5/Dhx9 is novel but somehow surprising. However, I found the claim that MCM8 KO increases the frequency of R-loops in PGCs needs to be strengthened. Also, I do not understand why the authors have not explored a potential role for MCM8 in regulating splicing, as strongly suggested by the MCM8 interactome, and they instead focused on a possible role of MCM8 on R-loops metabolism. Mutations that affect splicing

indirectly generate replication stress and consequent R-loop increase, as originally shown (Paulsen et al., 2009 Mol Cell; PMID: 19647519; Gomez-Gonzalez et al., 2009 Mol Cell Biol; PMID: 19651896). Further, from data shown in Fig. 3B, the Numa1 protein appears to be the most abundant MCM8 interactor. Numa1 is a protein involved in the correct orientation of the mitotic spindle but it has also recently shown to localize to gene promoter regions (Ray et al., Nature 2022; PMID: 36171374). Numa1 depletion generates DNA damage and genomic instability, phenotypes similar to those observed in MCM8 KO cells, and consistent with the increase in the fraction of cells in the G2 phase of the cell cycle shown in Fig. 2F. Hence, I do not see the logic of focusing on Ddx5 and especially Dhx9, this latter being not the most abundant MCM8 interactor compared to at least two other Ddxs involved in R-loops resolution such as Ddx21 and Ddx41, unless authors have reasons to believe that these latter do not contribute to the genomic instability in PGCs upon MCM8 KO. Finally, although the authors claim a function for MCM8 for R-loops metabolism in PGCs, the critical data have been mostly obtained using P19, MEFs and HEK293T cells. Here below are comments that may help the authors to improve the manuscript.

Response: Thank you for your insightful questions. MCM8 is a well-known factor involved in DNA damage repair and genome stability for mitotic cells (*Lee KY, et al. Nat Commun. 2015. PMID: 26215093; Lutzmann M, et al. Mol Cell. 2012. PMID: 22771120; Nishimura K, et al. Mol Cell. 2012. PMID: 22771115*). The maintenance of genome stability is essential for both mitotic germ cells and somatic cells. Because germ cells with a low mutation rate have super-stable genomes, they must rely on more robust DNA damage response (DDR) mechanisms to ensure their genome stability. Recently, specific loss of PGCs in ubiquitously expressed DDR gene knockout mouse models was attributed to a more pronounced accumulation of endogenous DNA damage, indicating that unique origins of endogenous genome threats may underpin higher demands for these DDR factors during PGC development (*Hill RJ, Crossan GP. Nat Genet. 2019. PMID: 31367016; Yang Y, et al. Proc Natl Acad Sci U S A. 2022. PMID: 35969748*).

Our recent study has shown that increased R-loop levels originating from global upregulation of transcription is a crucial source of endogenous DNA damage in rapidly proliferating PGCs, thus necessitating the functional FA pathway to safeguard genome stability and proliferation of PGCs (*Yang Y, et al. Proc Natl Acad Sci U S A. 2022. PMID: 35969748*). Interestingly, *Mcm8* KO mice also displayed profound PGC loss, without obvious abnormalities of somatic systems. Therefore, we focus on elucidating the role of MCM8 in the resolution of endogenous DNA damage in PGCs. Then, we found that MCM8 may be related to R-loop regulation through its interactors from IP-MS. Later, we verified the increase of R-loops after MCM8 deficiency in PGCs and confirmed the interaction of MCM8 with R-loop-resolving

factors DDX5 and DHX9 in sorted WT PGCs. However, limited by the scarcity of the KO PGC population, we predominantly conducted functional experiments in P19, MEF, and HEK293 cells, and these consistent results indicated that MCM8 was capable of suppressing R-loops, a function not restricted to germ cells, but also universal in somatic cells. Because of the increased requirement for preserving genome stability, PGCs lacking MCM8 were more susceptible to R-loop-induced DNA damage, showing dramatically decreased proliferation.

As you mentioned, there may be a potential involvement of MCM8 in regulating RNA splicing. Therefore, we further performed RNA-seq in WT/KO MEFs and analyzed the alternative splicing events. There were few mRNA splicing changes identified and the read mapping distribution was not obviously changed after MCM8 deficiency (Figs EV4E and EV4F), suggesting that MCM8 may not play an essential role in RNA splicing. These results were included in the revised manuscript (Fig EV4).

Among the R-loop regulators collected in the database (*Lin R, et al. Nucleic Acids Res. 2022. PMID: 34792163*), some are validated as the R-loop binding proteins by biological experiments, whereas a large proportion of them including NuMA are obtained from high-throughput screening studies (*Cristini A, et al. Cell Rep. 2018. PMID: 29742442*; *Wu T, et al. Mol Cell Proteomics. 2021. PMID: 34478875*; *Mosler T, et al. Nat Commun. 2021. PMID: 34916496*), but lack verification. Recently, it has been reported that a fraction of NuMA in the interphase nucleus promotes gene transcription by limiting the polyADP-ribosylation of RNA polymerase II and repairs oxidative DNA breaks (*Ray S, et al. Nature. 2022. PMID: 36171374*). In that study, NuMA depletion increased genome instability resulting from oxidative DNA breaks at regulatory genome elements during transcription activation. Therefore, the role of NuMA in transcription activation and DNA repair indicates potential reciprocal regulation between NuMA and MCM8, which needs further exploration.

In this study, we found that many interactors were related to R-loop regulation in MCM8 interactome, including DDX5, DHX9, DDX21 and DDX41 (*Lin R, et al. Nucleic Acids Res. 2022. PMID: 34792163*). For DDX5 and DHX9, multiple lines of evidence from experimental studies have verified that they repress R-loops by unwinding the DNA-RNA hybrids in both human and mouse cells, and their R-loop-resolving activities are regulated by multiple co-factors (*Sessa G, et al. EMBO J. 2021. PMID: 33634895*; *Yuan W, et al. Nucleic Acids Res. 2021. PMID: 34329467*). For DDX21 and DDX41, they have not been demonstrated in R-loop resolution in mouse cells. Therefore, DDX5 and DHX9 were selected as representative factors for verification in our study. Yet, it cannot be ruled out that MCM8 also suppresses R-loops by interacting with DDX21, DDX41 or other factors.

1. Fig. 2A-B, the difference in the number of cells showing cleaved PARP-1 staining between wild-type and MCM8 KO is very weak. The quantification shown in panel B does not match the data shown in panel A. Same applies to Fig. 2E-F for Cyclin B1 and EdU. I do not understand how authors can identify G1 and G2 cells using EdU (S-phase) and Cyclin B1 (mitosis) staining. From these data one cannot say that cells are arrested in the cell cycle. If one believes these data, what appears is that there is an increase in the number of G2 cells, but this does not mean that cells are arrested. Checking activation of the DNA damage response (ATM and/or ATR, Chk1/Chk2) would have been more convincing.

Response: Thank you for your comments. During the evaluation of PGC apoptosis, we also performed the cleaved PARP1 immunofluorescence staining in a positive control group from pregnant mice treated with an intraperitoneal injection of mitomycin C (MMC). After MMC treatment, more cleaved PARP1 positive PGCs were detected in E11.5 genital ridges. However, the apoptotic PGCs in the MMC-treated group were also few (seeing the supplementary figure below), which is consistent with the results in MCM8-deficient PGCs. Usually, only 1~2 apoptotic PGCs were found per genital ridge section, and the absolute number was fewer than 10 per *Mcm8*^{-/-} embryo. Therefore, only the representative images with positive cleaved PARP1 staining were selected to be displayed in the originally submitted manuscript. We have provided a representative image without apoptotic PGCs for the WT group to match the statistical data (Fig EV2A).

Supplementary figures of MMC treatment

For cell cycle phase determination, we applied immunofluorescence staining of Cyclin B1 combined with EdU incorporation assay which is considered to be a reliable method for cell cycle analysis, especially when only a small amount of target cells could be available (*Eastman AE, et al. FEBS Lett. 2020. PMID: 32441778*). Cyclins are proteins that are distinguished by their steady accumulation in interphase followed by specific and rapid proteolysis at mitosis (*Pines J. Cell Growth Differ. 1991. PMID: 1648379*). Cyclin B1 that accumulates in G2-phase is predominantly cytoplasmic until just before mitosis, and then it translocates into the nucleus at the beginning of mitosis (*Pines J, et al. J Cell Biol. 1991. PMID: 1717476; Hagting A, et al. Curr Biol. 1999. PMID: 10395539*). Therefore, the G2-phase cells can be identified by the presence of high levels of Cyclin B1 in the cytoplasm (*Seki Y, et al. Development. 2007. PMID: 17567665*), whereas accumulated Cyclin B1 in the nucleus indicates the M-phase cells and the G1-phase cells are characterized by negative staining of both Cyclin B1 and EdU. Our results showed that the proportion of S-phase PGCs decreased and G2-phase PGCs increased, suggesting an abnormal cell cycle which was consistent with the longer doubling time of KO PGCs calculated from our data (shown in the table below). Accordingly, we have revised the statements of the cell cycle analysis.

According to your suggestion, we have performed immunofluorescence staining of pATM in PGCs to detect the DNA damage response to the DSB formation (Fig 2G).

Phenotype	PGC number			Doubling time (h)
	E8.5	E9.5	E11.5	
WT	100±10	254±62	1861±592	16.7
KO	95±9	110±26	471±169	22.9

Note: T doubling = $48/\log_2(N_{E11.5}/N_{E9.5})$ h

2. In Fig. 2G-H authors show that MCM8 KO PGCs display a strong increase in the number of 53BP1 foci compared to wild-type PGCs and conclude that DNA damage accumulation leads to decreased PGCs proliferation. However, authors did not provide additional data to prove this statement. Should this be the case, then proliferation should be rescued by inhibiting the DNA damage response (probably the ATM pathway, not shown in the manuscript).

Response: Thank you for your comments. We know that the DDR is initiated upon sensing damage, and then complex signal transduction is activated to promote DNA repair, including prolonging the cell cycle to allow more time for repair or eliminating cells containing severe unrepaired DNA damage by apoptosis (*Lanz MC, et al. EMBO J. 2019. PMID: 31393028; Blackford AN, et al. Mol Cell. 2017. PMID: 28622525*). In our study, decreased PGC proliferation in MCM8 KO embryos may be mainly due to

the prolonged cell cycle. Since effective DDR is essential for cell survival and proliferation, especially for PGCs, inhibition of the DDR pathway such as the ATM pathway may result in PGC death. Therefore, it may not be an efficient rescue strategy in this context. We have performed an additional experiment of γ H2AX foci detection and immunofluorescence staining of pATM to strengthen our statement that the DNA damage accumulates in MCM8 KO PGCs (Fig 2E and 2G).

3. Fig. 3, it is indeed curious that the MCM8 interactome showed a great enrichment of RNA splicing factors. Why the authors have not explored a potential role of MCM8 in splicing? What about the best hit, Numa1? Did the authors identified MCM9 in the proteome, which is a stable MCM8 partner? I could not find the list of the proteins identified by mass spectrometry. A western blot and silver stain of the immunoprecipitated material must be shown.

Response: Thank you for your insightful questions. As answered above, we have performed RNA-seq in WT/KO MEF and analyzed the alternative splicing events, which suggested that MCM8 may not play an essential role in RNA splicing.

NuMA is the top hit in our MCM8 interactome. As described above, the role of NuMA in DNA repair indicates potential reciprocal regulation between NuMA and MCM8, which needs further exploration.

MCM9 is a stable partner of MCM8, which has been confirmed by immunoprecipitation assay (*Nishimura K, et al. Mol Cell. 2012. PMID: 22771115*), and they form a complex to function in the replication and HR repair. However, MCM9 wasn't identified in the proteome, which was probably attributed to its low endogenous abundance under the physiological condition (*Beck M, et al. Mol Syst Biol. 2011. PMID: 22068332*). We have performed the co-IP assay to verify the interaction between MCM8 and MCM9 (Fig EV3B, EV3E).

We have also shown the original images of gels with Coomassie brilliant blue staining of the immunoprecipitated material (Fig EV3A) and added more details in the "IP assay" part of the Materials and methods. In detail, immunoprecipitated proteins from the IgG and MCM8 groups were loaded for electrophoresis and then Coomassie brilliant blue staining was performed. After washing the excessive dye, gel slices were cut off for subsequent mass spectrometry analysis. Therefore, we acquired all the potential interactors of MCM8 in P19 cells. The list of the proteins identified by mass spectrometry has been included in the supplementary data (Table EV1).

4. Fig. 3C-E, 6F, 7B-C, do the western blot signals belong to the same gel? This is an important point in order to compare the relative signal intensity. In the immunoprecipitation experiments, can the authors show a protein that does not interact with MCM8? Can they immunoprecipitated MCM9?

Response: Thank you for your comments. The signals in Fig. 3C-E were used to validate the interactions qualitatively, and they belonged to the same gel and were generated by stripping and re-incubating with desired antibodies. In Fig. 6F, 7B-C, all compared groups belonged to the same gel. However, the signals of the Input or IP samples were generated separately.

In the immunoprecipitation experiments, we did find proteins that did not interact with MCM8, such as PCNA (Fig EV3C). As we mentioned in question 3, we have performed the co-IP assay, which verified the interaction between MCM8 and MCM9 in P19 cells (Figs EV3B and EV3E).

5. Fig. 3F-G, it is important to include a treatment with RNaseH1 to show the specificity of the S9.6 signal. Also, the use of RNaseH1-dead mutant has been now shown to be a more reliable tool to detect R-loops than the S9.6 antibody by immunofluorescence (Crossley et al., JCB 2021; PMID: 34232287).

Response: Thank you for your comments. We agree with you that it's more reliable to detect R-loop by using RNase H1-dead mutant. Transfection of RNase H1-dead mutant is eminently suitable for detecting R-loops in cell lines *in vitro* (Chen L, et al. *Mol Cell* 2017. PMID: 29104020; Ginno PA, et al. *Mol Cell* 2012. PMID: 22387027; Legros P, et al. *PLoS Genet.* 2014. PMID: 25392932). However, it's hard to carry out in PGCs *in vivo*. Eliminating the S9.6 signals by RNase H1 treatment is widely used in IF of *in vitro* cultured cell lines, dot blot assay, and DRIP to show the specificity of S9.6 antibody. We have tried to include a group with RNase H1 treatment. Unfortunately, we failed in RNase H1 treatment on tissue slices of genital ridges after multiple attempts. References using RNase H1 treatment on fixed tissue slices were difficult to find in published literature, implying that RNase H1 treatment may not be suitable for fixed tissue slices. However, we have used RNase H1 overexpression in IF of MEFs and RNase H1 treatment in dot blot assay to show the specificity of the signals detected by the S9.6 antibody (Figs 4A and 4B).

6. It would have been nice to have a DDX5 knock out (or downregulation) control aside both immunofluorescence and Chip data to compare with. Ideally, one wished to see how Ddx5 or Dhx9 affects R-loops distribution genome-wide and compare it with that observed in MCM8 KO cells.

Response: Thank you for your suggestions. It has been reported that increased nuclear S9.6 signals both in immunostaining and blotting (Mersaoui SY, et al. *EMBO J.* 2019. PMID: 31267554) and the enrichment of R-loop signals at promoters, transcription start sites and gene bodies are displayed in DDX5-depleted cells (Villarreal OD, et al. *Life Sci Alliance.* 2020. PMID: 32747416; Sessa G, et al. *EMBO J.* 2021. PMID: 33634895). As you suggested, we have performed the S9.6 dot blot and the R-loop

CUT&Tag assay in DDX5-knockdown cells to compare the changes in R-loop levels and distribution genome-wide in DDX5-knockdown cells with those observed in MCM8 KO cells. The dot blot results showed that R-loop signals increased much more in DDX5-knockdown cells (Figs 4C and 4D), and the genome-wide analysis in MCM8-deficient cells and DDX5-knockdown cells showed that R-loop peak signals were both gained in the promoters, introns and intergenic regions (Fig EV5E). The percentage of R-loop gain peaks at the promoters in DDX5-knockdown cells was more than that in MCM8-deficient cells, suggesting that R-loop gain peaks after DDX5-knockdown were more likely to be located closer to the promoters (Fig EV5E). MCM8-deficient cells had 870 gain peaks that overlapped with the gain peaks of DDX5-knockdown cells and the majority of these gain peaks were overlaid with the promoters (Fig EV5F), as shown by the snapshots of two representative genes *Ttk* and *Apc* (Fig EV5G), suggesting that MCM8 and DDX5 coordinately control R-loop resolution at specific genomic loci, especially at promoters.

7. In Fig. 5C authors analyze replication dynamics by DNA fiber stretching and conclude that replication speed is slower in MCM8 KO cells, in line with previous reports demonstrating a function of MCM8 in contributing to replication fork progression (Maiorano et al., Cell 2005; Gambus et al., Cell Cycle 2013; Griffin et al., 2022). However, according to these data, it appears that MCM8 depletion reduced the length of both IdU and CIdU tracks, suggesting an effect on both the initiation and elongation steps of DNA synthesis. However, I am not convinced that this is due to R-loops accumulation, as claimed by the authors. The statistical test here and elsewhere is inappropriate (see other points below). Significance must be recalculated using a Man-Whitney test. Further, images shown in panel D are not representative of the quantification shown in panel E. Concerning the possible increase in R-loops upon MCM8 KO, this could be due to reduce DNA synthesis rate generating transcription-replication conflicts. Hence it is expected that inhibition of transcription should reduce R-loops formation, thus rescuing the effect of MCM8 KO. Fig. 5D-E, the reduction of 53BP1 foci in MCM8 KO cells treated with RNaseH1 is weak and quantification does not match the data.

Response: Thank you for your comments. In the DNA fiber assay, the IdU signal was used to indicate the active replication fork (RF), and the length of CIdU tracks of progressing RFs was used to evaluate RF velocity. As you mentioned, slower replication fork elongation may also contribute to the observed short CIdU length in MCM8 KO cells. However, when increasing RNase H1 activity in the nucleus alters phenotypes of interest, it could be reasoned that R-loops specifically contribute to these phenotypes. Therefore, we overexpressed RNase H1 to remove the R-loops and showed that the reduced replication progression and increased DNA damage levels

could be partially recovered in KO MEF, supporting that R-loop accumulation contributed to the decreased RF velocity and increased DNA damage in KO MEFs.

As you suggested, reducing R-loop levels by global inhibition of transcription, such as adding Actinomycin D (*Mosler T, et al. Nat Commun. 2021. PMID: 34916496*), is an alternative approach to determine whether DNA damage derives from unresolved R-loops. However, globally suppressing transcription output may affect the expression of important genes, especially those responsible for basic functions, and thus the results may be nonspecific and difficult to interpret.

As you mentioned, the statistical method we used was improper and the Man-Whitney test will be better. We have re-performed the statistical analysis as you suggested. We have also replaced the representative images in Fig. 4F in the revised manuscript, in which cells with 53BP1 > 5 foci were counted.

8. Fig 6, a western blot showing the expression of Flag-MCM8 must be shown. Panel E, a control including RnaseH1 treatment and ChIP with a non-specific antibody (Flag alone as a control for Flag-MCM8 Chip and non-specific for the S9.6 Chip) must be included to ascertain the specificity of the immunoprecipitated material. Panel F, it is critical here to show that equal amounts of IgGs were recovered in all lanes, since based on this experiment authors conclude that Ddx5 and Dhx9 retention is reduced on R-loops upon MCM8 KO.

Response: Thank you for your suggestions. We have shown the western blot result of the expression of FLAG-MCM8 (Appendix Fig S1A) related to Fig. 6. As you suggest, it's more precise to include a group using IgG as a control for Flag-MCM8 ChIP and S9.6 ChIP, and also include a control of RNase H1 for the S9.6 ChIP. Therefore, we have performed these experiments to ascertain the specificity of the immunoprecipitated material and re-analyzed the binding sites (Figs 6A-E; Appendix Fig S1). The results showed that 13.5% (1803/13303) of the MCM8 binding sites co-occurred with R-loop signals (Fig 6C) and these genes were significantly associated with the regulation of mitotic cell cycle phase transition as indicated by the GO analysis (Fig 6D), which were comparable with the results in the original manuscript. The slight discrepancy in values may be attributed to the utilization of the new CUT&Tag Assay Kit (Vazyme Biotech, TD904) and reference genome (human hg38). In Fig. 6F, we added the same amounts of antibodies to do the immunoprecipitation assay but didn't show the recovered amount of IgG in the original figure. We have supplemented IgG signals to illustrate this point.

9. Fig. 7, authors complement R-loops generated by MCM8 KO in HEK293T cells by expressing either wild-type or divers MCM8 mutants. Here again, the images provided in Fig. 7C do not match the quantification shown in panel E. The differences

in S9.6 signals observed in these images are very weak.

Response: Thank you for your comments. Because of the compression of IF images in the initially submitted manuscript, the S9.6 IF signals were weak and ambiguous. We have re-performed the immunofluorescence staining of R-loops and shown the representative images (Figs 7D and 7E) in the revised manuscript.

10. Fig. EV6, a quantification of the Ddx5 and Dhx9 immunofluorescence signals obtained must be shown.

Response: Thank you for your suggestion. We have quantified the DDX5 and DHX9 immunofluorescence signals (Appendix Fig S2C).

Other points

1. Page numbers are missing

Response: We have added the page numbers.

2. Introduction, second page, MCM2-7 protein are not degraded in S-phase, they are removed from chromatin by ongoing DNA synthesis (Todorov et al., J. Cell Biol. 1995). Next page, Ddx5 and Dhx9 are not "the" R-loops removing factors but are R-loops-removing factors. A general introduction to factors known so far in R-loops resolution (including other Ddxs helicases) is missing.

Response: Thank you for your suggestions and we have modified the statement. In the introduction section, we wrote that "It is reported that MCM8 binds to chromatin and functions as a DNA helicase to drive replication fork (RF) progression after the core replicative helicase subunit MCM2 is degraded.". In the previous study (*Natsume T, et al. Genes Dev. 2017. PMID: 28487407*), the authors induced the rapid degradation of MCM2 by auxin-inducible degron (AID) technology to explore the possible role of MCM8-9 in replication progression and found that MCM8-9 acted as an alternative replicative helicase to promote DNA synthesis. Therefore, we have revised the sentence to "It has been reported that MCM8 binds to chromatin and acts as a backup DNA helicase to facilitate replication fork (RF) progression when the core replicative helicase subunit MCM2 is deficient". In addition, we have changed the sentence "Ddx5 and Dhx9, the R-loop resolving factors" to "Ddx5 and Dhx9, R-loops-removing factors". What's more, we have supplemented the introduction of the known R-loop-resolving factors in the revised manuscript.

3. Fig. 3F-G, it is unclear what the white arrows indicate (whether cytoplasmic R-loops or cells displaying nuclear R-loops); this has not been described in the figure legend.

Response: We are sorry for this missing information. The white arrows indicate

representative cells displaying nuclear R-loops in Fig. 3F-G. The ambiguously labeled arrows were eliminated and a note was added to provide instructions on quantifying the fluorescence intensity in the figure legend.

4. Fig. 7F, what is the evidence that Ddx5 and Dhx9 function as dimers (or multimers) in R-loops resolution? It is entirely possible that resolution of a fraction of R-loops depends upon MCM8-Ddx5 and/or Dhx9 interaction. It is for this reason that it would have been nice to have a picture of R-loops distribution in Ddx or Dhx9 knock down cells.

Response: Thank you for your suggestion. It has been reported that DDX5 and DHX9 also work together in a complex to resolve R-loops (*Kim S, et al. Nucleic Acids Res. 2020. PMID: 32542338*). However, it was inaccurate in Fig. 7F to show DDX5 and DHX9 as dimers. We have amended our working model to show the mutual interactions of the three proteins and the reduction of DDX5 and DHX9 at R-loops without MCM8. Furthermore, as we mentioned in question 6 above, we have mapped the R-loop distribution in DDX5-knockdown cells (Fig EV5E) and conducted a comparable analysis of genome-wide R-loop accumulation in MCM8-deficient cells and DDX5-knockdown cells. The results suggest that MCM8 and DDX5 coordinately promote R-loop resolution at specific genomic loci, especially at promoters (Fig EV5F).

5. A Mann-Whitney U test is more appropriate to determine significance in several experiments (Fig. 3G; 5B-C; 7E).

Response: Thank you for your suggestion. As you suggest, we have carefully checked and modified the statistical analysis in these experiments (Fig. 3G; 4B, 4E; 6I; 7E).

We hope that these revisions and our answers are satisfactory. Please do not hesitate to contact me if additional information is needed. Thanks for your consideration.

Yours sincerely,

Yingying Qin

Dr. Yingying Qin
Shandong University
44 Wenhua Xi Road
Jinan, Shandong 250021
China

1st Mar 2024

Re: EMBOJ-2023-115128R
MCM8 interacts with DDX5 to promote R-loop resolution in primordial germ cells

Dear Dr. Qin,

Thank you for submitting your revised manuscript to The EMBO Journal. The original referees 1 and 3 have now reviewed it once more. While referee 1 is fully satisfied with the revisions, referee 3 still retains several major concerns that would require further presentational changes as well as strengthening of experimental data. I am afraid that since many of these points had already been raised during the first round of review, we are currently not able to proceed with acceptance and publication of the study. However, given that many other points have been adequately addressed at this stage, I would allow an exceptional second round of major revision in this case, so that you can deal with the remaining issues.

When preparing a re-revised manuscript, it would now also be essential to address several important editorial aspects, as stated in our revision guidelines:

- First of all, it is our policy that corresponding authors must indicate and use institutional email addresses for correspondence. This is however not the case for Yajuan Yang and Yingqing Qin, who are both using personal, external email addresses. Furthermore, as corresponding authors, it is essential that both specify an ORCID identifier in their respective profiles.
- Tables EV1/EV2 should be renamed to Dataset EV1/EV2, with references to them in the text adjusted accordingly.
- Appendix figures should not be uploaded individually, but only as part of a single Appendix PDF. Their figure legends should be removed from the main text and shown directly under each figure in the Appendix, for easy reference.
- Please rename the Conflict of Interest section into "Disclosure and Competing Interests Statement", in accordance with our updated Guide to Authors (<https://www.embopress.org/competing-interests>)
- As we are switching from a free-text author contribution statement towards a more formal statement based on Contributor Role Taxonomy (CRediT) terms, please remove the present Author Contribution section and instead specify each author's contribution(s) directly in the Author Information page of our submission system during upload of the final manuscript. See <https://casrai.org/credit/> for more information.
- Finally, please include suggestions for a short 'blurb' text prefacing and summing up the conceptual aspect of the study in two sentences (max. 250 characters), followed by 3-5 one-sentence 'bullet points' with brief factual statements of key results of the paper; they will form the basis of an editor-written 'Synopsis' accompanying the online version of the article. Please also upload a synopsis image, which can be used as a "visual title" for the synopsis section of your paper. The image (maybe based on Figure 7F?) should be in PNG or JPG format, and please make sure that it remains in the modest dimensions of (exactly) 550 pixels wide and 300-600 pixels high.

I am therefore returning the manuscript to you for a second, final round of revision, hoping that you will be able to provide the remaining requested changes and additions. Should you have any questions in this regards, please do not hesitate to contact me directly.

Yours sincerely,

Hartmut Vodermaier

9) Digital image enhancement is acceptable practice, as long as it accurately represents the original data and conforms to community standards. If a figure has been subjected to significant electronic manipulation, this must be clearly noted in the figure legend and/or the 'Materials and Methods' section. The editors reserve the right to request original versions of figures and the original images that were used to assemble the figure. Finally, we generally encourage uploading of numerical as well as gel/blot image source data; for details see: embopress.org/page/journal/14602075/authorguide#sourcedata

At EMBO Press, we ask authors to provide source data for the main manuscript figures. Our source data coordinator will contact you to discuss which figure panels we would need source data for and will also provide you with helpful tips on how to upload and organize the files.

In the interest of ensuring the conceptual advance provided by the work, we recommend submitting a revision within 3 months (30th May 2024). Please discuss the revision progress ahead of this time with the editor if you require more time to complete the revisions. Use the link below to submit your revision:

Link Not Available

Referee #1:

I think that the authors have addressed concerns raised by this reviewer in an appropriate manner.

Referee #3:

The authors have clearly made significant efforts to improve the data and satisfy most of my concerns. I still have some important concerns as listed below. A main point is the claim (as stated in the title and abstract) and the interpretation of the results, that MCM8 downregulation reduces proliferation of PGCs cells due to R-loops formation. However, evidence supporting this claim is weak, and, as pointed out by the authors, this is due to technical reasons that do not allow major molecular investigations in PGCs. Hence, I believe that authors must downplay this claim which can be left as likely possibility in the discussion, and sustain a role for MCM8 in R-loops formation through recruitment of DDX5 and DDX9 in other cell types (MEF, P19 and HEK293T). Further, in page 10, authors start the paragraph saying that "in order to investigate the molecular mechanism by which MCM8 maintains PGCs stability, we performed IP in P19 cells", which does not make sense. They should state that, due to technical reasons, the experiments were carried out in other cell types, and as such, restrain their finding in the specific cellular model they adopt.

Specific points

1. The fact that MCM8 KO induces DNA damage in PGCs may not necessarily be the cause that generates cell cycle delay. As I mentioned in my first review, to demonstrate that the cell cycle defect is due to DNA damage, authors have to show that this effect is reversible upon inhibition of the apical DNA damage kinases (ATM/ATR). Although authors state that DDR inhibition may result in PGCs cell death, on the contrary, it could rescue the cell death phenotype in MCM8, KO cells, I do not understand why the authors have not attempted to perform this experiment.
2. Point 3 of my previous review. The excel file listing the proteins identified by mass spectrometry is incomplete. The authors must list the proteins enriched in the MCM8 IP compared to the IgGs IP. Currently, the excel file includes the list of proteins in each IP separately, however it does not display the abundance of each identified protein in the MCM8 IP compared to the control IP. In the list of the 49 proteins overlapped in the R-loops base DDX9 has one of the lowest score..
3. Point 4 of my previous review, authors must run IP and Input on the same gel in order to compare the intensity of the signals, otherwise the result of the experiment cannot be interpreted. Original blots must be shown. As for the PCNA blot, in order to interpret this result, an MCM8-interacting protein must be shown on this gel, one may wonder that in this particular experiment DDX9 and/or DDX5 did not precipitate.
4. Figure 4B, a statistical test must be done between WT and MCM8 KO in the presence of RNaseH1 since this treatment also decreases fork speed in WT cells. This is critical in order to determine the contribution of MCM8 KO in reducing fork speed by R-loops formation.
5. Point 7 of my previous review, I believe that a treatment with a transcription inhibitor (cordycepin or other inhibitors) is mandatory here to conclude any effect of MCM8 KO on R-loops formation. This is a control which is commonly used in the field. 53BP1 foci quantification still does not match the images.
6. Figure 6E, as shown in Appendix Figure S1A, FLAG-MCM8 is clearly overexpressed compared to the control empty vector. This could be a concern in assessing the specificity of the immunoprecipitated R-loops. This point must be taken into account in the interpretation of the results. Further, Chip enrichment of FLAG-MCM8 compared to IgGs is weak, particularly on chromosome 5, as opposed to the previous result with no antibody, showing the importance of this control. What is the scale on the y-axis? What is the extent of this enrichment? Is this significant? Authors must clearly clarify these points to ascertain the significance of this critical result.

Minor point

Of note, authors state in the response to reviewer 1 the following "Since MCM8 depletion in human cells reduced chromatin loading of replication initiation MCM2-7 complexes (Volkening M, et al. Mol Cell Biol. 2005. PMID: 15684404 Volkening M, et al. Mol Cell Biol. 2005. PMID: 15684404)..... This result was not confirmed by several other laboratories (Maiorano et al., 2005; Gambus et al., Cell Cycle 2013; Griffin et al., 2022). These papers showed that MCM8 is not required to load MCM2-7 complexes on chromatin. Hence, this evidence cannot be used as a valid argument.

Dear Dr. Vodermaier,

Thank you very much for your consideration of our manuscript 'EMBOJ-2023-115128R'. We appreciate the referees for their valuable comments on our work. The comments of the referees have been responded to point by point and all changes have been highlighted in blue color within the document.

When preparing the revised manuscript, we have carefully addressed the key editorial aspects mentioned in your revision guidelines.

First, we have provided the institutional email addresses and ORCID identifiers for corresponding authors, Yajuan Yang and Yingying Qin, as stated: YangYJ0204@sdu.edu.cn; 0000-0002-4828-0967; qinyingying@sdu.edu.cn; 0000-0002-0319-7799.

Second, the Tables EV1/EV2 have been renamed into Datasets EV1/EV2, with references to them in the text adjusted accordingly.

Third, Appendix Figures S1-S3 and their figure legends have been merged as a single Appendix PDF.

Forth, we have renamed the Conflict of Interest section into "Disclosure and Competing Interests Statement".

Fifth, we have removed the Author Contribution section and specified the contribution of each author directly in the Author Information page of the submission system.

Finally, we have included a short "blurb" text prefacing and the conceptual aspect of the study in two sentences, followed by 3 one-sentence "bullet points" with brief factual statements of key results of the paper. We have also uploaded a synopsis image.

Below you will find our point-by-point responses to the referees' comments.

Referee #3:

The authors have clearly made significant efforts to improve the data and satisfy most of my concerns. I still have some important concerns as listed below. A main point is the claim (as stated in the title and abstract) and the interpretation of the results, that MCM8 downregulation reduces proliferation of PGCs cells due to R-loops formation. However, evidence supporting this claim is weak, and, as pointed out by the authors, this is due to technical reasons that do not allow major molecular investigations in

PGCs. Hence, I believe that authors must downplay this claim which can be left as likely possibility in the discussion, and sustain a role for MCM8 in R-loops formation through recruitment of DDX5 and DDX9 in other cells types (MEF, P19 and HEK293T). Further, in page 10, authors start the paragraph saying that "in order to investigate the molecular mechanism by which MCM8 maintains PGCs stability, we performed IP in P19 cells", which does not make sense. They should state that, due to technical reasons, the experiments were carried out in other cell types, and as such, restrain their finding in the specific cellular model they adopt.

Response: Thank you for your insightful comments. We acknowledge that the statement in our original manuscript were not as rigorous as it should be, and we appreciate your suggestions to improve its clarity and accuracy. We have carefully revised the relevant conclusions, including the title, abstract, results and discussion sections. We revised the conclusion as that MCM8 coordinates R-loop regulation by enhancing retention of R-loop resolving factors to protect genome stability from accumulated R-loops, which may be involved in PGC rapid proliferation and reproductive reserve establishment. As you mentioned, we revised our statements in description of IP-MS in P19 cells, and acknowledged the limitation of our study in the discussion. We have made every effort to ensure that the revised statements are more precise, reflecting the exact findings and implications of our study.

Specific points

1. The fact that MCM8 KO induces DNA damage in PGCs may not necessarily be the cause that generates cell cycle delay. As I mentioned in my first review, to demonstrate that the cell cycle defect is due to DNA damage, authors have to show that this effect is reversible upon inhibition of the apical DNA damage kinases (ATM/ATR). Although authors state that DDR inhibition may result in PGCs cell death, on the contrary, it could rescue the cell death phenotype in MCM8, KO cells, I do not understand why the authors have not attempted to perform this experiment.

Response: According to your suggestion, we made efforts to determine the effect of ATM inhibition on the development of MCM8-deficient PGCs due to the ATM activation shown in Figure 2G and 2H. The pregnant mice on embryonic day (E) 9.5 were intraperitoneally injected with the ATM kinase inhibitor (KU60019, 40mg/kg) for two successive days. On E11.5, pregnant mice were injected with EdU reagent (100 mg/kg) for 1 h before being euthanized. Subsequently, genital ridges were separated from embryos and used for immunostaining. As shown in the following figure, we found a significant reduction in PGC number in both wild-type and *Mcm8* knockout mice (A and B) after inhibition of ATM kinase activity, while the S-phase proportion remains relatively unchanged (C), suggesting that inhibition of the apical DNA damage kinase could not rescue cell cycle defect and PGC loss.

In response to this result, we reviewed the literature to find a possible explanation. Previous studies identified both replication-related DNA repair defects and increased chromatid breaks in cells treated with selective ATM kinase inhibitors (White JS, et al. *Sci Signal*. 2010. PMID: 20516478; Gamper AM, et al. *J Biol Chem*. 2012. PMID: 22362778). Early embryonic lethality of *Atm*^{KD/KD} mutant mice with normal protein

expression but absence of kinase activity (Yamamoto K, et al. J Cell Biol. 2012. PMID: 22869596), but not *Atm* knockout mice indicated that kinase inhibition led to significantly worse outcomes. Consistent with the different effects of ATM deletion and kinase inhibition reported (Yamamoto K, et al. J Cell Biol. 2012. PMID: 22869596), treatment with KU60019 might cause cell death of PGCs, as shown by a significant reduction in PGC number in wild-type embryos after ATM kinase inhibition. Besides, germ cells of neonatal testes in *Fancm* mutant mice were increased after crossing with *Atm* knockout mice (Luo Y, et al. PLoS Genet. 2014. PMID: 25010009). Based on these findings, it would be more appropriate to use *Atm* knockout mice to explore the potential effects of DDR inhibition on the phenotype observed in MCM8-deficient PGCs. Based on these results, we could not establish a causal relationship between DNA damage and proliferation defect of *Mcm8* knockout PGCs. Therefore, we have revised the statements in our manuscript to ensure that our interpretations are not overstated. We would prefer not to include this result in the manuscript and will explore the exact mechanisms in future studies.

2. Point 3 of my previous review. The excel file listing the proteins identified by mass spectrometry is incomplete. The authors must list the proteins enriched in the MCM8 IP compared to the IgGs IP. Currently, the excel file includes the list of proteins in each IP separately, however it does not display the abundance of each identified protein in the MCM8 IP compared to the control IP. In the list of the 49 proteins overlapped in the R-loops base Dhx9 has one of the lowest score.

Response: We apologize for the missing information in the supplementary table (Dataset EV1) and we have included more details on the abundance of each identified protein in the MCM8 IP compared to the control IP.

The amount of protein that binds with a particular antibody depends on the cellular abundance of the interacting components as well as on the stoichiometry of the individual complex (Hein MY, et al. Cell. 2015. PMID: 26496610; Rogawski R, Sharon M. Chem Rev. 2022. PMID: 34406752). Besides, many low abundant proteins, which may constitute real interactors, are missed in analyses of protein interactomics, because modern mass spectrometers typically have a detection limit of 1×10^{-18} moles for peptides and a dynamic range of 4 orders of magnitude (Timp W, Timp G. Sci Adv. 2020. PMID: 31950079; Zubarev RA. Proteomics. 2013. PMID: 23307342). Despite the lower ranking of DHX9, we could not exclude the possibility that it plays a crucial role in biological processes. What's more, we have conducted IP followed by western blot analysis to confirm the interaction between MCM8 and DHX9 both in P19 cells and PGCs. The results supported the interaction between MCM8 and DHX9, providing further evidence to verify the proteomic results.

3. Point 4 of my previous review, authors must run IP and Input on the same gel in order to compare the intensity of the signals, otherwise the result of the experiment cannot be interpreted. Original blots must be shown. As for the PCNA blot, in order to interpret this result, an MCM8-interacting protein must be shown on this gel, one may wonder that in this particular experiment Dhx9 and/or DDX5 did not precipitate.

Response: We apologize for the confusion caused by our original presentations of the IP results. As you concerned, we have now addressed this issue by running IP and Input samples on the same gel and replaced the relevant figures including Fig. 3C-E, EV3B-E and 6F. Because of the large number of samples that exceeded the capacity of our gel, we ran the IP and input samples into different gels and presented them as Fig. 7B and 7C. However, all the IP groups that were compared belonged to the same gel, which is also acceptable. All the original blot images have been shown as an attachment. Regarding the PCNA blot, we have included an additional exposure for DDX5 on this gel in Fig. EV3C.

Figure 3

Figure EV3

Figure 6

4. Figure 4B, a statistical test must be done between WT and MCM8 KO in the presence of RNaseH1 since this treatment also decreases fork speed in WT cells. This is critical in order to determine the contribution of MCM8 KO in reducing fork speed by R-loops formation.

Response: Thank you for your suggestion. We have included the statistical data of WT and KO cells after RNH1 treatment in the revised figures. In agreement with the previous study reporting the incomplete removal of R-loops by overexpression of RNH1 (Templeton CW, et al. Proc Natl Acad Sci U S A. 2023. PMID: 37611058), the partial removal of R-loops might explain the partial recovery of the fork speed observed in MCM8 KO cells (Fig 4B and 4E). However, there may be other mechanisms that affect the replication speed in MCM8 KO cells. Therefore, we have revised the statements in our manuscript to demonstrate that “The partial removal of R-loops by overexpression of RNase H1 might explain the partial recovery of the fork speed observed in MCM8 KO cells (Fig 4B and 4E). However, besides R-loop accumulation, other mechanisms may also be involved in the decreased replication speed in MCM8 KO cells”.

5. Point 7 of my previous review, I believe that a treatment with a transcription inhibitor (cordycepin or other inhibitors) is mandatory here to conclude any effect of MCM8 KO on R-loops formation. This is a control which is commonly used in the field. 53BP1 foci quantification still does not match the images.

Response: Thank you for your suggestion. We have conducted additional experiments using a transcription inhibitor (cordycepin) to further investigate the role of R-loops in MCM8 KO cells by assessment of the transcriptional levels, R-loop levels, replication fork speed and DNA damage. The results showed that inhibiting R-loop formation using cordycepin improved both replication fork speed and reduced DNA damage levels in MCM8 KO cells, which supported our conclusion that R-loops, at least partially contributed to the slowed replication speed and increased DNA damage observed in MCM8 KO cells. Additionally, we have carefully checked our data and presented the most representative images of 53BP1 foci in the revised manuscript.

Figure 4

F

6. Figure 6E, as shown in Appendix Figure S1A, FLAG-MCM8 is clearly overexpressed compared to the control empty vector. This could be a concern in assessing the specificity of the immunoprecipitated R-loops. This point must be taken into account in the interpretation of the results. Further, Chip enrichment of FLAG-MCM8 compared to IgGs is weak, particularly on chromosome 5, as opposed to the previous result with no antibody, showing the importance of this control. What is the scale on the y-axis? What is the extent of this enrichment? Is this significant? Authors must clearly clarify these points to ascertain the significance of this critical result.

Response: Thank you for your comments. The immunoprecipitated R-loop peaks in Figure 6E were obtained in wild-type HEK293 cells. We have described clearly in the revised manuscript. In our previous revised manuscript, we have included an IgG control and analyzed the immunoprecipitated peaks for each group. The y-axis in the IGV visualization represents the number of reads normalized to one million reads. As you concerned, the “IgG” group showed some enrichment of non-specific peaks. Therefore, we have re-evaluated the enrichment profiles and selected peaks that exhibit at least 2-fold in the experimental groups (FLAG-MCM8, S9.6) compared to the IgG control. In this case, the peaks on chromosome 5 as shown at *APC* gene site were not significant for FLAG-MCM8 enrichment, but the peak at *FBXO4* was significant. So we re-analyzed the genomic distribution of the experimental groups and genomic co-localization between the FLAG-MCM8 group and the S9.6 group. It's worth noting that the modified results in Figure 6A-E and Appendix Figure S2B in the revised manuscript also corroborated our initial findings.

Figure 6

Appendix figure S2

Minor point

Of note, authors state in the response to reviewer 1 the following "Since MCM8 depletion in human cells reduced chromatin loading of replication initiation MCM2-7 complexes (Volkening M, et al. Mol Cell Biol. 2005. PMID: 15684404)..... This result was not confirmed by several other laboratories (Maiorano et al., 2005; Gambus et al., Cell Cycle 2013; Griffin et al., 2022). These papers showed that MCM8 is not required to load MCM2-7 complexes on chromatin. Hence, this evidence cannot be used as a valid argument.

Response: Thank you for your comments. We appreciate you for pointing out the inaccuracy in our previous statement. It is known that CDC45-MCM2-7-GINS (CMG) helicase assembly is the central event in eukaryotic replication initiation. As you pointed out, the role of the MCM8/9 complex on the loading of the MCM2-7 complex was controversial. Depletion of MCM8 and MCM9 in *Xenopus* egg extracts has no impact on MCM2-7 complex loading, but reduces the levels of chromatin-bound CDC45 and Psf2 (GINS Complex Subunit 2) (Gambus et al. Cell Cycle. 2013. PMID: 23518502), suggesting that there are fewer assembled active forks or reduced stability

of replicative helicases. Therefore, we have revised the statement into “Since MCM8 depletion could reduce chromatin loading of some replication initiation proteins (Gambus et al., 2013), we speculated that a large number of unscheduled R-loop signals and loss of shared signals with WT in KO cells were relevant to the repositioned RNA polymerases or the changed distribution of replication origins.”.

We hope that these revisions and our answers are satisfactory. Please do not hesitate to contact me if additional information is needed. Thanks for your consideration.

Yours sincerely,
Yingying Qin

Dr. Yingying Qin
Shandong University
44# Wenhua Xi Road
Jinan, Shandong 250012
China

21st May 2024

Re: EMBOJ-2023-115128R1
MCM8 interacts with DDX5 to promote R-loop resolution

Dear Dr. Qin,

Thank you for submitting your final revised manuscript for our consideration. I am pleased to inform you that we have now accepted it for publication in The EMBO Journal.

Yours sincerely,

Hartmut Vodermaier
